# Decentralized Finite-Sum Optimization over Time-Varying Networks

## Abstract

We consider decentralized time-varying stochastic optimization problems where each of the functions held by the nodes has a finite sum structure. Such problems can be efficiently solved using variance reduction techniques. Our aim is to explore the lower complexity bounds (for communication and number of stochastic oracle calls) and find optimal algorithms. The paper studies strongly convex and nonconvex scenarios. To the best of our knowledge, variance reduced schemes and lower bounds for time-varying graphs have not been studied in the literature. For nonconvex objectives, we obtain lower bounds and develop an optimal method GT-PAGE. For strongly convex objectives, we propose the first decentralized time-varying variance-reduction method ADOM+VR and establish lower bound in this scenario, highlighting the open question of matching the algorithms complexity and lower bounds even in static network case.

## 1 Introduction

We consider a sum-type problem

$$\min_{x \in \mathbb{R}^d} \ F(x) := \sum_{i=1}^{m} F_i(x), \tag{1}$$

where $F_i(x) = \frac{1}{n} \sum_{j=1}^{n} f_{ij}(x)$. We assume that for each $i = 1, \ldots, m$ the set of functions $\{f_{ij}\}_{j=1}^{n}$ is stored at node $i$. Decentralized optimization has applications in power system control (Ram et al., 2009; Gan et al., 2012), distributed statistical inference (Forero et al., 2010; Nedić et al., 2017), vehicle coordination and control (Ren and Beard, 2008), distributed sensing (Rabbat and Nowak, 2004; Bazerque and Giannakis, 2009). In most scenarios, the data is generated in a distributed way. In applications such as federated learning (Konečný et al., 2016; McMahan et al., 2017), centralized data processing is not allowed by privacy constraints.

In this paper, we focus on time-varying networks. That is, between consequent data exchanges, the topology of communication graph may change (Zadeh, 1961; Nedić, 2020). The set of nodes remains the same, while the set of edges changes. The instability of links practically happens due to malfunctions in communication, such as a loss of wireless connection between sensors or drones.

**Our Contribution**. We propose lower bounds in the strongly convex and nonconvex case, an optimal algorithm in the nonconvex case, and an algorithm in the strongly convex case with an open question about its optimality.

1. We propose a method for decentralized finite-sum optimization over time-varying graphs ADOM+VR (Algorithm 1). The method is based on the combination of ADOM+ algorithm for non-stochastic decentralized optimization over time-varying networks (Kovalev et al., 2021a) and loopless Katyusha approach for finite-sum problems (Kovalev et al., 2020a).

2. For nonconvex decentralized optimization over time-varying graphs, we propose an optimal algorithm GT-PAGE (Algorithm 2). The main idea is to implement the PAGE gradient

estimator (Li et al., 2021) for finite-sum problem into the gradient tracking (Nedic et al., 2017).

3. We give lower complexity bounds for decentralized finite-sum optimization over time-varying networks for strongly-convex (Theorem 4.3) and nonconvex (Theorem 4.5) objectives with taking into account the sensivity of smoothness constants from Assumptions 2.1, 2.2 and 2.3

| Algorithm | Comp. | Comm. |
|---|---|---|
| ADOM (Kovalev et al., 2021b) | $n\sqrt{\frac{L}{\mu}}$ | $\chi\sqrt{\frac{L}{\mu}}$ (dual) |
| ADOM+ (Kovalev et al., 2021a) | $n\sqrt{\frac{L}{\mu}}$ | $\chi\sqrt{\frac{L}{\mu}}$ |
| Acc-GT (Li and Lin, 2021) | $n\sqrt{\frac{L}{\mu}}$ | $\chi\sqrt{\frac{L}{\mu}}$ |
| ADFS (Hendrikx et al., 2021) | $n+\sqrt{n\max_i \frac{\overline{L}_i}{\mu_i}}$ | $\sqrt{\chi}\sqrt{\max_i \frac{L_i}{\mu_i}}$ (static + dual) |
| Acc-VR-EXTRA (Li et al., 2020) | $n+\sqrt{n\frac{\overline{L}}{\mu}}$ | $\sqrt{\chi\frac{L}{\mu}}$ (static) |
| ADOM+VR Alg. 1, this paper | $n+\sqrt{n\frac{\overline{L}}{\mu}}$ | $\chi\sqrt{\frac{L}{\mu}}$ |
| Lower bound Th. 4.3, this paper | $n+\sqrt{n\max_i \frac{\overline{L}_i}{\mu_i}}$ | $\chi\sqrt{\max_i \frac{L_i}{\mu_i}}$ |

Table 1: Computational (the number of stochastic oracle calls per node) and communication complexities of decentralized methods for finite-sum **strongly convex** optimization over time-varying graphs. $O(\cdot)$ notation and $\log(1/\varepsilon)$ factor are omitted. Comment "static" means that the method only works over time-static networks. Comment "dual" means that the method is dual-based. For notation, see Section 2.

| Algorithm | Comp. | Comm. |
|---|---|---|
| GT-SAGA (Xin et al., 2021b) | $\left(1+\frac{n^{2/3}}{m^{1/3}}+\sqrt{n}\right)\frac{L_s\Delta}{\varepsilon^2}$ | $\sqrt{\chi}\left(1+\frac{n^{2/3}}{m^{1/3}}+\sqrt{n}\right)\frac{L_s\Delta}{\varepsilon^2}$ (static) |
| GT-SARAH (Xin et al., 2022) | $(m+\sqrt{nm}+n^{1/3}m^{2/3})\frac{L_s\Delta}{\varepsilon^2}$ | $\sqrt{\chi}\frac{L_s\Delta}{\varepsilon^2}$ (static) |
| DESTRESS (Li et al., 2022a) | $n+\frac{\sqrt{n}L_s\Delta}{\varepsilon^2}$ | $\sqrt{\chi}\left(\sqrt{mn}+\frac{L_s\Delta}{\varepsilon^2}\right)$ (static) |
| DEAREST (Luo and Ye, 2022) | $n+\frac{\sqrt{n}\hat{L}\Delta}{\varepsilon^2}$ | $\sqrt{\chi}\frac{\hat{L}\Delta}{\varepsilon^2}$ (static) |
| GT-PAGE Alg. 2, this paper | $n+\frac{\sqrt{n}\hat{L}\Delta}{\varepsilon^2}$ | $\chi\frac{L\Delta}{\varepsilon^2}$ |
| Lower bound Th. 4.5, this paper | $n+\frac{\sqrt{n}\hat{L}\Delta}{\varepsilon^2}$ | $\chi\frac{L\Delta}{\varepsilon^2}$ |

Table 2: Computational (the number of stochastic oracle calls per node) and communication complexities of decentralized methods for finite-sum **nonconvex** optimization over time-varying graphs. $O(\cdot)$ notation is omitted. Comment "static" means that the method only works over time-static networks. Here $L_s = \max_{i,j} L_{ij}$ from Assumption 2.1, $L$ from Assumption 2.2 and $\hat{L}$ from Assumption 2.3. For notation, see Section 2.

**Related Work**. Decentralized optimization over static and time-varying networks has been actively developing in recent years. In (Scaman et al., 2017), dual-based methods and lower bounds for (non-stochastic) strongly convex optimization over static graphs were proposed. Optimal primal methods were obtained in (Kovalev et al., 2020b). For time-varying networks, non-accelerated primal (Nedic et al., 2017) and dual (Maros and Jaldén, 2018) methods were proposed. After that, accelerated algorithms were given in (Kovalev et al., 2021b) for dual oracle and in (Kovalev et al., 2021a; Li and Lin, 2021) for primal oracle. These methods match the lower complexity bounds for time-varying graphs developed in (Kovalev et al., 2021a).

Our paper is devoted to variance reduced schemes. Classical variance reduction methods such as SAGA (Defazio et al., 2014) and SVRG (Johnson and Zhang, 2013) allow to

enhance the rates for stochastic optimization problems with finite-sum structure. Accelerated variance reduced schemes require adding a negative momentum, also referred to as Katyusha momentum (Allen-Zhu, 2017). Considering nonconvex problems, recent development starts with (Reddi et al., 2016) and (Allen-Zhu and Hazan, 2016), where algorithms based on SVRG were proposed. More recently, other modifications of SVRG scheme with the same gradient complexity $\mathcal{O}\left(n + n^{2/3}/\epsilon^2\right)$ were proposed in (Li and Li, 2018), (Ge et al., 2019) and (Horváth and Richtárik, 2019). Moreover, optimal algorithms was presented, such as Spider (Fang et al., 2018), SNVRG (Zhou et al., 2020), methods based on SARAH (Nguyen et al., 2017) (e.g. SpiderBoost (Wang et al., 2018), ProxSARAH (Pham et al., 2020), Geom-SARAH (Horváth et al., 2022)) and PAGE (Li et al., 2021), which have $\mathcal{O}\left(n + \sqrt{n}/\epsilon^2\right)$ gradient estimation complexity.

In strongly-convex decentralized optimization over static graphs, optimal dual variance reduced method ADFS was proposed in (Hendrikx et al., 2019). The corresponding lower bounds were provided in (Hendrikx et al., 2021). In the narrower setting in (Li et al., 2022b), the Acc-VR-EXTRA algorithm was introduced. To the best of our knowledge, the optimality of this algorithm remains an open question. For variational inequalities, variance reduction is also applicable (Alacaoglu and Malitsky, 2022). Moreover, several methods for decentralized finite-sum variational inequalities were proposed in (Kovalev et al., 2022) both for static and time-varying networks. See an overview of methods for strongly-convex objectives in Table 1.

In the nonconvex case, the result of first application of variance reduction and gradient tracking to decentralized optimization for static graphs was the method D-GET (Sun et al., 2020). Later, algorithms GT-SAGA (Xin et al., 2021b), GT-HSGD (Xin et al., 2021a) and GT-SARAH (Xin et al., 2022) were proposed, which improve the complexity of communication rounds and local computations comparing to D-GET. A relatively new result was achieved by the method DESTRESS (Li et al., 2022a), which is optimal in terms of local computations, but ineffective in terms of number of communications in case of static graphs. This method was improved into DEAREST (Luo and Ye, 2022), which is optimal. Nevertheless, the application of variance reduction has not been studied for the case of nonconvex decentralized optimization over time-varying graphs. In Table 2 we present an overview of methods for which it is possible to explicitly write out complexities in terms of constants of smoothness and $\chi$. For an overview of other algorithms, see Table 1 in (Xin et al., 2022) and Table 1 in (Xin et al., 2021a).

*Remark* 1.1. It is necessary to clarify that optimality of DESTRESS and DEAREST is not clear in terms of dependence on smoothness constants. Indeed, mentioned constants $L_s, \hat{L}$ and $L$ are sensitive to $n$. In Appendix D.5 we show that ratios $\sqrt{n}L = \hat{L}$ and $nL = L_s$ can be achieved.

**Paper Organization**. We organize the paper as follows. In Section 2, we introduce notation and assumptions on the objectives and communication network. In Section 3, we describe our methods and give complexity results. Section 3.2 describes ADOM+VR for strongly convex objectives and Section 3.3 covers GT-PAGE for nonconvex objectives. Lower bounds are provided in Section 4. Finally, in Section 6 we give concluding remarks.

## 2 NOTATION AND ASSUMPTIONS

Throughout this paper, we adopt the following notations: We denote by $||\cdot|| = ||\cdot||_2$ the norm in $L_2$ space. The Kronecker product of two matrices is denoted as $A \otimes B$. We use $\mathcal{D}(X)$ to denote some distribution over a finite set $X$. The sets of batch indices are denoted by $S$, expressed as $S = (\xi^1, \ldots, \xi^b)$, where $\xi^j$ is a tuple of $m$ elements, each corresponding to a node, specifically $\xi^j = (\xi_1^j, \ldots, \xi_m^j)$, with $\xi_i^j$ being the index of the local function on $i$-th node in $j$-th element of the batch. Also for each $i = 1, \ldots, m$ define $S_i = (\xi_i^1, \ldots, \xi_i^b)$. Each node maintains its own copy of a variable corresponding to a specific variable in the algorithm. The variables in the algorithm are aggregations of the corresponding node variables:

$$x = (x_1, x_2, \ldots, x_m) \in \mathbb{R}^{d \times m}.$$

With a slight abuse of notation we will denote $F(x) = \sum_{i=1}^m F_i(x_i)$ and $\nabla F(x) = (\nabla F_1(x_1), \ldots, \nabla F_m(x_m))$. Linear operations and scalar products are performed component-

wise in a decentralized way. Let us introduce an auxiliary subspace $\mathcal{L} = \{x \in \mathbb{R}^{d \times m} | x_1 = \ldots = x_m\}$, respectively $\mathcal{L}^\perp = \{x \in \mathbb{R}^{d \times m} | x_1 + \ldots + x_m = 0\}$. We also let $x^* = \arg\min_{x \in \mathbb{R}^d} F(x)$ or $x^* = \arg\min_{x \in \mathbb{R}^{d \times m}} F(x)$, depending on the context.

Let us pass to assumptions on objective functions. Firstly, we assume that objectives are smooth, which is a standard assumption for optimization. We introduce different concepts of smoothness: Assumptions 2.1, 2.2 and 2.4 are used in Algorithm 1; Assumptions 2.2 and 2.3 are for Algorithm 2.

**Assumption 2.1.** For each $i = 1, \ldots, m$ and $j = 1, \ldots, n$ function $f_{ij}$ is convex and $L_{ij}$-smooth, i.e. $\|\nabla f_{ij}(y) - \nabla f_{ij}(x)\| \leq L_{ij}\|y - x\|$. For each $i = 1, \ldots, m$ let us define $\overline{L}_i = \frac{1}{n}\sum_{j=1}^n L_{ij}$, $\overline{L} = \max_i \{\overline{L}_i\}$.

**Assumption 2.2.** For each $i = 1, \ldots, m$ function $F_i$ is $L$-smooth, i.e. $\|\nabla F_i(y) - \nabla F_i(x)\| \leq L\|y - x\|$.

Note that in the context of Assumption 2.1 and Assumption 2.2, the following holds for the smallest possible $L_{ij}$ and $L$: $L \leq \overline{L} \leq nL$.

In the next assumption, we introduce average smoothness constants. That is used in analysis of Algorithm 2.

**Assumption 2.3.** For each $i = 1, \ldots, m$ function $F_i$ is $\hat{L}$-average smooth, i.e. $\frac{1}{n}\sum_{j=1}^n \|\nabla f_{ij}(y) - \nabla f_{ij}(x)\|^2 \leq \hat{L}^2\|y - x\|^2$.

Finally, we introduce an assumption on strong convexity

**Assumption 2.4.** For each $i = 1, \ldots, m$ function $F_i$ is $\mu$-strongly convex, i.e. $F_i(y) \geq F_i(x) + \langle \nabla F_i(x), y - x \rangle + \frac{\mu}{2}\|y - x\|_2^2$.

Decentralized communication is represented by a sequence of graphs $\{\mathcal{G}^k = (\mathcal{V}, \mathcal{E}^k)\}_{k=0}^\infty$. With each graph, we associate a gossip matrix $\mathbf{W}(k)$.

**Assumption 2.5.** For each $k = 0, 1, 2, \ldots$ it holds 1) $[\mathbf{W}(k)]_{i,j} \neq 0$ if and only if $(i, j) \in \mathcal{E}^k$ or $i = j$, 2) $\ker \mathbf{W}(k) \supset \{(x_1, \ldots, x_n) \in \mathbb{R}^n : x_1 = \ldots = x_n\}$, 3) $\mathrm{range}\,\mathbf{W}(k) \subset \{(x_1, \ldots, x_n) \in \mathbb{R}^n : \sum_{i=1}^n x_i = 0\}$, 4) There exists $\chi \geq 1$, such that

$$\|\mathbf{W}(k)x - x\|^2 \leq (1 - \chi^{-1})\|x\|^2 \text{ for all } x \in \{(x_1, \ldots, x_n) \in \mathbb{R}^n : \sum_{i=1}^n x_i = 0\}. \quad (2)$$

In particular, matrices $\mathbf{W}(k)$ can be chosen as $\mathbf{W}(k) = \mathbf{L}(\mathcal{G}^k)/\lambda_{\max}(\mathbf{L}(\mathcal{G}^k))$, where $\mathbf{L}(\mathcal{G}^k)$ denotes a graph Laplacian matrix. Moreover, if the network is constant ($\mathcal{G}^k \equiv \mathcal{G}$), we have $\chi = \lambda_{\max}(\mathbf{L}(\mathcal{G}))/\lambda_{\min}^+(\mathbf{L}(\mathcal{G}))$, i.e. $\chi$ equals the graph condition number.

## 3 ALGORITHMS

In this section, we propose new methods for decentralized finite-sum optimization: Algorithm 1 for strongly convex case and optimal Algorithm 2 for nonconvex case. Both algorithms use a variance reduction technique. The main idea of variance reduced methods is a special gradient estimator. The estimator is computed w.r.t. a snapshot of the full gradient. If the objective is a sum of $q$ functions, one recomputes the full gradient (over all samples) once in $O(q)$ iterations (Johnson and Zhang, 2013; Allen-Zhu, 2017). In a loopless approach (Kovalev et al., 2020a) the full gradient is computed with a probability of order $O(1/q)$ at each iteration. In this paper, we use the latter technique.

We measure the complexity in two ways: number of communications and number of stochastic oracle calls. The computational complexity of the algorithm iterations can be controlled using mini-batching of the gradient. That is, we take $b$ gradient estimations and average them. If the batch size is large, the number of algorithm iterations decreases, but the number of oracle calls per iteration is increased by $b$ times. In Katyusha (Allen-Zhu, 2017) it is shown that an optimal batch size is $b \sim \sqrt{n}$. In the analysis of Algorithms 1 and 2, we obtain optimal batch sizes, as well.

### 3.1 Multi-Stage Consensus

There is a universal way to divide oracle and communication complexities of a decentralized optimization method. Instead of performing one synchronized communication, let us perform several iterations in a row. Following (Kovalev et al., 2021a), we introduce

$$\mathbf{W}(k;T) = \mathbf{I}_m - \Pi_{q=kT}^{(k+1)T-1}(\mathbf{I}_m - \mathbf{W}(q))$$

It can be shown that if we take $T = \lceil \chi \rceil$, then condition number of $\mathbf{W}(k;T)$ reduces to $O(1)$. To see that, note that for all $x \in \mathcal{L}^\perp$ it holds

$$\|\mathbf{W}(k;T)x - x\|^2 \le (1 - \chi^{-1})^T \|x\|^2 \le \exp(-T\chi^{-1}) \le e^{-1}.$$

In other words, by using multi-stage consensus we reduce $\chi$ to $O(1)$ by paying a $\lceil \chi \rceil$ times more communications per iteration.

*Remark* 3.1. For static networks, Chebyshev acceleration replaces multi-stage consensus (Scaman et al., 2017). Term $\chi$ in complexity is reduced to $O(1)$ at the cost of performing $\lceil \sqrt{\chi} \rceil$ communications per iteration. (Static) gossip matrix $\mathbf{W}$ is replaced by a Chebyshev polynomial $P(\mathbf{W})$.

### 3.2 Strongly Convex Case

For the strongly convex case, we take ADOM+ (Kovalev et al., 2021a) as a base decentralized method. We also use a gradient estimator averaged over a mini-batch and a negative Katyusha momentum (Allen-Zhu, 2017; Kovalev et al., 2020a).

Let us briefly discuss the idea of ADOM+. The given optimization problem can be written in decentralized form as follows:

$$\min_{x \in \mathcal{L}} F(x).$$

This can be further reformulated as follows, which is the basis for the ADOM+ method:

$$\min_{x \in \mathbb{R}^{d \times m}} \max_{y \in \mathbb{R}^{d \times m}} \max_{z \in \mathcal{L}^\perp} \left[ F(x) - \frac{\nu}{2} \|x\|^2 \right.$$
$$\left. - \langle y, x \rangle - \frac{1}{2\nu} \|y + z\|^2 \right].$$

It is not difficult to show that in case $\nu < \mu$, this saddle point problem is strongly convex, which means that it has a single solution $(x^*, y^*, z^*)$ satisfying the optimality conditions:

$$0 = \nabla F(x^*) - \nu x^* - y^*, \quad (3)$$
$$0 = \nu^{-1}(y^* + z^*) + x^*, \quad (4)$$
$$0 \ni y^* + z^*. \quad (5)$$

---

**Algorithm 1** ADOM+VR

1: **input:** $x^0, y^0, m^0, \omega^0 \in (\mathbb{R}^d)^\mathcal{V}$, $z^0 \in \mathcal{L}^\perp$
2: $x_f^0 = \omega^0 = x^0$, $y_f^0 = y^0$, $z_f^0 = z^0$
3: **for** $k = 0, 1, \ldots, N-1$ **do**
4: $\quad x_g^k = \tau_1 x^k + \tau_0 \omega^k + (1 - \tau_1 - \tau_0)x_f^k$
5: $\quad S_i^k \sim \mathcal{D}_i^b(\{1, 2, \ldots, n\}), p_{ij} = \frac{L_{ij}}{nL_i}$
6: $\quad (\nabla^k)_i = \frac{1}{b} \sum_{j \in S_i^k} \frac{1}{np_{ij}} \left[ \nabla f_{ij}(x_{g,i}^k) - \nabla f_{ij}(\omega_i^k) \right]$
$\qquad\qquad + \nabla F_i(\omega_i^k)$
7: $\quad x^{k+1} = x^k + \eta\alpha(x_g^k - x^{k+1})$
$\qquad\qquad - \eta \left[ \nabla^k - \nu x_g^k - y^{k+1} \right]$
8: $\quad x_f^{k+1} = x_g^k + \tau_2(x^{k+1} - x^k)$
9: $\quad \omega_i^{k+1} = \begin{cases} x_{f,i}^k, & \text{with prob. } p_1 \\ x_{g,i}^k, & \text{with prob. } p_2 \\ \omega_i^k, & \text{with prob. } 1 - p_1 - p_2 \end{cases}$
10: $\quad y_g^k = \sigma_1 y^k + (1 - \sigma_1)y_f^k$
11: $\quad y^{k+1} = y^k + \theta\beta(\nabla^k - \nu x_g^k - y^{k+1})$
$\qquad\qquad - \theta \left[ \nu^{-1}(y_g^k + z_g^k) + x^{k+1} \right]$
12: $\quad y_f^{k+1} = y_g^k + \sigma_2(y^{k+1} - y^k)$
13: $\quad z_g^k = \sigma_1 z^k + (1 - \sigma_1)z_f^k$
14: $\quad z^{k+1} = z^k + \gamma\delta(z_g^k - z^k)$
$\qquad\qquad - (\mathbf{W}(k) \otimes \mathbf{I}_d) \left[ \gamma\nu^{-1}(y_g^k + z_g^k) + m^k \right]$
15: $\quad m^{k+1} = \gamma\nu^{-1}(y_g^k + z_g^k) + m^k$
$\qquad\qquad - (\mathbf{W}(k) \otimes \mathbf{I}_d) \left[ \gamma\nu^{-1}(y_g^k + z_g^k) + m^k \right]$
16: $\quad z_f^{k+1} = z_g^k - \zeta(\mathbf{W}(k) \otimes \mathbf{I}_d)(y_g^k + z_g^k)$
17: **end for**
18: **return** $x^N$

---

The idea is described in more detail in (Kovalev et al., 2021a).

Let us discuss the gradient estimator for strongly convex setup. Consider a minimization problem $\min_{x \in \mathbb{R}^d} g(x) = \frac{1}{q} \sum_{i=1}^q g_i(x)$. At step $k$, instead of the gradient $\nabla g(x^k)$ one uses an estimator

$$\nabla^k = \frac{1}{b} \sum_{i \in S} [\nabla g_i(x^k) - \nabla g_i(w^k)] + \nabla g(w^k), \quad (6)$$

where $S$ is a random batch of indices of size $b$, $x^k$ is the current iterate and $w^k$ is a reference point at which the full gradient is computed. Gradient estimator (6) is used in such methods as SVRG (Johnson and Zhang, 2013) and Katyusha (Allen-Zhu, 2017).

**Theorem 3.2.** *Let Assumptions 2.1, 2.2, 2.4, 2.5 and $b \geq \overline{L}/L$ hold. Then Algorithm 1 requires $N$ iterations to yield $x^N$ such that $\|x^N - x^*\|^2 \leq \varepsilon$, where*

$$N = \mathcal{O}\left(\left(\frac{n}{b} + \left(\frac{\sqrt{n}}{b} + \frac{n\overline{L}}{b^2 L} + \chi\right)\sqrt{\frac{L}{\mu}}\right)\log\frac{1}{\epsilon}\right).$$

**Corollary 3.3.** *In the setting of Theorem 3.2, with $b \sim \max\left\{\sqrt{n\overline{L}/L}, n\sqrt{\mu/L}\right\}$ and the number of communications per iteration $\sim \chi$, the algorithm requires*

$$\mathcal{O}\left(n + \sqrt{\frac{n\overline{L}}{\mu}}\right)\log\frac{1}{\epsilon} \quad \text{oracle calls per node and} \quad \mathcal{O}\left(\chi\sqrt{\frac{L}{\mu}}\log\frac{1}{\epsilon}\right) \quad \text{communications}$$

*to reach $\|x^N - x^*\|^2 \leq \epsilon$.*

*Proof.* The proof may be found in Appendix B. $\qquad\square$

### 3.3 NONCONVEX CASE

For the nonconvex setup, we propose a method based on a combination of gradient tracking and PAGE gradient estimator (Li et al., 2021). The main idea of this approach consists of two parts.

**Gradient Tracking.** Gradient tracking scheme can be written as in (Nedic et al., 2017):

$$x^{k+1} = \mathbf{W}^k x^k - \eta y^k$$
$$y^{k+1} = \mathbf{W}^k y^k + \nabla F(x^{k+1}) - \nabla F(x^k)$$

Such an algorithm leads to $y_i^k$ being an approximation of the average gradient from all devices in the network at each iteration.

**PAGE.** The key meaning of PAGE is as follows. Calculating the full gradient can be expensive, but finite-sum construction allows to count the batched gradient, which is clearly lower in com-

---

**Algorithm 2** GT-PAGE

1: **Input:** Initial point $x^0 = (\mathbf{1}_m \otimes \mathbf{I}_d)x_0$, $y^0 = \nabla F(x^0)$, $v^0 = \frac{1}{m}(\mathbf{1}_m\mathbf{1}_m^\top \otimes \mathbf{I}_d)y^0$, step size $\eta$, minibatch size $b$.
2: **for** $k = 0, 1, \ldots, N-1$ **do**
3: $\quad x^{k+1} = ((\mathbf{I}_m - \mathbf{W}(k)) \otimes \mathbf{I}_d)x^k - \eta v^k$
4: $\quad S_i^k \sim \mathcal{D}_i^b(\{1, 2, \ldots, n\})$, $p_{ij} = \frac{1}{n}$
5: $\quad \left(\nabla^k\right)_i = y_i^k + \frac{1}{b}\sum\limits_{j \in S_i^k}\left(\nabla f_{ij}(x_i^{k+1}) - \nabla f_{ij}(x_i^k)\right)$
6: $\quad y^{k+1} = \begin{cases}\nabla F(x^{k+1}), & \text{with prob. } p, \\ \nabla^k, & \text{with prob. } 1-p\end{cases}$
7: $\quad v^{k+1} = ((\mathbf{I}_m - \mathbf{W}(k)) \otimes \mathbf{I}_d)v^k + y^{k+1} - y^k$
8: **end for**
9: **return** $x$ chosen uniformly from $\{x^k\}_{k=0}^{N-1}$

---

putational cost. Moreover, PAGE update does not have any loops (as, for example, in SVRG (Johnson and Zhang, 2013)) and can be computed recursively as follows:

$$\nabla^{k+1} = \frac{1}{b}\sum_{i \in S}\left[\nabla g_i(x^{k+1}) - \nabla g_i(x^k)\right] + \nabla^k,$$

where $S$ denotes a random set of indices of size $b$. Note that unlike estimator (6) for strongly convex case, PAGE estimator stores the gradient from previous iteration, not only the gradient snapshot.

**Theorem 3.4.** *Let Assumptions 2.2, 2.3 and 2.5 hold. Then, Algorithm 2 requires $N$ iterations to yield $\hat{x}^N$, which is randomly taken from $\{\bar{x}^k\}_{k=0}^{N-1}$ such that $\mathbb{E}\left[\|\nabla F(\hat{x}^N)\|^2\right] \leq \epsilon^2$, where*

$$N = \mathcal{O}\left(\frac{\chi^3 L \Delta\left(1 + \sqrt{\frac{(1-p)\tilde{L}^2}{bpL^2}}\right)}{\epsilon^2}\right),$$

*where $\Delta = F(x_0) - F^*$.*

**Corollary 3.5.** *In the setting of Theorem 3.4, let $b = \frac{\sqrt{n}\hat{L}}{L}$, $p = \frac{b}{n+b}$ and number of communications per iteration $\chi$. Then Algorithm 2 requires*

$$\mathcal{O}\left(n + \frac{\sqrt{n}\hat{L}\Delta}{\varepsilon^2}\right) \text{ oracle calls per node and } \mathcal{O}\left(\frac{\chi L\Delta}{\varepsilon^2}\right) \text{ communications}$$

*to reach accuracy $\varepsilon$, i.e. $\mathbb{E}\left[\|\nabla F(\hat{x}^N)\|^2\right] \leq \epsilon^2$.*

Proofs of Theorem 3.4 and Corollary 3.5 can be found in Appendix D.3 and Appendix D.4 respectively.

*Remark* 3.6. It should be clarified that in the case of time-static graphs the multi-step communication procedure called Chebyshev acceleration allows us to go from $\chi$ to $\sqrt{\chi}$ in the estimation on the number of communications.

## 4 LOWER BOUNDS

In this section, we present lower bounds for the strongly convex case in terms of (Hendrikx et al., 2021) and for the nonconvex case. It is important to note that the setup for the strongly convex case in which lower bounds are considered is different from the class of problems for which the algorithm was analyzed, which will be discussed in more detail later.

**Strongly Convex Case**. Lower bounds for a static network for non-stochastic problems were first presented in (Scaman et al., 2017). It has been shown that to reach an $\epsilon$-solution of the problem, the system requires $\Omega\left(\sqrt{\chi L/\mu}\log(1/\epsilon)\right)$ communication iterations and $\Omega\left(\sqrt{L/\mu}\log(1/\epsilon)\right)$ computational iterations. In (Kovalev et al., 2021a), lower bounds for a time-varying setting were presented, the differs occur in communication complexity, in particular one needs to perform $\Omega\left(\chi\sqrt{L/\mu}\log(1/\epsilon)\right)$ communication iterations to reach $\varepsilon$-solution. Regarding stochastic setup, in (Hendrikx et al., 2021), lower bounds of $\Omega(\sqrt{\chi\kappa_b}\log(1/\varepsilon))$ communication iterations and $\Omega(n + \sqrt{n\kappa_s}\log(1/\varepsilon))$ oracle calls per node were presented, where $\kappa_b = \max_i\{L_i/\mu_i\}$ is the maximum of the condition numbers of functions at nodes, and $\kappa_s = \max_i\{\hat{L}_i/\mu_i\}$ is stochastic condition number among local function at nodes. Also, an optimal dual-based method was proposed.

**Nonconvex Case**. At first, lower bounds for finite-sum nonconvex problem were presented in (Fang et al., 2018). It has been shown that for reaching $\epsilon$-accuracy ($\mathbb{E}\left[\|\nabla F(x)\|^2\right] \leq \epsilon^2$) $\Omega(\sqrt{n}\hat{L}/\epsilon^2)$ gradient estimates is required. Moreover, this lower bound was extended in (Li et al., 2021) to $\Omega(n + \sqrt{n}\hat{L}/\epsilon^2)$.

Considering a decentralized optimization problem without variance reduction, there are both estimates of lower bounds for static (e.g. (Yuan et al., 2022)) and time-varying (e.g. (Huang and Yuan, 2022)) graphs, which are equal to $\Omega(\sqrt{\chi}L\Delta/\epsilon^2)$ and $\Omega(\chi L\Delta/\epsilon^2)$ communications respectively.

The combination of decentralized nonconvex optimization with variance reduction has been studied only in the case of static graphs, e.g., in (Luo and Ye, 2022), where authors show that lower bounds are $\Omega(\sqrt{\chi}\hat{L}\Delta/\epsilon^2)$ and $\Omega(n + \sqrt{n}\hat{L}\Delta/\epsilon^2)$ in their assumptions for the number of communication rounds and local computations per node respectively.

### 4.1 FIRST-ORDER DECENTRALIZED ALGORITHMS

Following (Kovalev et al., 2021b) and (Hendrikx et al., 2021), let us formalize the concept of a decentralized optimization algorithm. The procedure will consist of two types of iterations: communicational iterations, in which nodes cannot access the oracle, but only exchange information with neighbors, and computational iterations, in which nodes do not communicate with each other, but only perform local computations in their memory. Let time be discrete, each iteration $k$ is either communicational or computational. For any vertex $i$, denote by $\mathcal{H}_i(k)$ the local memory at $k$th iteration. Then the following inclusions hold:

1. For all $i = 1, \ldots, m$, if nodes perform a local computation at step $k$, local information is updated as

$$\mathcal{H}_i(k+1) \subseteq \text{span}\left( \bigcup_{j=1}^n \{x, \nabla f_{ij}(x), \nabla f_{ij}^*(x) \mid x \in \mathcal{H}_i(k)\} \right).$$

2. For all $i = 1, \ldots, m$, if nodes perform a communicational iteration at time step $k$, local information is updated as

$$\mathcal{H}_i(k+1) \subseteq \text{span}\left( \bigcup_{j \in \mathcal{N}_i^k} \mathcal{H}_j(k) \cup \mathcal{H}_i(k) \right),$$

where $\mathcal{N}_i^k$ is neighbours of node $i$ at $k$th step.

## 4.2 Strongly Convex Case

In the strongly convex case, we formulate the lower bounds under slightly different assumptions. We let each function $F_i$ have its own smoothness and strong convexity parameters.

**Assumption 4.1.** For each $i = 1, \ldots, m$ function $F_i$ is $\mu_i$-strongly convex and $L_i$-smooth.

**Assumption 4.2.** For all $i = 1, \ldots, m$, we have $\kappa_b \geq L_i/\mu_i$ and $\kappa_s \geq \frac{1}{n} \sum_{j=1}^n L_{ij}/\mu_i$.

In this case, we allow functions on nodes to have different constants of strong convexity, preserving the constraints on condition numbers. This plays a role in lower bounds, because in the counterexample problem the strong convexity constants on the nodes can differ by a factor of $m$.

**Theorem 4.3.** *For any $\chi > 24$, for any $\kappa_b > 0$, there exists a constant $\kappa_s > 0$, a time-varying network $\{\mathcal{G}^k\}_{k=1}^\infty$ on $m$ nodes, the corresponding sequence of gossip matrices $\{\mathbf{W}(k)\}_{k=1}^\infty$ satisfying Assumption 2.5, and functions $\{f_{ij}\}$, such that the problem (1) satisfies Assumptions 2.1, 4.1, 4.2 and for any first-order decentralized algorithm holds*

$$\frac{1}{nm} \sum_{i=1}^m \sum_{j=1}^n \frac{\|x_{ij} - x^*\|^2}{\|x_{ij}^0 - x^*\|^2} \geq \max\{T_1, T_2\},$$

*where*

$$T_1 = \left(1 - \frac{2}{\sqrt{\frac{2}{3}\kappa_b + \frac{1}{3}} + 1}\right)^{2 + 16N_c/(\chi - 24)} , \quad T_2 = \left(1 - \frac{2n}{\sqrt{n}\sqrt{\frac{2}{3}\kappa_s + n/3 + n}}\right)^{4N_s/n} ,$$

*$N_c$ is the number of communication iterations, $N_s$ is the maximum number of stochastic oracle calls on any node, and $x_{ij} \in \mathcal{H}_i(k)$, $k$ is the number of the last time step.*

*Proof.* The proof may be found in Appendix C. $\square$

**Corollary 4.4.** *For any $\chi > 0$ and any $\kappa_b > 0$, there exists a decentralized problem satisfying Assumptions 2.1, 2.5, 4.1, and 4.2, such that for any first-order decentralized algorithm for each node to reach an $\epsilon$-solution of problem (1), a minimum of $N_c$ communication iterations and $N_s$ stochastic oracle calls on some node are required, where*

$$N_s = \Omega\left((n + \sqrt{n\kappa_s})\log\left(\frac{1}{\varepsilon}\right)\right), \quad N_c = \Omega\left(\chi\sqrt{\kappa_b}\log\left(\frac{1}{\varepsilon}\right)\right).$$

As we can see, the obtained lower bound has different setting than the class of problems on which the work of Algorithm 1 is analysed, the same problem is present in (Li et al., 2020) and (Kovalev et al., 2022). This difficulty appears to arise in a decentralised setup, so the question of how to make the lower bound correct, how to interpret it and what would be the optimal primal algorithm in the case of static and time-varying network remains open. The lower bounds are presented in Table Table 1.

### 4.3 Nonconvex Case

In the nonconvex case, we use the same assumptions that for Algorithm 2.

**Theorem 4.5.** *For any $L > 0$, $m \geq 3$, there exists a set $\{F_i\}_{i=1}^n$ which satisfy Assumption 2.2 and Assumption 2.3, and a sequence of matrices $\{\mathbf{W}(k)\}_{k=0}^\infty$ which satisfy Assumption 2.5, such that for any output $\hat{x}^N$ of any first-order decentralized algorithm after $N$ communications and $K$ local computations we get:*

$$\mathbb{E}\left[\|\nabla F(\hat{x}^N)\|^2\right] = \Omega\left(\tfrac{\chi L \Delta}{N}\right), \ \mathbb{E}\left[\|\nabla F(\hat{x}^N)\|^2\right] = \Omega\left(\tfrac{\sqrt{n}\Delta\hat{L}}{K}\right).$$

*Proof.* See Appendix D.5. $\square$

**Corollary 4.6.** *In the setting of Theorem 4.5, the number of communication rounds $N_c$ and local oracle calls $N_s$ required to reach $\epsilon$-accuracy ($\mathbb{E}\left[\|\nabla F(\hat{x}^N)\|^2\right] \leq \epsilon^2$) is lower bounded as*

$$N_s = \Omega\left(n + \tfrac{\sqrt{n}\Delta\hat{L}}{\epsilon^2}\right), \quad N_c = \Omega\left(\tfrac{\chi L \Delta}{\epsilon^2}\right),$$

*respectively.*

*Remark* 4.7. The lower bound for communication rounds $N_s$ is obtained the following way. From Theorem 4.5 we get $N_s = \Omega(\sqrt{n}\Delta\hat{L}/\varepsilon^2)$. Additionally, in (Li et al., 2021) it was shown that $N_s = \Omega(n)$ even for non-distributed optimization. Consequently, we have

$$N_s = \Omega\left(\max\left(n, \tfrac{\sqrt{n}\Delta\hat{L}}{\varepsilon^2}\right)\right) = \Omega\left(n + \tfrac{\sqrt{n}\Delta\hat{L}}{\varepsilon^2}\right).$$

The main idea of the proof starts from the example of "bad" nonconvex function (see (Arjevani et al., 2023)). Next, we extend the lower bound for decentralized nonconvex optimization over static graphs (see (Yuan et al., 2022)) by considering time-varying graphs and finite-sum constructions. The lower bounds for nonconvex case are also presented in Table 2.

*Remark* 4.8. Since one of the main ideas of the proof of Theorem 4.5 is the selection of a special sequence of time-varying graphs, that is why we get an estimate on the number of communications $\sim \chi$. But, as has been shown in some papers (e.g., (Yuan et al., 2022)), a lower bound on the number of communications for decentralized optimization on static graphs is $\sim \sqrt{\chi}$. Applying the same topology to our proof and taking into account Remark 3.6, we can conclude that GT-PAGE is optimal for the case of static graphs as well.

## 5 Numerical experiments

In this section, we present numerical experiments comparing the proposed methods of this paper with state-of-the-art methods for both strongly convex and nonconvex problems.

### 5.1 Setup

**Datasets.** We utilize LibSVM Chang and Lin (2011) datasets in our experiments: a9a and w8a. Each dataset in an individual experiment is randomly distributed among the agents in the communication network.
**Topology.** We consider a random geometric graph with 50 vertices as the time-varying structure of the network.
**Loss function.** As an objective functions we choose logistic loss with $l_2$-regularization and non-linear least squares loss for strongly convex and nonconvex problems respectively.
**Optimization methods.** For our experiments we implemented proposed algorithms (Algorithm 1 and Algorithm 2) with other existing approaches (see Fig. 1 and Fig. 2 for more detail).

## 5.2 Results

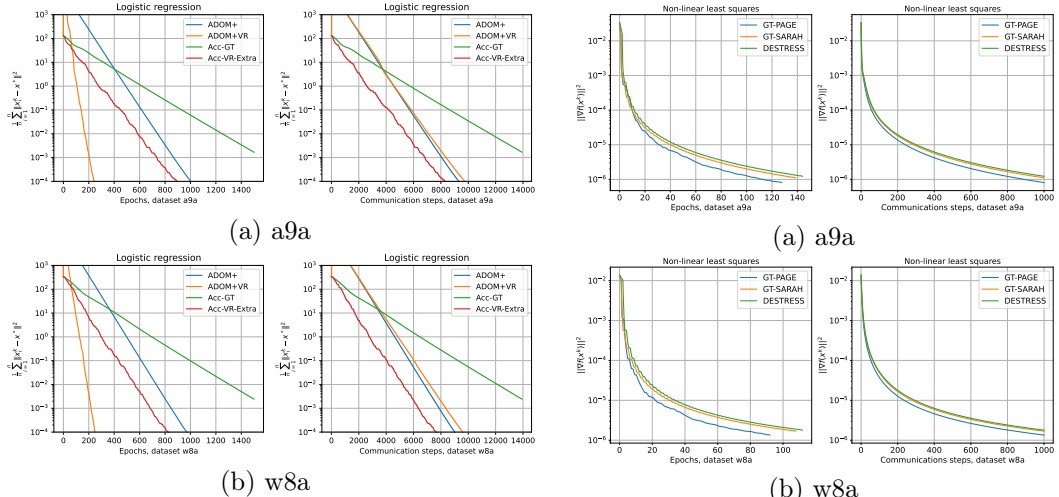

(a) a9a

(a) a9a

(b) w8a

(b) w8a

Figure 1: Comparison of communication and oracle complexities of Algorithm 1 (ADOM+VR), ADOM+, Accelerated-GT (Acc-GT) and Accelerated-VR-Extra (Acc-VR-Extra) on logistic regression problem on LibSVM datasets.

Figure 2: Comparison of communication and oracle complexities of Algorithm 2 (GT-PAGE), GT-SARAH and DESTRESS on non-linear least squares problem on LibSVM datasets.

Experimental outcomes are shown in Fig. 1 and Fig. 2. Regarding the logistic regression problem, ADOM+VR outperforms other methods with respect to the number of epochs, i.e. the number of oracle calls. However, there is no gain in communication complexity compared to state-of-the-art approaches. At the same time, for the non-linear least squares problem, GT-PAGE behaves better with respect to other methods, but it does not demonstrate a strong superiority.

## 6 Conclusion

This paper establishes lower bounds for stochastic decentralized optimization in both non-convex and strongly convex scenarios. For the nonconvex case, we derived a lower bound of $\Omega\left(n + \sqrt{n}\hat{L}\Delta/\varepsilon^2\right)$ for stochastic oracle calls at a certain node, and $\Omega\left(\chi L\Delta/\varepsilon^2\right)$ for communication rounds, while also proposing the optimal GT-PAGE algorithm. In the strongly convex case, the lower bound of $\Omega\left((n + \sqrt{n\kappa_s})\log(1/\varepsilon)\right)$ for stochastic oracle calls and $\Omega\left(\chi\sqrt{\kappa_b}\log(1/\varepsilon)\right)$ for communication iterations was introduced. The paper also proposes the ADOM+VR algorithm, which optimal in terms of communication iterations. Despite it, the questions of whether existing decentralised VR algorithms are optimal and whether there is a similar lower bound for a narrower class of problems were highlighted. These questions remain open in both time-varying and static scenarios, presenting a way for future research.

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

## A    Appendix / supplemental material

We now establish the convergence rate of Algorithm 1. This proof is for the most part a modified analysis of the ADOM+ algorithm with the addition of techniques corresponding to variance reduction setting. The parts not affected by the change were kept for the sake of completeness.

## B    Proof of Theorem 3.2

By $\mathrm{D}_F(x, y)$ we denote Bregman distance $\mathrm{D}_F(x, y) := F(x) - F(y) - \langle \nabla F(y), x - y \rangle$.

By $\mathrm{G}_F(x, y)$ we denote $\mathrm{G}_F(x, y) := \mathrm{D}_F(x, y) - \frac{\nu}{2} \|x - y\|^2$.

**Lemma B.1.**

$$\mathbb{E}_{S^k} \left[ \|\nabla^k - \nabla F(x_g^k)\|^2 \right] \le \frac{2\overline{L}}{b} \left( \mathrm{G}_F(\omega^k, x^*) - \mathrm{G}_F(x_g^k, x^*) \right) \tag{7}$$
$$- \frac{2\overline{L}}{b} \langle \nabla F(x_g^k) - \nabla F(x^*) - \nu x_g^k + \nu x^*, \omega^k - x_g^k \rangle.$$

*Proof.* Firstly note, that if $g^k = \nabla f_i(x_g^k) - \nabla f_i(\omega^k) + \nabla f_i(\omega^k)$, then

$$\mathbb{E}_i \left[ \|g^k - \nabla f(x_g^k)\|^2 \right] = \mathbb{E}_i \left[ \|\nabla f_i(x_g^k) - \nabla f_i(w^k) - \mathbb{E}_i \left[ \nabla f_i(x_g^k) - \nabla f_i(w^k) \right] \|^2 \right]$$
$$\le \mathbb{E}_i \left[ \|\nabla f_i(x^k) - \nabla f_i(w^k)\|^2 \right] \tag{8}$$
$$\le 2\overline{L} \left( f(w^k) - f(x^k) - \langle \nabla f(x^k), w^k - x^k \rangle \right).$$

Let us describe the main term

$$\mathbb{E}_{S_i^k} \left[ \|(\nabla^k)_i - \nabla F_i(x_{g,i}^k)\|^2 \right] = \mathbb{E}_{S_i^k} \left[ \left\| \frac{1}{b} \sum_{j \in S_i^k} \frac{1}{np_{ij}} \left[ \nabla f_{ij}(x_{g,i}^k) - \nabla f_{ij}(\omega_i^k) \right] + \nabla F_i(\omega_i^k) - \nabla F_i(x_{g,i}^k) \right\|^2 \right]$$

$$\overset{(1)}{=} \frac{1}{b} \mathbb{E}_j \left[ \left\| \frac{1}{np_{ij}} \left[ \nabla f_{ij}(x_{g,i}^k) - \nabla f_{ij}(\omega_i^k) \right] + \nabla F_i(\omega_i^k) - \nabla F_i(x_{g,i}^k) \right\|^2 \right]$$

$$= \frac{1}{b} \mathbb{E}_j \left[ \left\| \frac{1}{np_{ij}} \left[ \left( \nabla f_{ij}(x_{g,i}^k) - \nabla f_{ij}(x^*) \right) - \left( \nabla f_{ij}(\omega_i^k) - \nabla f_{ij}(x^*) \right) \right] + \nabla F_i(\omega_i^k) - \nabla F_i(x_{g,i}^k) \right\|^2 \right]$$

$$\overset{(2)}{\le} \frac{1}{b} \mathbb{E}_j \left[ \left\| \frac{1}{np_{ij}} \left[ \left( \nabla f_{ij}(x_{g,i}^k) - \nabla f_{ij}(x^*) - \nu x_{g,i}^k + \nu x^* \right) - \left( \nabla f_{ij}(\omega_i^k) - \nabla f_{ij}(x^*) - \nu \omega_i + \nu x^* \right) \right] \right\|^2 \right]$$

$$\overset{(3)}{\le} \sum_{j=1}^n \frac{p_{ij}}{b} \frac{2L_{ij}}{n^2 p_{ij}^2} \mathrm{G}_{f_{ij}}(x_{g,i}^k, x^*) \overset{(4)}{=} \frac{2\overline{L}_i}{b} \left( \mathrm{G}_{F_i}(\omega_i^k, x^*) - \mathrm{G}_{F_i}(x_{g,i}^k, x^*) - \langle \nabla \mathrm{G}_{F_i}(x_{g,i}^k, x^*), \omega_i^k - x_{g,i}^k \rangle \right),$$

where

 (1)  is due to independency of $(\xi_i^1, \xi_i^2, \ldots, \xi_i^b)$,

 (2)  follows from the inequality $\mathbb{E} \left[ \|\xi\|^2 \right] \le \mathbb{E} \left[ \|\xi + c\|^2 \right]$ if $\mathbb{E}[\xi] = 0$ and $c$ is constant,

 (3)  follows from (8) inequality,

 (4)  follows from $p_{ij} = L_{ij}/(n\overline{L}_i)$ definition.

The required inequality is the simple consequence of the previous statement.    □

Further we will assume that the basis of the expectation is clear from the context.

**Lemma B.2.** *Let $\tau_2$ be defined as follows:*

$$\tau_2 = \min\left\{\frac{1}{2}, \max\left\{1, \frac{\sqrt{n}}{b}\right\}\sqrt{\frac{\mu}{L}}\right\}. \tag{9}$$

*Let $\tau_1$ be defined as follows:*

$$\tau_1 = (1 - \tau_0)(1/\tau_2 + 1/2)^{-1}. \tag{10}$$

*Let $\tau_0$ be defined as follows:*

$$\tau_0 = \frac{\overline{L}}{2Lb}. \tag{11}$$

*Let $\eta$ be defined as follows:*

$$\eta = \left(L\left(\tau_2 + \frac{2\tau_1}{1 - \tau_1}\right)\right)^{-1}. \tag{12}$$

*Let $\alpha$ be defined as follows:*

$$\alpha = \mu/2. \tag{13}$$

*Let $\nu$ be defined as follows:*

$$\nu = \mu/2. \tag{14}$$

*Let $\Psi_x^k$ be defined as follows:*

$$\Psi_x^k = \left(\frac{1}{\eta} + \alpha\right)\|x^k - x^*\|^2 + \frac{2}{\tau_2}\left(\mathrm{D}_f(x_f^k, x^*) - \frac{\nu}{2}\|x_f^k - x^*\|^2\right) \tag{15}$$

*Then the following inequality holds:*

$$\Psi_x^{k+1} \le \left(1 - \frac{1}{20}\min\left\{\sqrt{\frac{\mu}{L}}, b\sqrt{\frac{\mu}{nL}}\right\}\right)\Psi_x^k + 2\mathbb{E}\left[\langle y^{k+1} - y^*, x^{k+1} - x^*\rangle\right]$$

$$+ \frac{\overline{L}}{Lb}\left(\frac{1}{\tau_1} - 1\right)\left(\mathrm{G}_F(\omega^k, x^*) - \mathrm{G}_F(x_g^k, x^*)\right) - \mathrm{G}_F(x_g^k, x^*) - \frac{1}{2}\mathrm{G}_F(x_f^k, x^*) \tag{16}$$

$$+ \frac{\overline{L}}{Lb}\langle\nabla F(x_g^k) - \nabla F(x^*) - \nu x_g^k + \nu x^*, \omega^k - x_g^k\rangle.$$

*Proof.*

$$\frac{1}{\eta}\|x^{k+1} - x^*\|^2 = \frac{1}{\eta}\|x^k - x^*\|^2 + \frac{2}{\eta}\langle x^{k+1} - x^k, x^{k+1} - x^*\rangle - \frac{1}{\eta}\|x^{k+1} - x^k\|^2.$$

Using Line 7 of Algorithm 1 we get

$$\frac{1}{\eta}\|x^{k+1} - x^*\|^2 = \frac{1}{\eta}\|x^k - x^*\|^2 + 2\alpha\langle x_g^k - x^{k+1}, x^{k+1} - x^*\rangle$$

$$- 2\langle\nabla^k - \nu x_g^k - y^{k+1}, x^{k+1} - x^*\rangle - \frac{1}{\eta}\|x^{k+1} - x^k\|^2$$

$$= \frac{1}{\eta}\|x^k - x^*\|^2 + 2\alpha\langle x_g^k - x^* - x^{k+1} + x^*, x^{k+1} - x^*\rangle$$

$$- 2\langle\nabla^k - \nu\hat{x}_g^k - y^{k+1}, x^{k+1} - x^*\rangle - \frac{1}{\eta}\|x^{k+1} - x^k\|^2$$

$$\le \frac{1}{\eta}\|x^k - x^*\|^2 - \alpha\|x^{k+1} - x^*\|^2 + \alpha\|x_g^k - x^*\|^2$$

$$- 2\langle\nabla^k - \nu x_g^k - y^{k+1}, x^{k+1} - x^*\rangle - \frac{1}{\eta}\|x^{k+1} - x^k\|^2.$$

Using optimality condition (3) we get

$$\frac{1}{\eta}\|x^{k+1} - x^*\|^2 \le \frac{1}{\eta}\|x^k - x^*\|^2 - \alpha\|x^{k+1} - x^*\|^2 + \alpha\|x_g^k - x^*\|^2 - \frac{1}{\eta}\|x^{k+1} - x^k\|^2$$

$$- 2\langle\nabla F(x_g^k) - \nabla F(x^*), x^{k+1} - x^*\rangle + 2\nu\langle x_g^k - x^*, x^{k+1} - x^*\rangle$$

$$+ 2\langle y^{k+1} - y^*, x^{k+1} - x^* \rangle - 2\langle \nabla^k - \nabla F(x_g^k), x^{k+1} - x^* \rangle.$$

Using Line 8 of Algorithm 1 we get

$$\frac{1}{\eta}\|x^{k+1} - x^*\|^2 \le \frac{1}{\eta}\|x^k - x^*\|^2 - \alpha\|x^{k+1} - x^*\|^2 + \alpha\|x_g^k - x^*\|^2 - \frac{1}{\eta\tau_2^2}\|x_f^{k+1} - x_g^k\|^2$$

$$- 2\langle \nabla F(x_g^k) - \nabla F(x^*), x^k - x^* \rangle + 2\nu\langle x_g^k - x^*, x^k - x^* \rangle$$

$$+ 2\langle y^{k+1} - y^*, x^{k+1} - x^* \rangle - \frac{2}{\tau_2}\langle \nabla F(x_g^k) - \nabla F(x^*), x_f^{k+1} - x_g^k \rangle$$

$$+ \frac{2\nu}{\tau_2}\langle x_g^k - x^*, x_f^{k+1} - x_g^k \rangle - 2\langle \nabla^k - \nabla F(x_g^k) - \nu(\hat{x}_g^k - x_g^k), x^{k+1} - x^* \rangle$$

$$= \frac{1}{\eta}\|x^k - x^*\|^2 - \alpha\|x^{k+1} - x^*\|^2 + \alpha\|x_g^k - x^*\|^2 - \frac{1}{\eta\tau_2^2}\|x_f^{k+1} - x_g^k\|^2$$

$$- 2\langle \nabla F(x_g^k) - \nabla F(x^*), x^k - x^* \rangle + 2\nu\langle x_g^k - x^*, x^k - x^* \rangle$$

$$+ 2\langle y^{k+1} - y^*, x^{k+1} - x^* \rangle - \frac{2}{\tau_2}\langle \nabla F(x_g^k) - \nabla F(x^*), x_f^{k+1} - x_g^k \rangle$$

$$+ \frac{\nu}{\tau_2}\left(\|x_f^{k+1} - x^*\|^2 - \|x_g^k - x^*\|^2 - \|x_f^{k+1} - x_g^k\|^2\right)$$

$$- 2\langle \nabla^k - \nabla F(x_g^k), x^{k+1} - x^* \rangle.$$

Using the $L$-smoothness property of $\mathrm{D}_F(x, x^*)$ with respect to $x$, which is derived from the $L$-smoothness of $F(x)$, we obtain

$$\frac{1}{\eta}\|x^{k+1} - x^*\|^2 \le \frac{1}{\eta}\|x^k - x^*\|^2 - \alpha\|x^{k+1} - x^*\|^2 + \alpha\|x_g^k - x^*\|^2 - \frac{1}{\eta\tau_2^2}\|x_f^{k+1} - x_g^k\|^2$$

$$- 2\langle \nabla F(x_g^k) - \nabla F(x^*), x^k - x^* \rangle + 2\nu\langle x_g^k - x^*, x^k - x^* \rangle$$

$$+ 2\langle y^{k+1} - y^*, x^{k+1} - x^* \rangle - \frac{2}{\tau_2}\langle \nabla F(x_g^k) - \nabla F(x^*), x_f^{k+1} - x_g^k \rangle$$

$$+ \frac{\nu}{\tau_2}\left(\|x_f^{k+1} - x^*\|^2 - \|x_g^k - x^*\|^2 - \|x_f^{k+1} - x_g^k\|^2\right) - 2\langle \nabla^k - \nabla F(x_g^k), x^{k+1} - x^* \rangle$$

$$\le \frac{1}{\eta}\|x^k - x^*\|^2 - \alpha\|x^{k+1} - x^*\|^2 + \alpha\|x_g^k - x^*\|^2 - \frac{1}{\eta\tau_2^2}\|x_f^{k+1} - x_g^k\|^2$$

$$- 2\langle \nabla F(x_g^k) - \nabla F(x^*), x^k - x^* \rangle + 2\nu\langle x_g^k - x^*, x^k - x^* \rangle + 2\langle y^{k+1} - y^*, x^{k+1} - x^* \rangle$$

$$- \frac{2}{\tau_2}\left(\mathrm{D}_f(x_f^{k+1}, x^*) - \mathrm{D}_f(x_g^k, x^*) - \frac{L}{2}\|x_f^{k+1} - x_g^k\|^2\right)$$

$$+ \frac{\nu}{\tau_2}\left(\|x_f^{k+1} - x^*\|^2 - \|x_g^k - x^*\|^2 - \|x_f^{k+1} - x_g^k\|^2\right) - 2\langle \nabla^k - \nabla F(x_g^k), x^{k+1} - x^* \rangle$$

$$= \frac{1}{\eta}\|x^k - x^*\|^2 - \alpha\|x^{k+1} - x^*\|^2 + \alpha\|x_g^k - x^*\|^2 + \left(\frac{L - \nu}{\tau_2} - \frac{1}{\eta\tau_2^2}\right)\|x_f^{k+1} - x_g^k\|^2$$

$$- 2\langle \nabla F(x_g^k) - \nabla F(x^*), x^k - x^* \rangle + 2\nu\langle x_g^k - x^*, x^k - x^* \rangle + 2\langle y^{k+1} - y^*, x^{k+1} - x^* \rangle$$

$$- \frac{2}{\tau_2}\left(\mathrm{D}_f(x_f^{k+1}, x^*) - \mathrm{D}_f(x_g^k, x^*)\right) + \frac{\nu}{\tau_2}\left(\|x_f^{k+1} - x^*\|^2 - \|x_g^k - x^*\|^2\right)$$

$$- 2\langle \nabla^k - \nabla F(x_g^k), x^{k+1} - x^* \rangle$$

Using Line 4 of Algorithm 1 we get

$$\frac{1}{\eta}\|x^{k+1} - x^*\|^2 \le \frac{1}{\eta}\|x^k - x^*\|^2 - \alpha\|x^{k+1} - x^*\|^2 + \alpha\|x_g^k - x^*\|^2$$

$$+ \left(\frac{L - \nu}{\tau_2} - \frac{1}{\eta\tau_2^2}\right)\|x_f^{k+1} - x_g^k\|^2 - 2\langle \nabla F(x_g^k) - \nabla F(x^*), x_g^k - x^* \rangle + 2\nu\|x_g^k - x^*\|^2$$

$$+ \frac{2(1 - \tau_1 - \tau_0)}{\tau_1}\langle \nabla F(x_g^k) - \nabla F(x^*), x_f^k - x_g^k \rangle + \frac{2\tau_0}{\tau_1}\langle \nabla F(x_g^k) - \nabla F(x^*), \omega^k - x_g^k \rangle$$

$$+ \frac{2\nu(1 - \tau_1 - \tau_0)}{\tau_1}\langle x_g^k - x_f^k, x_g^k - x^* \rangle + \frac{2\nu\tau_0}{\tau_1}\langle x_g^k - \omega^k, x_g^k - x^* \rangle$$

$$+ 2\langle y^{k+1} - y^*, x^{k+1} - x^* \rangle - \frac{2}{\tau_2}\left(D_f(x_f^{k+1}, x^*) - D_f(x_g^k, x^*)\right)$$

$$+ \frac{\nu}{\tau_2}\left(\|x_f^{k+1} - x^*\|^2 - \|x_g^k - x^*\|^2\right) - 2\langle \nabla^k - \nabla F(x_g^k), x^{k+1} - x^* \rangle$$

$$= \frac{1}{\eta}\|x^k - x^*\|^2 - \alpha\|x^{k+1} - x^*\|^2 + \alpha\|x_g^k - x^*\|^2 + \left(\frac{L - \nu}{\tau_2} - \frac{1}{\eta\tau_2^2}\right)\|x_f^{k+1} - x_g^k\|^2$$

$$- 2\langle \nabla F(x_g^k) - \nabla F(x^*), x_g^k - x^* \rangle + 2\nu\|x_g^k - x^*\|^2$$

$$+ \frac{2(1 - \tau_1 - \tau_0)}{\tau_1}\langle \nabla F(x_g^k) - \nabla F(x^*), x_f^k - x_g^k \rangle + \frac{2\tau_0}{\tau_1}\langle \nabla F(x_g^k) - \nabla F(x^*), \omega^k - x_g^k \rangle$$

$$+ \frac{\nu(1 - \tau_1 - \tau_0)}{\tau_1}\left(\|x_g^k - x_f^k\|^2 + \|x_g^k - x^*\|^2 - \|x_f^k - x^*\|^2\right) + \frac{2\nu\tau_0}{\tau_1}\langle x_g^k - \omega^k, x_g^k - x^* \rangle$$

$$+ 2\langle y^{k+1} - y^*, x^{k+1} - x^* \rangle - \frac{2}{\tau_2}\left(D_f(x_f^{k+1}, x^*) - D_f(x_g^k, x^*)\right)$$

$$+ \frac{\nu}{\tau_2}\left(\|x_f^{k+1} - x^*\|^2 - \|x_g^k - x^*\|^2\right) - 2\langle \nabla^k - \nabla F(x_g^k), x^{k+1} - x^* \rangle.$$

By applying $\mu$-strong convexity of $D_F(x, x^*)$ in $x$, following from $\mu$-strong convexity of $F(x)$, we obtain

$$\frac{1}{\eta}\|x^{k+1} - x^*\|^2 \le \frac{1}{\eta}\|x^k - x^*\|^2 - \alpha\|x^{k+1} - x^*\|^2 + \alpha\|x_g^k - x^*\|^2$$

$$+ \left(\frac{L - \nu}{\tau_2} - \frac{1}{\eta\tau_2^2}\right)\|x_f^{k+1} - x_g^k\|^2 - 2D_F(x_g^k, x^*) - \mu\|x_g^k - x^*\|^2 + 2\nu\|x_g^k - x^*\|^2$$

$$+ \frac{2(1 - \tau_1 - \tau_0)}{\tau_1}\left(D_F(x_f^k, x^*) - D_F(x_g^k, x^*) - \frac{\mu}{2}\|x_f^k - x_g^k\|^2\right)$$

$$+ \frac{2\tau_0}{\tau_1}\langle \nabla F(x_g^k) - \nabla F(x^*), \omega^k - x_g^k \rangle + \frac{2\nu\tau_0}{\tau_1}\langle x_g^k - \omega^k, x_g^k - x^* \rangle$$

$$+ \frac{\nu(1 - \tau_1 - \tau_0)}{\tau_1}\left(\|x_g^k - x_f^k\|^2 + \|x_g^k - x^*\|^2 - \|x_f^k - x^*\|^2\right)$$

$$+ 2\langle y^{k+1} - y^*, x^{k+1} - x^* \rangle - \frac{2}{\tau_2}\left(D_f(x_f^{k+1}, x^*) - D_f(x_g^k, x^*)\right)$$

$$+ \frac{\nu}{\tau_2}\left(\|x_f^{k+1} - x^*\|^2 - \|x_g^k - x^*\|^2\right) - 2\langle \nabla^k - \nabla F(x_g^k), x^{k+1} - x^* \rangle.$$

$$= \frac{1}{\eta}\|x^k - x^*\|^2 - \alpha\|x^{k+1} - x^*\|^2 + \frac{2(1 - \tau_1 - \tau_0)}{\tau_1}\left(D_F(x_f^k, x^*) - \frac{\nu}{2}\|x_f^k - x^*\|^2\right)$$

$$- \frac{2}{\tau_2}\left(D_f(x_f^{k+1}, x^*) - \frac{\nu}{2}\|x_f^{k+1} - x^*\|^2\right) + 2\langle y^{k+1} - y^*, x^{k+1} - x^* \rangle$$

$$+ 2\left(\frac{1}{\tau_2} - \frac{1 - \tau_0}{\tau_1}\right)D_F(x_g^k, x^*) + \left(\alpha - \mu + \nu + \frac{(1 - \tau_0)\nu}{\tau_1} - \frac{\nu}{\tau_2}\right)\|x_g^k - x^*\|^2$$

$$+ \left(\frac{L - \nu}{\tau_2} - \frac{1}{\eta\tau_2^2}\right)\|x_f^{k+1} - x_g^k\|^2 + \frac{(1 - \tau_1 - \tau_0)(\nu - \mu)}{\tau_1}\|x_f^k - x_g^k\|^2$$

$$+ \frac{2\tau_0}{\tau_1}\langle \nabla F(x_g^k) - \nabla F(x^*), \omega^k - x_g^k \rangle + \frac{2\nu\tau_0}{\tau_1}\langle x_g^k - \omega^k, x_g^k - x^* \rangle$$

$$- 2\langle \nabla^k - \nabla F(x_g^k), x^{k+1} - x^* \rangle.$$

Utilizing $\eta$ as defined in (12), $\tau_1$ as defined in (10), and considering that $\nu < \mu$, we derive

$$\frac{1}{\eta}\|x^{k+1} - x^*\|^2 \le \frac{1}{\eta}\|x^k - x^*\|^2 - \alpha\|x^{k+1} - x^*\|^2 + \frac{2(1 - \tau_2/2)}{\tau_2}G_F(x_f^k, x^*)$$

$$- \frac{2}{\tau_2}G_F(x_f^{k+1}, x^*) + 2\langle y^{k+1} - y^*, x^{k+1} - x^* \rangle$$

$$- D_F(x_g^k, x^*) + \left(\alpha - \mu + \frac{3\nu}{2}\right)\|x_g^k - x^*\|^2 - \frac{2L\tau_1}{\tau_2^2(1 - \tau_1)}\|x_f^{k+1} - x_g^k\|^2$$

$$+ \frac{2\tau_0}{\tau_1} \langle \left( \nabla F(x_g^k) - \nu x_g^k \right) - \left( \nabla F(x^*) - \nu x^* \right), \omega^k - x_g^k \rangle$$

$$- 2 \langle \nabla^k - \nabla F(x_g^k), x^{k+1} - x^* \rangle.$$

Using $\alpha$ defined by (13) and $\nu$ defined by (14) we get

$$\frac{1}{\eta} \|x^{k+1} - x^*\|^2 \le \frac{1}{\eta} \|x^k - x^*\|^2 - \alpha \|x^{k+1} - x^*\|^2 + \frac{2(1 - \tau_2/2)}{\tau_2} G_F(x_f^k, x^*)$$

$$- \frac{2}{\tau_2} G_F(x_f^{k+1}, x^*) + 2 \langle y^{k+1} - y^*, x^{k+1} - x^* \rangle$$

$$- \left( D_F(x_g^k, x^*) - \frac{\nu}{2} \|x_g^k - x^*\|^2 \right) - \frac{2L\tau_1}{\tau_2^2 (1 - \tau_1)} \|x_f^{k+1} - x_g^k\|^2$$

$$+ \frac{2\tau_0}{\tau_1} \langle \left( \nabla F(x_g^k) - \nu x_g^k \right) - \left( \nabla F(x^*) - \nu x^* \right), \omega^k - x_g^k \rangle$$

$$- 2 \langle \nabla^k - \nabla F(x_g^k), x^{k+1} - x^* \rangle.$$

Taking the expectation over $i$ at the $k$th step, using that $x^k - x^*$ is independent of $i$ and that $\mathbb{E} \left[ \nabla^k - \nabla F(x_g^k) \right] = 0$ we get

$$\frac{1}{\eta} \mathbb{E} \left[ \|x^{k+1} - x^*\|^2 \right] \le \frac{1}{\eta} \|x^k - x^*\|^2 - \alpha \mathbb{E} \left[ \|x^{k+1} - x^*\|^2 \right] + \frac{2(1 - \tau_2/2)}{\tau_2} G_F(x_f^k, x^*)$$

$$- \frac{2}{\tau_2} \mathbb{E} \left[ G_F(x_f^{k+1}, x^*) \right] + 2 \mathbb{E} \left[ \langle y^{k+1} - y^*, x^{k+1} - x^* \rangle \right]$$

$$- G_F(x_g^k, x^*) - \frac{2L\tau_1}{\tau_2^2 (1 - \tau_1)} \mathbb{E} \left[ \|x_f^{k+1} - x_g^k\|^2 \right]$$

$$+ \frac{2\tau_0}{\tau_1} \langle \left( \nabla F(x_g^k) - \nu x_g^k \right) - \left( \nabla F(x^*) - \nu x^* \right), \omega^k - x_g^k \rangle$$

$$- 2 \mathbb{E} \left[ \langle \nabla^k - \nabla F(x_g^k), x^{k+1} - x^k \rangle \right].$$

Using Line 8 of Algorithm 1 and the Cauchy–Schwarz inequality for $\langle \nabla^k - \nabla F(x_g^k), x_f^{k+1} - x_g^k \rangle$ we get

$$\frac{1}{\eta} \mathbb{E} \left[ \|x^{k+1} - x^*\|^2 \right] \le \frac{1}{\eta} \|x^k - x^*\|^2 - \alpha \mathbb{E} \left[ \|x^{k+1} - x^*\|^2 \right] + \frac{2(1 - \tau_2/2)}{\tau_2} G_F(x_f^k, x^*)$$

$$- \frac{2}{\tau_2} \mathbb{E} \left[ G_F(x_f^{k+1}, x^*) \right] + 2 \mathbb{E} \left[ \langle y^{k+1} - y^*, x^{k+1} - x^* \rangle \right]$$

$$- G_F(x_g^k, x^*) + \frac{1 - \tau_1}{2L\tau_1} \mathbb{E} \left[ \|\nabla^k - \nabla F(x_g^k)\|^2 \right]$$

$$+ \frac{2\tau_0}{\tau_1} \langle \left( \nabla F(x_g^k) - \nu x_g^k \right) - \left( \nabla F(x^*) - \nu x^* \right), \omega^k - x_g^k \rangle.$$

Using lemma B.1 and $\tau_0$ definition (11) we get

$$\frac{1}{\eta} \mathbb{E} \left[ \|x^{k+1} - x^*\|^2 \right] \le \frac{1}{\eta} \|x^k - x^*\|^2 - \alpha \mathbb{E} \left[ \|x^{k+1} - x^*\|^2 \right] + \frac{2(1 - \tau_2/2)}{\tau_2} G_F(x_f^k, x^*)$$

$$- \frac{2}{\tau_2} \mathbb{E} \left[ G_F(x_f^{k+1}, x^*) \right] + 2 \mathbb{E} \left[ \langle y^{k+1} - y^*, x^{k+1} - x^* \rangle \right] - G_F(x_g^k, x^*)$$

$$+ \frac{\overline{L}}{Lb} \left( \frac{1}{\tau_1} - 1 \right) \left( G_F(\omega^k, x^*) - G_F(x_g^k, x^*) - \langle \nabla F(x_g^k) - \nabla F(x^*) - \nu x_g^k + \nu x^*, \omega^k - x_g^k \rangle \right)$$

$$+ \frac{2\tau_0}{\tau_1} \langle \left( \nabla F(x_g^k) - \nu x_g^k \right) - \left( \nabla F(x^*) - \nu x^* \right), \omega^k - x_g^k \rangle$$

$$= \frac{1}{\eta} \|x^k - x^*\|^2 - \alpha \mathbb{E} \left[ \|x^{k+1} - x^*\|^2 \right] + \frac{2(1 - \tau_2/2)}{\tau_2} G_F(x_f^k, x^*)$$

$$- \frac{2}{\tau_2} \mathbb{E} \left[ G_F(x_f^{k+1}, x^*) \right] + 2 \mathbb{E} \left[ \langle y^{k+1} - y^*, x^{k+1} - x^* \rangle \right]$$

$$+ \frac{\overline{L}}{Lb} \left( \frac{1}{\tau_1} - 1 \right) \left( \mathrm{G}_F(\omega^k, x^*) - \mathrm{G}_F(x_g^k, x^*) \right) - \mathrm{G}_F(x_g^k, x^*)$$

$$+ \frac{\overline{L}}{Lb} \langle \nabla F(x_g^k) - \nabla F(x^*) - \nu x_g^k + \nu x^*, \omega^k - x_g^k \rangle.$$

After rearranging and using $\Psi_x^k$ definition (15) we get

$$\mathbb{E}\left[\Psi_x^{k+1}\right] \leq \max\left\{1 - \tau_2/4, 1/(1 + \eta\alpha)\right\} \Psi_x^k + 2\mathbb{E}\left[\langle y^{k+1} - y^*, x^{k+1} - x^* \rangle\right]$$

$$+ \frac{\overline{L}}{Lb} \left( \frac{1}{\tau_1} - 1 \right) \left( \mathrm{G}_F(\omega^k, x^*) - \mathrm{G}_F(x_g^k, x^*) \right) - \mathrm{G}_F(x_g^k, x^*) - \frac{1}{2}\mathrm{G}_F(x_f^k, x^*)$$

$$+ \frac{\overline{L}}{Lb} \langle \nabla F(x_g^k) - \nabla F(x^*) - \nu x_g^k + \nu x^*, \omega^k - x_g^k \rangle$$

$$\leq \left( 1 - \frac{1}{20} \min\left\{ \sqrt{\frac{\mu}{L}}, b\sqrt{\frac{\mu}{nL}} \right\} \right) \Psi_x^k + 2\mathbb{E}\left[\langle y^{k+1} - y^*, x^{k+1} - x^* \rangle\right]$$

$$+ \frac{\overline{L}}{Lb} \left( \frac{1}{\tau_1} - 1 \right) \left( \mathrm{G}_F(\omega^k, x^*) - \mathrm{G}_F(x_g^k, x^*) \right) - \mathrm{G}_F(x_g^k, x^*) - \frac{1}{2}\mathrm{G}_F(x_f^k, x^*)$$

$$+ \frac{\overline{L}}{Lb} \langle \nabla F(x_g^k) - \nabla F(x^*) - \nu x_g^k + \nu x^*, \omega^k - x_g^k \rangle.$$

The last inequality follows from $\eta$, $\alpha$, $\tau_0$, $\tau_1$, $\tau_2$ definitions (12), (13), (11), (10) and (9). Estimating the second term:

$$\frac{1}{1 + \eta\alpha} \leq 1 - \frac{\eta\alpha}{2} \leq 1 - \frac{\mu}{4}\left( L\left(\tau_2 + \frac{2\tau_1}{1-\tau_1}\right) \right)^{-1} \leq 1 - \frac{\mu}{4}\left( L\left(\tau_2 + \frac{2\tau_2}{1-\tau_2}\right) \right)^{-1}$$

$$\leq 1 - \frac{\mu}{4}\left( L\left(\tau_2 + 4\tau_2\right) \right)^{-1} = 1 - \frac{\mu}{20L\tau_2} \leq 1 - \frac{1}{20\max\left\{1, \frac{\sqrt{n}}{b}\right\}}\sqrt{\frac{\mu}{L}}$$

$$\leq 1 - \frac{1}{20}\min\left\{ \sqrt{\frac{\mu}{L}}, b\sqrt{\frac{\mu}{nL}} \right\}.$$

Estimating the first term:

$$1 - \tau_2/4 \leq 1 - \min\left\{ \frac{1}{8}, \frac{1}{4}\sqrt{\frac{\mu}{L}} \right\}.$$

$\square$

**Lemma B.3.** *The following inequality holds:*

$$-\|y^{k+1} - y^*\|^2 \leq \frac{(1-\sigma_1)}{\sigma_1}\|y_f^k - y^*\|^2 - \frac{1}{\sigma_2}\|y_f^{k+1} - y^*\|^2$$
$$- \left( \frac{1}{\sigma_1} - \frac{1}{\sigma_2} \right)\|y_g^k - y^*\|^2 + (\sigma_2 - \sigma_1)\|y^{k+1} - y^k\|^2. \tag{17}$$

*Proof.* Lines 10 and 12 of Algorithm 1 imply

$$y_f^{k+1} = y_g^k + \sigma_2(y^{k+1} - y^k)$$

$$= y_g^k + \sigma_2 y^{k+1} - \frac{\sigma_2}{\sigma_1}\left( y_g^k - (1-\sigma_1)y_f^k \right)$$

$$= \left( 1 - \frac{\sigma_2}{\sigma_1} \right)y_g^k + \sigma_2 y^{k+1} + \left( \frac{\sigma_2}{\sigma_1} - \sigma_2 \right)y_f^k.$$

After subtracting $y^*$ and rearranging we get

$$(y_f^{k+1} - y^*) + \left( \frac{\sigma_2}{\sigma_1} - 1 \right)(y_g^k - y^*) = \sigma_2(y^{k+1} - y^*) + \left( \frac{\sigma_2}{\sigma_1} - \sigma_2 \right)(y_f^k - y^*).$$

Multiplying both sides by $\frac{\sigma_1}{\sigma_2}$ gives

$$\frac{\sigma_1}{\sigma_2}(y_f^{k+1} - y^*) + \left(1 - \frac{\sigma_1}{\sigma_2}\right)(y_g^k - y^*) = \sigma_1(y^{k+1} - y^*) + (1 - \sigma_1)(y_f^k - y^*).$$

Squaring both sides gives

$$\frac{\sigma_1}{\sigma_2}\|y_f^{k+1} - y^*\|^2 + \left(1 - \frac{\sigma_1}{\sigma_2}\right)\|y_g^k - y^*\|^2 - \frac{\sigma_1}{\sigma_2}\left(1 - \frac{\sigma_1}{\sigma_2}\right)\|y_f^{k+1} - y_g^k\|^2$$
$$\leq \sigma_1\|y^{k+1} - y^*\|^2 + (1 - \sigma_1)\|y_f^k - y^*\|^2.$$

Rearranging gives

$$-\|y^{k+1} - y^*\|^2 \leq -\left(\frac{1}{\sigma_1} - \frac{1}{\sigma_2}\right)\|y_g^k - y^*\|^2 + \frac{(1 - \sigma_1)}{\sigma_1}\|y_f^k - y^*\|^2$$
$$- \frac{1}{\sigma_2}\|y_f^{k+1} - y^*\|^2 + \frac{1}{\sigma_2}\left(1 - \frac{\sigma_1}{\sigma_2}\right)\|y_f^{k+1} - y_g^k\|^2.$$

Using Line 12 of Algorithm 1 we get

$$-\|y^{k+1} - y^*\|^2 \leq -\left(\frac{1}{\sigma_1} - \frac{1}{\sigma_2}\right)\|y_g^k - y^*\|^2 + \frac{(1 - \sigma_1)}{\sigma_1}\|y_f^k - y^*\|^2$$
$$- \frac{1}{\sigma_2}\|y_f^{k+1} - y^*\|^2 + (\sigma_2 - \sigma_1)\|y^{k+1} - y^k\|^2.$$

$\square$

**Lemma B.4.** *Let $\beta$ be defined as follows:*

$$\beta = 1/(2L). \tag{18}$$

*Let $\sigma_1$ be defined as follows:*

$$\sigma_1 = (1/\sigma_2 + 1/2)^{-1}. \tag{19}$$

*Then the following inequality holds:*

$$\left(\frac{1}{\theta} + \frac{\beta}{2}\right)\mathbb{E}\left[\|y^{k+1} - y^*\|^2\right] + \frac{\beta}{2\sigma_2}\mathbb{E}\left[\|y_f^{k+1} - y^*\|^2\right]$$
$$\leq \frac{1}{\theta}\|y^k - y^*\|^2 + \frac{\beta(1 - \sigma_2/2)}{2\sigma_2}\|y_f^k - y^*\|^2 - 2\mathbb{E}\left[\langle x^{k+1} - x^*, y^{k+1} - y^*\rangle\right]$$
$$+ G_F(x_g^k, x^*) - 2\nu^{-1}\mathbb{E}\left[\langle y_g^k + z_g^k - (y^* + z^*), y^{k+1} - y^*\rangle\right] - \frac{\beta}{4}\|y_g^k - y^*\|^2$$
$$+ \left(\frac{\beta\sigma_2^2}{4} - \frac{1}{\theta}\right)\mathbb{E}\left[\|y^{k+1} - y^k\|^2\right]$$
$$+ \frac{\overline{L}}{Lb}\left(G_F(\omega^k, x^*) - G_F(x_g^k, x^*) - \langle\nabla F(x_g^k) - \nabla F(x^*) - \nu x_g^k + \nu x^*, \omega^k - x_g^k\rangle\right). \tag{20}$$

*Proof.*

$$\frac{1}{\theta}\|y^{k+1} - y^*\|^2 = \frac{1}{\theta}\|y^k - y^*\|^2 + \frac{2}{\theta}\langle x^{k+1} - x^k, x^{k+1} - x^*\rangle - \frac{1}{\theta}\|y^{k+1} - y^k\|^2.$$

Using Line 11 of Algorithm 1 we get

$$\frac{1}{\theta}\|y^{k+1} - y^*\|^2 = \frac{1}{\theta}\|y^k - y^*\|^2 + 2\beta\langle\nabla^k - \nu x_g^k - y^{k+1}, y^{k+1} - y^*\rangle$$
$$- 2\langle\nu^{-1}(y_g^k + z_g^k) + x^{k+1}, y^{k+1} - y^*\rangle - \frac{1}{\theta}\|y^{k+1} - y^k\|^2.$$

Using optimality condition (3) we get

$$\frac{1}{\theta}\|y^{k+1} - y^*\|^2 = \frac{1}{\theta}\|y^k - y^*\|^2 + 2\beta\langle\nabla^k - \nu x_g^k - (\nabla F(x^*) - \nu x^*) + y^* - y^{k+1}, y^{k+1} - y^*\rangle$$

$$-2\langle \nu^{-1}(y_g^k + z_g^k) + x^{k+1}, y^{k+1} - y^* \rangle - \frac{1}{\theta}\|y^{k+1} - y^k\|^2$$

$$= \frac{1}{\theta}\|y^k - y^*\|^2 + 2\beta\langle \nabla^k - \nu x_g^k - (\nabla F(x^*) - \nu x^*), y^{k+1} - y^* \rangle$$

$$- 2\beta\|y^{k+1} - y^*\|^2 - 2\langle \nu^{-1}(y_g^k + z_g^k) + x^{k+1}, y^{k+1} - y^* \rangle - \frac{1}{\theta}\|y^{k+1} - y^k\|^2$$

$$\leq \frac{1}{\theta}\|y^k - y^*\|^2 + \beta\|\nabla^k - \nu x_g^k - (\nabla F(x^*) - \nu x^*)\|^2 - \beta\|y^{k+1} - y^*\|^2$$

$$- 2\langle \nu^{-1}(y_g^k + z_g^k) + x^{k+1}, y^{k+1} - y^* \rangle - \frac{1}{\theta}\|y^{k+1} - y^k\|^2.$$

Taking expectation over $i$ and using the property $\mathbb{E}\left[\|\xi\|^2\right] = \mathbb{E}\left[\|\xi - \mathbb{E}\left[\xi\right]\|^2\right] + \|\mathbb{E}\left[\xi\right]\|^2$ we get

$$\frac{1}{\theta}\mathbb{E}\left[\|y^{k+1} - y^*\|^2\right] \leq \frac{1}{\theta}\|y^k - y^*\|^2 + \beta\|\nabla F(x_g^k) - \nu x_g^k - (\nabla F(x^*) - \nu x^*)\|^2 - \beta\|y^{k+1} - y^*\|^2$$

$$- 2\langle \nu^{-1}(y_g^k + z_g^k) + x^{k+1}, y^{k+1} - y^* \rangle$$

$$- \frac{1}{\theta}\|y^{k+1} - y^k\|^2 + \beta\mathbb{E}\left[\|\nabla^k - \nabla F(x_g^k)\|^2\right].$$

Function $F(x) - \frac{\nu}{2}\|x\|^2$ is convex and $L$-smooth, together with (B.1) it implies

$$\frac{1}{\theta}\|y^{k+1} - y^*\|^2 \leq \frac{1}{\theta}\|y^k - y^*\|^2 + 2\beta L\left(\mathrm{D}_F(x_g^k, x^*) - \frac{\nu}{2}\|x_g^k - x^*\|^2\right) - \beta\mathbb{E}\left[\|y^{k+1} - y^*\|^2\right]$$

$$- 2\mathbb{E}\left[\langle \nu^{-1}(y_g^k + z_g^k) + x^{k+1}, y^{k+1} - y^* \rangle\right] - \frac{1}{\theta}\mathbb{E}\left[\|y^{k+1} - y^k\|^2\right]$$

$$+ \frac{2\overline{L}\beta}{b}\left(\mathrm{G}_F(\omega^k, x^*) - \mathrm{G}_F(x_g^k, x^*) - \langle \nabla F(x_g^k) - \nabla F(x^*) - \nu x_g^k + \nu x^*, \omega^k - x_g^k \rangle\right).$$

Using $\beta$ definition (18) we get

$$\frac{1}{\theta}\mathbb{E}\left[\|y^{k+1} - y^*\|^2\right] \leq \frac{1}{\theta}\|y^k - y^*\|^2 + \mathrm{G}_F(x_g^k, x^*) - \beta\mathbb{E}\left[\|y^{k+1} - y^*\|^2\right]$$

$$- 2\mathbb{E}\left[\langle \nu^{-1}(y_g^k + z_g^k) + x^{k+1}, y^{k+1} - y^* \rangle\right] - \frac{1}{\theta}\mathbb{E}\left[\|y^{k+1} - y^k\|^2\right]$$

$$+ \frac{\overline{L}}{Lb}\left(\mathrm{G}_F(\omega^k, x^*) - \mathrm{G}_F(x_g^k, x^*) - \langle \nabla F(x_g^k) - \nabla F(x^*) - \nu x_g^k + \nu x^*, \omega^k - x_g^k \rangle\right).$$

Using optimality condition (4) we get

$$\frac{1}{\theta}\mathbb{E}\left[\|y^{k+1} - y^*\|^2\right] \leq \frac{1}{\theta}\|y^k - y^*\|^2 - \beta\mathbb{E}\left[\|y^{k+1} - y^*\|^2\right]$$

$$- 2\nu^{-1}\mathbb{E}\left[\langle y_g^k + z_g^k - (y^* + z^*), y^{k+1} - y^* \rangle\right]$$

$$- 2\mathbb{E}\left[\langle x^{k+1} - x^*, y^{k+1} - y^* \rangle\right] - \frac{1}{\theta}\mathbb{E}\left[\|y^{k+1} - y^k\|^2\right] + \mathrm{G}_F(x_g^k, x^*)$$

$$+ \frac{\overline{L}}{Lb}\left(\mathrm{G}_F(\omega^k, x^*) - \mathrm{G}_F(x_g^k, x^*) - \langle \nabla F(x_g^k) - \nabla F(x^*) - \nu x_g^k + \nu x^*, \omega^k - x_g^k \rangle\right).$$

Using (17) together with $\sigma_1$ definition (19) we get

$$\frac{1}{\theta}\mathbb{E}\left[\|y^{k+1} - y^*\|^2\right] \leq \frac{1}{\theta}\|y^k - y^*\|^2 - \frac{\beta}{2}\mathbb{E}\left[\|y^{k+1} - y^*\|^2\right] + \frac{\beta(1 - \sigma_2/2)}{2\sigma_2}\|y_f^k - y^*\|^2$$

$$- \frac{\beta}{2\sigma_2}\mathbb{E}\left[\|y_f^{k+1} - y^*\|^2\right] - \frac{\beta}{4}\|y_g^k - y^*\|^2 + \frac{\beta(\sigma_2 - \sigma_1)}{2}\mathbb{E}\left[\|y^{k+1} - y^k\|^2\right]$$

$$+ \mathrm{G}_F(x_g^k, x^*) - 2\nu^{-1}\mathbb{E}\left[\langle y_g^k + z_g^k - (y^* + z^*), y^{k+1} - y^* \rangle\right]$$

$$- 2\mathbb{E}\left[\langle x^{k+1} - x^*, y^{k+1} - y^* \rangle\right] - \frac{1}{\theta}\mathbb{E}\left[\|y^{k+1} - y^k\|^2\right]$$

$$+ \frac{\overline{L}}{Lb}\left(\mathrm{G}_F(\omega^k, x^*) - \mathrm{G}_F(x_g^k, x^*) - \langle \nabla F(x_g^k) - \nabla F(x^*) - \nu x_g^k + \nu x^*, \omega^k - x_g^k \rangle\right).$$

$$
\leq \frac{1}{\theta}\|y^k - y^*\|^2 - \frac{\beta}{2}\mathbb{E}\left[\|y^{k+1} - y^*\|^2\right] + \frac{\beta(1 - \sigma_2/2)}{2\sigma_2}\|y_f^k - y^*\|^2
$$

$$
- \frac{\beta}{2\sigma_2}\mathbb{E}\left[\|y_f^{k+1} - y^*\|^2\right] - \frac{\beta}{4}\|y_g^k - y^*\|^2 + \left(\frac{\beta\sigma_2^2}{4} - \frac{1}{\theta}\right)\mathbb{E}\left[\|y^{k+1} - y^k\|^2\right]
$$

$$
+ \mathrm{G}_F(x_g^k, x^*) - 2\nu^{-1}\mathbb{E}\left[\langle y_g^k + z_g^k - (y^* + z^*), y^{k+1} - y^*\rangle\right]
$$

$$
- 2\mathbb{E}\left[\langle x^{k+1} - x^*, y^{k+1} - y^*\rangle\right]
$$

$$
+ \frac{\overline{L}}{Lb}\left(\mathrm{G}_F(\omega^k, x^*) - \mathrm{G}_F(x_g^k, x^*) - \langle \nabla F(x_g^k) - \nabla F(x^*) - \nu x_g^k + \nu x^*, \omega^k - x_g^k\rangle\right).
$$

Rearranging gives

$$
\left(\frac{1}{\theta} + \frac{\beta}{2}\right)\mathbb{E}\left[\|y^{k+1} - y^*\|^2\right] + \frac{\beta}{2\sigma_2}\mathbb{E}\left[\|y_f^{k+1} - y^*\|^2\right]
$$

$$
\leq \frac{1}{\theta}\|y^k - y^*\|^2 + \frac{\beta(1 - \sigma_2/2)}{2\sigma_2}\|y_f^k - y^*\|^2 - 2\mathbb{E}\left[\langle x^{k+1} - x^*, y^{k+1} - y^*\rangle\right]
$$

$$
+ \mathrm{G}_F(x_g^k, x^*) - 2\nu^{-1}\mathbb{E}\left[\langle y_g^k + z_g^k - (y^* + z^*), y^{k+1} - y^*\rangle\right]
$$

$$
- \frac{\beta}{4}\|y_g^k - y^*\|^2 + \left(\frac{\beta\sigma_2^2}{4} - \frac{1}{\theta}\right)\mathbb{E}\left[\|y^{k+1} - y^k\|^2\right]
$$

$$
+ \frac{\overline{L}}{Lb}\left(\mathrm{G}_F(\omega^k, x^*) - \mathrm{G}_F(x_g^k, x^*) - \langle \nabla F(x_g^k) - \nabla F(x^*) - \nu x_g^k + \nu x^*, \omega^k - x_g^k\rangle\right).
$$

$\square$

**Lemma B.5.** *The following inequality holds:*

$$\|m^k\|_{\mathbf{P}}^2 \le 8\chi^2\gamma^2\nu^{-2}\|y_g^k + z_g^k\|_{\mathbf{P}}^2 + 4\chi(1 - (4\chi)^{-1})\|m^k\|_{\mathbf{P}}^2 - 4\chi\|m^{k+1}\|_{\mathbf{P}}^2. \qquad (21)$$

*Proof.* Using Line 15 of Algorithm 1 we get

$$\|m^{k+1}\|_{\mathbf{P}}^2 = \|\gamma\nu^{-1}(y_g^k + z_g^k) + m^k - (\mathbf{W}(k) \otimes \mathbf{I}_d)\left[\gamma\nu^{-1}(y_g^k + z_g^k) + m^k\right]\|_{\mathbf{P}}^2$$
$$= \|\mathbf{P}\left[\gamma\nu^{-1}(y_g^k + z_g^k) + m^k\right] - (\mathbf{W}(k) \otimes \mathbf{I}_d)\mathbf{P}\left[\gamma\nu^{-1}(y_g^k + z_g^k) + m^k\right]\|^2.$$

Using property (2) we obtain

$$\|m^{k+1}\|_{\mathbf{P}}^2 \le (1 - \chi^{-1})\|m^k + \gamma\nu^{-1}(y_g^k + z_g^k)\|_{\mathbf{P}}^2.$$

Using inequality $\|a + b\|^2 \le (1 + c)\|a\|^2 + (1 + c^{-1})\|b\|^2$ with $c = \frac{1}{2(\chi - 1)}$ we get

$$\|m^{k+1}\|_{\mathbf{P}}^2 \le (1 - \chi^{-1})\left[\left(1 + \frac{1}{2(\chi - 1)}\right)\|m^k\|_{\mathbf{P}}^2 + (1 + 2(\chi - 1))\gamma^2\nu^{-2}\|y_g^k + z_g^k\|_{\mathbf{P}}^2\right]$$
$$\le (1 - (2\chi)^{-1})\|m^k\|_{\mathbf{P}}^2 + 2\chi\gamma^2\nu^{-2}\|y_g^k + z_g^k\|_{\mathbf{P}}^2.$$

Rearranging gives

$$\|m^k\|_{\mathbf{P}}^2 \le 8\chi^2\gamma^2\nu^{-2}\|y_g^k + z_g^k\|_{\mathbf{P}}^2 + 4\chi(1 - (4\chi)^{-1})\|m^k\|_{\mathbf{P}}^2 - 4\chi\|m^{k+1}\|_{\mathbf{P}}^2.$$

$\square$

**Lemma B.6.** *Let $\hat{z}^k$ be defined as follows:*

$$\hat{z}^k = z^k - \mathbf{P}m^k. \qquad (22)$$

*Then the following inequality holds:*

$$\frac{1}{\gamma}\|\hat{z}^{k+1} - z^*\|^2 + \frac{4}{3\gamma}\|m^{k+1}\|_{\mathbf{P}}^2 \le \left(\frac{1}{\gamma} - \delta\right)\|\hat{z}^k - z^*\|^2 + \left(1 - (4\chi)^{-1} + \frac{3\gamma\delta}{2}\right)\frac{4}{3\gamma}\|m^k\|_{\mathbf{P}}^2$$
$$- 2\nu^{-1}\langle y_g^k + z_g^k - (y^* + z^*), z^k - z^*\rangle + \gamma\nu^{-2}(1 + 6\chi)\|y_g^k + z_g^k\|_{\mathbf{P}}^2$$
$$+ 2\delta\|z_g^k - z^*\|^2 + (2\gamma\delta^2 - \delta)\|z_g^k - z^k\|^2.$$

$$(23)$$

*Proof.*

$$\frac{1}{\gamma}\|\hat{z}^{k+1} - z^*\|^2 = \frac{1}{\gamma}\|\hat{z}^k - z^*\|^2 + \frac{2}{\gamma}\langle\hat{z}^{k+1} - \hat{z}^k, \hat{z}^k - z^*\rangle + \frac{1}{\gamma}\|\hat{z}^{k+1} - \hat{z}^k\|^2.$$

The combination of Lines 14 and 15 in Algorithm 1, coupled with the definition of $\hat{z}^k$ in (22), imply

$$\hat{z}^{k+1} - \hat{z}^k = \gamma\delta(z_g^k - z^k) - \gamma\nu^{-1}\mathbf{P}(y_g^k + z_g^k).$$

Hence,

$$\frac{1}{\gamma}\|\hat{z}^{k+1} - z^*\|^2 = \frac{1}{\gamma}\|\hat{z}^k - z^*\|^2 + 2\delta\langle z_g^k - z^k, \hat{z}^k - z^*\rangle$$
$$- 2\nu^{-1}\langle\mathbf{P}(y_g^k + z_g^k), \hat{z}^k - z^*\rangle + \frac{1}{\gamma}\|\hat{z}^{k+1} - \hat{z}^k\|^2$$
$$= \frac{1}{\gamma}\|\hat{z}^k - z^*\|^2 + \delta\|z_g^k - \mathbf{P}m^k - z^*\|^2 - \delta\|\hat{z}^k - z^*\|^2 - \delta\|z_g^k - z^k\|^2$$
$$- 2\nu^{-1}\langle\mathbf{P}(y_g^k + z_g^k), \hat{z}^k - z^*\rangle + \gamma\|\delta(z_g^k - z^k) - \nu^{-1}\mathbf{P}(y_g^k + z_g^k)\|^2$$
$$\le \left(\frac{1}{\gamma} - \delta\right)\|\hat{z}^k - z^*\|^2 + 2\delta\|z_g^k - z^*\|^2 + 2\delta\|m^k\|_{\mathbf{P}}^2 - \delta\|z_g^k - z^k\|^2$$
$$- 2\nu^{-1}\langle\mathbf{P}(y_g^k + z_g^k), \hat{z}^k - z^*\rangle + 2\gamma\delta^2\|z_g^k - z^k\|^2 + \gamma\|\nu^{-1}\mathbf{P}(y_g^k + z_g^k)\|^2$$
$$\le \left(\frac{1}{\gamma} - \delta\right)\|\hat{z}^k - z^*\|^2 + 2\delta\|z_g^k - z^*\|^2 + (2\gamma\delta^2 - \delta)\|z_g^k - z^k\|^2$$

$$- 2\nu^{-1}\langle \mathbf{P}(y_g^k + z_g^k), z^k - z^* \rangle + \gamma \| \nu^{-1}\mathbf{P}(y_g^k + z_g^k) \|^2$$
$$+ 2\delta \| m^k \|_{\mathbf{P}}^2 + 2\nu^{-1}\langle \mathbf{P}(y_g^k + z_g^k), m^k \rangle.$$

Using the fact that $z^k \in \mathcal{L}^\perp$ for all $k = 0, 1, 2 \ldots$ and optimality condition (5) we get

$$\frac{1}{\gamma}\|\hat{z}^{k+1} - z^*\|^2 \le \left( \frac{1}{\gamma} - \delta \right) \|\hat{z}^k - z^*\|^2 + 2\delta\|z_g^k - z^*\|^2$$
$$+ \left( 2\gamma\delta^2 - \delta \right)\|z_g^k - z^k\|^2 + \gamma\nu^{-2}\|y_g^k + z_g^k\|_{\mathbf{P}}^2$$
$$- 2\nu^{-1}\langle y_g^k + z_g^k - (y^* + z^*), z^k - z^* \rangle$$
$$+ 2\delta\|m^k\|_{\mathbf{P}}^2 + 2\nu^{-1}\langle \mathbf{P}(y_g^k + z_g^k), m^k \rangle.$$

Using Young's inequality we get

$$\frac{1}{\gamma}\|\hat{z}^{k+1} - z^*\|^2 \le \left( \frac{1}{\gamma} - \delta \right) \|\hat{z}^k - z^*\|^2 + 2\delta\|z_g^k - z^*\|^2 + \left( 2\gamma\delta^2 - \delta \right)\|z_g^k - z^k\|^2$$
$$- 2\nu^{-1}\langle y_g^k + z_g^k - (y^* + z^*), z^k - z^* \rangle + \gamma\nu^{-2}\|y_g^k + z_g^k\|_{\mathbf{P}}^2$$
$$+ 2\delta\|m^k\|_{\mathbf{P}}^2 + 3\gamma\chi\nu^{-2}\|y_g^k + z_g^k\|_{\mathbf{P}}^2 + \frac{1}{3\gamma\chi}\|m^k\|_{\mathbf{P}}^2.$$

Using (21) we get

$$\frac{1}{\gamma}\|\hat{z}^{k+1} - z^*\|^2 \le \left( \frac{1}{\gamma} - \delta \right) \|\hat{z}^k - z^*\|^2 + 2\delta\|z_g^k - z^*\|^2 + \left( 2\gamma\delta^2 - \delta \right)\|z_g^k - z^k\|^2$$
$$- 2\nu^{-1}\langle y_g^k + z_g^k - (y^* + z^*), z^k - z^* \rangle + \gamma\nu^{-2}\|y_g^k + z_g^k\|_{\mathbf{P}}^2$$
$$+ 2\delta\|m^k\|_{\mathbf{P}}^2 + 6\gamma\nu^{-2}\chi\|y_g^k + z_g^k\|_{\mathbf{P}}^2 + \frac{4(1 - (4\chi)^{-1})}{3\gamma}\|m^k\|_{\mathbf{P}}^2 - \frac{4}{3\gamma}\|m^{k+1}\|_{\mathbf{P}}^2$$
$$= \left( \frac{1}{\gamma} - \delta \right) \|\hat{z}^k - z^*\|^2 + 2\delta\|z_g^k - z^*\|^2 + \left( 2\gamma\delta^2 - \delta \right)\|z_g^k - z^k\|^2$$
$$- 2\nu^{-1}\langle y_g^k + z_g^k - (y^* + z^*), z^k - z^* \rangle + \gamma\nu^{-2}\left( 1 + 6\chi \right)\|y_g^k + z_g^k\|_{\mathbf{P}}^2$$
$$+ \left( 1 - (4\chi)^{-1} + \frac{3\gamma\delta}{2} \right)\frac{4}{3\gamma}\|m^k\|_{\mathbf{P}}^2 - \frac{4}{3\gamma}\|m^{k+1}\|_{\mathbf{P}}^2.$$

$\square$

**Lemma B.7.** *The following inequality holds:*

$$2\langle y_g^k + z_g^k - (y^* + z^*), y^k + z^k - (y^* + z^*) \rangle \ge 2\|y_g^k + z_g^k - (y^* + z^*)\|^2$$
$$+ \frac{(1 - \sigma_2/2)}{\sigma_2}\left( \|y_g^k + z_g^k - (y^* + z^*)\|^2 - \|y_f^k + z_f^k - (y^* + z^*)\|^2 \right). \tag{24}$$

*Proof.*

$$2\langle y_g^k + z_g^k - (y^* + z^*), y^k + z^k - (y^* + z^*) \rangle$$
$$= 2\|y_g^k + z_g^k - (y^* + z^*)\|^2 + 2\langle y_g^k + z_g^k - (y^* + z^*), y^k + z^k - (y_g^k + z_g^k) \rangle.$$

Using Lines 10 and 13 of Algorithm 1 we get

$$2\langle y_g^k + z_g^k - (y^* + z^*), y^k + z^k - (y^* + z^*) \rangle$$
$$= 2\|y_g^k + z_g^k - (y^* + z^*)\|^2 + \frac{2(1 - \sigma_1)}{\sigma_1}\langle y_g^k + z_g^k - (y^* + z^*), y_g^k + z_g^k - (y_f^k + z_f^k) \rangle$$
$$= 2\|y_g^k + z_g^k - (y^* + z^*)\|^2$$
$$+ \frac{(1 - \sigma_1)}{\sigma_1}\left( \|y_g^k + z_g^k - (y^* + z^*)\|^2 + \|y_g^k + z_g^k - (y_f^k + z_f^k)\|^2 - \|y_f^k + z_f^k - (y^* + z^*)\|^2 \right)$$
$$\ge 2\|y_g^k + z_g^k - (y^* + z^*)\|^2$$

$$+ \frac{(1 - \sigma_1)}{\sigma_1} \left( \|y_g^k + z_g^k - (y^* + z^*)\|^2 - \|y_f^k + z_f^k - (y^* + z^*)\|^2 \right).$$

Using $\sigma_1$ definition (19) we get

$$2\langle y_g^k + z_g^k - (y^* + z^*), y^k + z^k - (y^* + z^*) \rangle \geq 2\|y_g^k + z_g^k - (y^* + z^*)\|^2$$
$$+ \frac{(1 - \sigma_2/2)}{\sigma_2} \left( \|y_g^k + z_g^k - (y^* + z^*)\|^2 - \|y_f^k + z_f^k - (y^* + z^*)\|^2 \right).$$

$\square$

**Lemma B.8.** *Let $\zeta$ be defined by*

$$\zeta = 1/2. \tag{25}$$

*Then the following inequality holds:*

$$-2\langle y^{k+1} - y^k, y_g^k + z_g^k - (y^* + z^*) \rangle$$
$$\leq \frac{1}{\sigma_2} \|y_g^k + z_g^k - (y^* + z^*)\|^2 - \frac{1}{\sigma_2} \|y_f^{k+1} + z_f^{k+1} - (y^* + z^*)\|^2 \tag{26}$$
$$+ 2\sigma_2 \|y^{k+1} - y^k\|^2 - \frac{1}{2\sigma_2\chi} \|y_g^k + z_g^k\|_{\mathbf{P}}^2.$$

*Proof.*

$$\|y_f^{k+1} + z_f^{k+1} - (y^* + z^*)\|^2$$
$$= \|y_g^k + z_g^k - (y^* + z^*)\|^2 + 2\langle y_f^{k+1} + z_f^{k+1} - (y_g^k + z_g^k), y_g^k + z_g^k - (y^* + z^*) \rangle$$
$$+ \|y_f^{k+1} + z_f^{k+1} - (y_g^k + z_g^k)\|^2$$
$$\leq \|y_g^k + z_g^k - (y^* + z^*)\|^2 + 2\langle y_f^{k+1} + z_f^{k+1} - (y_g^k + z_g^k), y_g^k + z_g^k - (y^* + z^*) \rangle$$
$$+ 2\|y_f^{k+1} - y_g^k\|^2 + 2\|z_f^{k+1} - z_g^k\|^2.$$

Using Line 12 of Algorithm 1 we get

$$\|y_f^{k+1} + z_f^{k+1} - (y^* + z^*)\|^2$$
$$\leq \|y_g^k + z_g^k - (y^* + z^*)\|^2 + 2\sigma_2\langle y^{k+1} - y^k, y_g^k + z_g^k - (y^* + z^*) \rangle$$
$$+ 2\sigma_2^2 \|y^{k+1} - y^k\|^2 + 2\langle z_f^{k+1} - z_g^k, y_g^k + z_g^k - (y^* + z^*) \rangle + 2\|z_f^{k+1} - z_g^k\|^2.$$

Using Line 16 of Algorithm 1 and optimality condition (5) we get

$$\|y_f^{k+1} + z_f^{k+1} - (y^* + z^*)\|^2$$
$$\leq \|y_g^k + z_g^k - (y^* + z^*)\|^2 + 2\sigma_2\langle y^{k+1} - y^k, y_g^k + z_g^k - (y^* + z^*) \rangle + 2\sigma_2^2 \|y^{k+1} - y^k\|^2$$
$$- 2\zeta\langle (\mathbf{W}(k) \otimes \mathbf{I}_d)(y_g^k + z_g^k), y_g^k + z_g^k - (y^* + z^*) \rangle + 2\zeta^2 \|(\mathbf{W}(k) \otimes \mathbf{I}_d)(y_g^k + z_g^k)\|^2$$
$$= \|y_g^k + z_g^k - (y^* + z^*)\|^2 + 2\sigma_2\langle y^{k+1} - y^k, y_g^k + z_g^k - (y^* + z^*) \rangle + 2\sigma_2^2 \|y^{k+1} - y^k\|^2$$
$$- 2\zeta\langle (\mathbf{W}(k) \otimes \mathbf{I}_d)(y_g^k + z_g^k), y_g^k + z_g^k \rangle + 2\zeta^2 \|(\mathbf{W}(k) \otimes \mathbf{I}_d)(y_g^k + z_g^k)\|^2.$$

Using $\zeta$ definition (25) we get

$$\|y_f^{k+1} + z_f^{k+1} - (y^* + z^*)\|^2$$
$$\leq \|y_g^k + z_g^k - (y^* + z^*)\|^2 + 2\sigma_2\langle y^{k+1} - y^k, y_g^k + z_g^k - (y^* + z^*) \rangle + 2\sigma_2^2 \|y^{k+1} - y^k\|^2$$
$$- \langle (\mathbf{W}(k) \otimes \mathbf{I}_d)(y_g^k + z_g^k), y_g^k + z_g^k \rangle + \frac{1}{2} \|(\mathbf{W}(k) \otimes \mathbf{I}_d)(y_g^k + z_g^k)\|^2$$
$$= \|y_g^k + z_g^k - (y^* + z^*)\|^2 + 2\sigma_2\langle y^{k+1} - y^k, y_g^k + z_g^k - (y^* + z^*) \rangle + 2\sigma_2^2 \|y^{k+1} - y^k\|^2$$
$$- \frac{1}{2} \|(\mathbf{W}(k) \otimes \mathbf{I}_d)(y_g^k + z_g^k)\|^2 - \frac{1}{2} \|y_g^k + z_g^k\|^2$$
$$+ \frac{1}{2} \|(\mathbf{W}(k) \otimes \mathbf{I}_d)(y_g^k + z_g^k) - (y_g^k + z_g^k)\|^2 + \frac{1}{2} \|(\mathbf{W}(k) \otimes \mathbf{I}_d)(y_g^k + z_g^k)\|^2$$

$$\leq \|y_g^k + z_g^k - (y^* + z^*)\|^2 + 2\sigma_2 \langle y^{k+1} - y^k, y_g^k + z_g^k - (y^* + z^*) \rangle + 2\sigma_2^2 \|y^{k+1} - y^k\|^2$$
$$- \frac{1}{2} \|y_g^k + z_g^k\|_{\mathbf{P}}^2 + \frac{1}{2} \|(\mathbf{W}(k) \otimes \mathbf{I}_d)(y_g^k + z_g^k) - (y_g^k + z_g^k)\|_{\mathbf{P}}^2.$$
$$= \|y_g^k + z_g^k - (y^* + z^*)\|^2 + 2\sigma_2 \langle y^{k+1} - y^k, y_g^k + z_g^k - (y^* + z^*) \rangle + 2\sigma_2^2 \|y^{k+1} - y^k\|^2$$
$$- \frac{1}{2} \|y_g^k + z_g^k\|_{\mathbf{P}}^2 + \frac{1}{2} \|(\mathbf{W}(k) \otimes \mathbf{I}_d)\mathbf{P}(y_g^k + z_g^k) - \mathbf{P}(y_g^k + z_g^k)\|^2.$$

Using condition (2) we get

$$\|y_f^{k+1} + z_f^{k+1} - (y^* + z^*)\|^2$$
$$\leq \|y_g^k + z_g^k - (y^* + z^*)\|^2 + 2\sigma_2 \langle y^{k+1} - y^k, y_g^k + z_g^k - (y^* + z^*) \rangle + 2\sigma_2^2 \|y^{k+1} - y^k\|^2$$
$$- (2\chi)^{-1} \|y_g^k + z_g^k\|_{\mathbf{P}}^2.$$

Rearranging gives

$$- 2\langle y^{k+1} - y^k, y_g^k + z_g^k - (y^* + z^*) \rangle$$
$$\leq \frac{1}{\sigma_2} \|y_g^k + z_g^k - (y^* + z^*)\|^2 - \frac{1}{\sigma_2} \|y_f^{k+1} + z_f^{k+1} - (y^* + z^*)\|^2$$
$$+ 2\sigma_2 \|y^{k+1} - y^k\|^2 - \frac{1}{2\sigma_2\chi} \|y_g^k + z_g^k\|_{\mathbf{P}}^2.$$

$\square$

**Lemma B.9.** *Let $\delta$ be defined as follows:*

$$\delta = \frac{1}{17L}. \tag{27}$$

*Let $\gamma$ be defined as follows:*

$$\gamma = \frac{\nu}{14\sigma_2\chi^2}. \tag{28}$$

*Let $\theta$ be defined as follows:*

$$\theta = \frac{\nu}{4\sigma_2}. \tag{29}$$

*Let $\sigma_2$ be defined as follows:*

$$\sigma_2 = \frac{\sqrt{\mu}}{16\chi\sqrt{L}}. \tag{30}$$

*Let $\Psi_{yz}^k$ be the following Lyapunov function*

$$\Psi_{yz}^k = \left( \frac{1}{\theta} + \frac{\beta}{2} \right) \|y^k - y^*\|^2 + \frac{\beta}{2\sigma_2} \|y_f^k - y^*\|^2 + \frac{1}{\gamma} \|\hat{z}^k - z^*\|^2$$
$$+ \frac{4}{3\gamma} \|m^k\|_{\mathbf{P}}^2 + \frac{\nu^{-1}}{\sigma_2} \|y_f^k + z_f^k - (y^* + z^*)\|^2. \tag{31}$$

*Then the following inequality holds:*

$$\mathbb{E}\left[ \Psi_{yz}^{k+1} \right] \left( 1 - \frac{\sqrt{\mu}}{32\chi\sqrt{L}} \right) \Psi_{yz}^k - 2\mathbb{E}\left[ \langle x^{k+1} - x^*, y^{k+1} - y^* \rangle \right] + \mathrm{G}_F(x_g^k, x^*)$$
$$+ \frac{\overline{L}}{Lb} \left( \mathrm{G}_F(\omega^k, x^*) - \mathrm{G}_F(x_g^k, x^*) - \langle \nabla F(x_g^k) - \nabla F(x^*) - \nu x_g^k + \nu x^*, \omega^k - x_g^k \rangle \right). \tag{32}$$

*Proof.* Combining (20) and (23) gives

$$\left( \frac{1}{\theta} + \frac{\beta}{2} \right) \mathbb{E}\left[ \|y^{k+1} - y^*\|^2 \right] + \frac{\beta}{2\sigma_2} \mathbb{E}\left[ \|y_f^{k+1} - y^*\|^2 \right] + \frac{1}{\gamma} \|\hat{z}^{k+1} - z^*\|^2 + \frac{4}{3\gamma} \|m^{k+1}\|_{\mathbf{P}}^2$$
$$\leq \left( \frac{1}{\gamma} - \delta \right) \|\hat{z}^k - z^*\|^2 + \left( 1 - (4\chi)^{-1} + \frac{3\gamma\delta}{2} \right) \frac{4}{3\gamma} \|m^k\|_{\mathbf{P}}^2 + \frac{1}{\theta} \|y^k - y^*\|^2$$

$$+ \frac{\beta(1 - \sigma_2/2)}{2\sigma_2} \|y_f^k - y^*\|^2 - 2\nu^{-1}\langle y_g^k + z_g^k - (y^* + z^*), y^k + z^k - (y^* + z^*)\rangle$$

$$- 2\nu^{-1}\mathbb{E}\left[\langle y_g^k + z_g^k - (y^* + z^*), y^{k+1} - y^k\rangle\right] + \gamma\nu^{-2}(1 + 6\chi)\|y_g^k + z_g^k\|_{\mathbf{P}}^2$$

$$+ \left(\frac{\beta\sigma_2^2}{4} - \frac{1}{\theta}\right)\mathbb{E}\left[\|y^{k+1} - y^k\|^2\right] + 2\delta\|z_g^k - z^*\|^2 - \frac{\beta}{4}\|y_g^k - y^*\|^2$$

$$- 2\mathbb{E}\left[\langle x^{k+1} - x^*, y^{k+1} - y^*\rangle\right] + \left(2\gamma\delta^2 - \delta\right)\|z_g^k - z^k\|^2 + \mathrm{G}_F(x_g^k, x^*)$$

$$+ \frac{\overline{L}}{Lb}\left(\mathrm{G}_F(\omega^k, x^*) - \mathrm{G}_F(x_g^k, x^*) - \langle \nabla F(x_g^k) - \nabla F(x^*) - \nu x_g^k + \nu x^*, \omega^k - x_g^k\rangle\right).$$

Using (24) and (26) we get

$$\left(\frac{1}{\theta} + \frac{\beta}{2}\right)\mathbb{E}\left[\|y^{k+1} - y^*\|^2\right] + \frac{\beta}{2\sigma_2}\mathbb{E}\left[\|y_f^{k+1} - y^*\|^2\right] + \frac{1}{\gamma}\|\hat{z}^{k+1} - z^*\|^2 + \frac{4}{3\gamma}\|m^{k+1}\|_{\mathbf{P}}^2$$

$$\leq \left(\frac{1}{\gamma} - \delta\right)\|\hat{z}^k - z^*\|^2 + \left(1 - (4\chi)^{-1} + \frac{3\gamma\delta}{2}\right)\frac{4}{3\gamma}\|m^k\|_{\mathbf{P}}^2 + \frac{1}{\theta}\|y^k - y^*\|^2$$

$$+ \frac{\beta(1 - \sigma_2/2)}{2\sigma_2}\|y_f^k - y^*\|^2 - 2\nu^{-1}\|y_g^k + z_g^k - (y^* + z^*)\|^2$$

$$+ \frac{\nu^{-1}(1 - \sigma_2/2)}{\sigma_2}\left(\|y_f^k + z_f^k - (y^* + z^*)\|^2 - \|y_g^k + z_g^k - (y^* + z^*)\|^2\right)$$

$$+ \frac{\nu^{-1}}{\sigma_2}\|y_g^k + z_g^k - (y^* + z^*)\|^2 - \frac{\nu^{-1}}{\sigma_2}\mathbb{E}\left[\|y_f^{k+1} + z_f^{k+1} - (y^* + z^*)\|^2\right]$$

$$+ 2\nu^{-1}\sigma_2\mathbb{E}\left[\|y^{k+1} - y^k\|^2\right] - \frac{\nu^{-1}}{2\sigma_2\chi}\|y_g^k + z_g^k\|_{\mathbf{P}}^2 + \gamma\nu^{-2}(1 + 6\chi)\|y_g^k + z_g^k\|_{\mathbf{P}}^2$$

$$+ \left(\frac{\beta\sigma_2^2}{4} - \frac{1}{\theta}\right)\mathbb{E}\left[\|y^{k+1} - y^k\|^2\right] + 2\delta\|z_g^k - z^*\|^2$$

$$- \frac{\beta}{4}\|y_g^k - y^*\|^2 - 2\mathbb{E}\left[\langle x^{k+1} - x^*, y^{k+1} - y^*\rangle\right]$$

$$+ \left(2\gamma\delta^2 - \delta\right)\|z_g^k - z^k\|^2 + \mathrm{G}_F(x_g^k, x^*)$$

$$+ \frac{\overline{L}}{Lb}\left(\mathrm{G}_F(\omega^k, x^*) - \mathrm{G}_F(x_g^k, x^*) - \langle \nabla F(x_g^k) - \nabla F(x^*) - \nu x_g^k + \nu x^*, \omega^k - x_g^k\rangle\right)$$

$$= \left(\frac{1}{\gamma} - \delta\right)\|\hat{z}^k - z^*\|^2 + \left(1 - (4\chi)^{-1} + \frac{3\gamma\delta}{2}\right)\frac{4}{3\gamma}\|m^k\|_{\mathbf{P}}^2 + \frac{1}{\theta}\|y^k - y^*\|^2$$

$$+ \frac{\beta(1 - \sigma_2/2)}{2\sigma_2}\|y_f^k - y^*\|^2 + \frac{\nu^{-1}(1 - \sigma_2/2)}{\sigma_2}\|y_f^k + z_f^k - (y^* + z^*)\|^2$$

$$- \frac{\nu^{-1}}{\sigma_2}\mathbb{E}\left[\|y_f^{k+1} + z_f^{k+1} - (y^* + z^*)\|^2\right]$$

$$+ 2\delta\|z_g^k - z^*\|^2 - \frac{\beta}{4}\|y_g^k - y^*\|^2 + \nu^{-1}\left(\frac{1}{\sigma_2} - \frac{(1 - \sigma_2/2)}{\sigma_2} - 2\right)\|y_g^k + z_g^k - (y^* + z^*)\|^2$$

$$+ \left(\gamma\nu^{-2}(1 + 6\chi) - \frac{\nu^{-1}}{2\sigma_2\chi}\right)\|y_g^k + z_g^k\|_{\mathbf{P}}^2 + \left(\frac{\beta\sigma_2^2}{4} + 2\nu^{-1}\sigma_2 - \frac{1}{\theta}\right)\mathbb{E}\left[\|y^{k+1} - y^k\|^2\right]$$

$$+ \left(2\gamma\delta^2 - \delta\right)\|z_g^k - z^k\|^2 - 2\mathbb{E}\left[\langle x^{k+1} - x^*, y^{k+1} - y^*\rangle\right] + \mathrm{G}_F(x_g^k, x^*)$$

$$+ \frac{\overline{L}}{Lb}\left(\mathrm{G}_F(\omega^k, x^*) - \mathrm{G}_F(x_g^k, x^*) - \langle \nabla F(x_g^k) - \nabla F(x^*) - \nu x_g^k + \nu x^*, \omega^k - x_g^k\rangle\right)$$

$$= \left(\frac{1}{\gamma} - \delta\right)\|\hat{z}^k - z^*\|^2 + \left(1 - (4\chi)^{-1} + \frac{3\gamma\delta}{2}\right)\frac{4}{3\gamma}\|m^k\|_{\mathbf{P}}^2 + \frac{1}{\theta}\|y^k - y^*\|^2$$

$$+ \frac{\beta(1 - \sigma_2/2)}{2\sigma_2}\|y_f^k - y^*\|^2 + \frac{\nu^{-1}(1 - \sigma_2/2)}{\sigma_2}\|y_f^k + z_f^k - (y^* + z^*)\|^2$$

$$- \frac{\nu^{-1}}{\sigma_2}\mathbb{E}\left[\|y_f^{k+1} + z_f^{k+1} - (y^* + z^*)\|^2\right]$$

$$+ 2\delta\|z_g^k - z^*\|^2 - \frac{\beta}{4}\|y_g^k - y^*\|^2 - \frac{3\nu^{-1}}{2}\|y_g^k + z_g^k - (y^* + z^*)\|^2 + \left(2\gamma\delta^2 - \delta\right)\|z_g^k - z^k\|^2$$

$$+ \left(\gamma\nu^{-2}\left(1 + 6\chi\right) - \frac{\nu^{-1}}{2\sigma_2\chi}\right)\|y_g^k + z_g^k\|_{\mathbf{P}}^2 + \left(\frac{\beta\sigma_2^2}{4} + 2\nu^{-1}\sigma_2 - \frac{1}{\theta}\right)\mathbb{E}\left[\|y^{k+1} - y^k\|^2\right]$$

$$+ 2\mathbb{E}\left[\langle x^{k+1} - x^*, y^{k+1} - y^*\rangle\right] + \mathrm{G}_F(x_g^k, x^*)$$

$$+ \frac{\overline{L}}{Lb}\left(\mathrm{G}_F(\omega^k, x^*) - \mathrm{G}_F(x_g^k, x^*) - \langle\nabla F(x_g^k) - \nabla F(x^*) - \nu x_g^k + \nu x^*, \omega^k - x_g^k\rangle\right).$$

Using $\beta$ definition (18) and $\nu$ definition (14) we get

$$\left(\frac{1}{\theta} + \frac{\beta}{2}\right)\mathbb{E}\left[\|y^{k+1} - y^*\|^2\right] + \frac{\beta}{2\sigma_2}\mathbb{E}\left[\|y_f^{k+1} - y^*\|^2\right] + \frac{1}{\gamma}\|\hat{z}^{k+1} - z^*\|^2 + \frac{4}{3\gamma}\|m^{k+1}\|_{\mathbf{P}}^2$$

$$\leq \left(\frac{1}{\gamma} - \delta\right)\|\hat{z}^k - z^*\|^2 + \left(1 - (4\chi)^{-1} + \frac{3\gamma\delta}{2}\right)\frac{4}{3\gamma}\|m^k\|_{\mathbf{P}}^2 + \frac{1}{\theta}\|y^k - y^*\|^2$$

$$+ \frac{\beta(1 - \sigma_2/2)}{2\sigma_2}\|y_f^k - y^*\|^2 + \frac{\nu^{-1}(1 - \sigma_2/2)}{\sigma_2}\|y_f^k + z_f^k - (y^* + z^*)\|^2$$

$$- \frac{\nu^{-1}}{\sigma_2}\mathbb{E}\left[\|y_f^{k+1} + z_f^{k+1} - (y^* + z^*)\|^2\right]$$

$$+ 2\delta\|z_g^k - z^*\|^2 - \frac{1}{8L}\|y_g^k - y^*\|^2 - \frac{3}{\mu}\|y_g^k + z_g^k - (y^* + z^*)\|^2 + \left(2\gamma\delta^2 - \delta\right)\|z_g^k - z^k\|^2$$

$$+ \left(\gamma\nu^{-2}\left(1 + 6\chi\right) - \frac{\nu^{-1}}{2\sigma_2\chi}\right)\|y_g^k + z_g^k\|_{\mathbf{P}}^2 + \left(\frac{\beta\sigma_2^2}{4} + 2\nu^{-1}\sigma_2 - \frac{1}{\theta}\right)\mathbb{E}\left[\|y^{k+1} - y^k\|^2\right]$$

$$- 2\mathbb{E}\left[\langle x^{k+1} - x^*, y^{k+1} - y^*\rangle\right] + \mathrm{G}_F(x_g^k, x^*)$$

$$+ \frac{\overline{L}}{Lb}\left(\mathrm{G}_F(\omega^k, x^*) - \mathrm{G}_F(x_g^k, x^*) - \langle\nabla F(x_g^k) - \nabla F(x^*) - \nu x_g^k + \nu x^*, \omega^k - x_g^k\rangle\right).$$

Using $\delta$ definition (27) we get

$$\left(\frac{1}{\theta} + \frac{\beta}{2}\right)\mathbb{E}\left[\|y^{k+1} - y^*\|^2\right] + \frac{\beta}{2\sigma_2}\mathbb{E}\left[\|y_f^{k+1} - y^*\|^2\right] + \frac{1}{\gamma}\|\hat{z}^{k+1} - z^*\|^2 + \frac{4}{3\gamma}\|m^{k+1}\|_{\mathbf{P}}^2$$

$$\leq \left(\frac{1}{\gamma} - \delta\right)\|\hat{z}^k - z^*\|^2 + \left(1 - (4\chi)^{-1} + \frac{3\gamma\delta}{2}\right)\frac{4}{3\gamma}\|m^k\|_{\mathbf{P}}^2 + \frac{1}{\theta}\|y^k - y^*\|^2$$

$$+ \frac{\beta(1 - \sigma_2/2)}{2\sigma_2}\|y_f^k - y^*\|^2 + \frac{\nu^{-1}(1 - \sigma_2/2)}{\sigma_2}\|y_f^k + z_f^k - (y^* + z^*)\|^2$$

$$- \frac{\nu^{-1}}{\sigma_2}\mathbb{E}\left[\|y_f^{k+1} + z_f^{k+1} - (y^* + z^*)\|^2\right]$$

$$+ \left(\gamma\nu^{-2}\left(1 + 6\chi\right) - \frac{\nu^{-1}}{2\sigma_2\chi}\right)\|y_g^k + z_g^k\|_{\mathbf{P}}^2 + \left(\frac{\beta\sigma_2^2}{4} + 2\nu^{-1}\sigma_2 - \frac{1}{\theta}\right)\mathbb{E}\left[\|y^{k+1} - y^k\|^2\right]$$

$$+ \left(2\gamma\delta^2 - \delta\right)\|z_g^k - z^k\|^2 - 2\mathbb{E}\left[\langle x^{k+1} - x^*, y^{k+1} - y^*\rangle\right] + \mathrm{G}_F(x_g^k, x^*)$$

$$+ \frac{\overline{L}}{Lb}\left(\mathrm{G}_F(\omega^k, x^*) - \mathrm{G}_F(x_g^k, x^*) - \langle\nabla F(x_g^k) - \nabla F(x^*) - \nu x_g^k + \nu x^*, \omega^k - x_g^k\rangle\right).$$

Using $\gamma$ definition (28) we get

$$\left(\frac{1}{\theta} + \frac{\beta}{2}\right)\mathbb{E}\left[\|y^{k+1} - y^*\|^2\right] + \frac{\beta}{2\sigma_2}\mathbb{E}\left[\|y_f^{k+1} - y^*\|^2\right] + \frac{1}{\gamma}\|\hat{z}^{k+1} - z^*\|^2 + \frac{4}{3\gamma}\|m^{k+1}\|_{\mathbf{P}}^2$$

$$\leq \left(\frac{1}{\gamma} - \delta\right)\|\hat{z}^k - z^*\|^2 + \left(1 - (4\chi)^{-1} + \frac{3\gamma\delta}{2}\right)\frac{4}{3\gamma}\|m^k\|_{\mathbf{P}}^2 + \frac{1}{\theta}\|y^k - y^*\|^2$$

$$+ \frac{\beta(1 - \sigma_2/2)}{2\sigma_2}\|y_f^k - y^*\|^2 + \frac{\nu^{-1}(1 - \sigma_2/2)}{\sigma_2}\|y_f^k + z_f^k - (y^* + z^*)\|^2$$

$$- \frac{\nu^{-1}}{\sigma_2}\mathbb{E}\left[\|y_f^{k+1} + z_f^{k+1} - (y^* + z^*)\|^2\right]$$

$$+ \left( \frac{\beta \sigma_2^2}{4} + 2\nu^{-1}\sigma_2 - \frac{1}{\theta} \right) \mathbb{E}\left[ \|y^{k+1} - y^k\|^2 \right] + \left( 2\gamma\delta^2 - \delta \right) \|z_g^k - z^k\|^2$$

$$- 2\mathbb{E}\left[ \langle x^{k+1} - x^*, y^{k+1} - y^* \rangle \right] + \mathrm{G}_F(x_g^k, x^*)$$

$$+ \frac{\overline{L}}{Lb} \left( \mathrm{G}_F(\omega^k, x^*) - \mathrm{G}_F(x_g^k, x^*) - \langle \nabla F(x_g^k) - \nabla F(x^*) - \nu x_g^k + \nu x^*, \omega^k - x_g^k \rangle \right).$$

Using $\theta$ definition together with (14), (18) and (30) gives

$$\left( \frac{1}{\theta} + \frac{\beta}{2} \right) \mathbb{E}\left[ \|y^{k+1} - y^*\|^2 \right] + \frac{\beta}{2\sigma_2} \mathbb{E}\left[ \|y_f^{k+1} - y^*\|^2 \right] + \frac{1}{\gamma} \|\hat{z}^{k+1} - z^*\|^2 + \frac{4}{3\gamma} \|m^{k+1}\|_{\mathbf{P}}^2$$

$$\leq \left( \frac{1}{\gamma} - \delta \right) \|\hat{z}^k - z^*\|^2 + \left( 1 - (4\chi)^{-1} + \frac{3\gamma\delta}{2} \right) \frac{4}{3\gamma} \|m^k\|_{\mathbf{P}}^2 + \frac{1}{\theta} \|y^k - y^*\|^2$$

$$+ \frac{\beta(1 - \sigma_2/2)}{2\sigma_2} \|y_f^k - y^*\|^2 + \frac{\nu^{-1}(1 - \sigma_2/2)}{\sigma_2} \|y_f^k + z_f^k - (y^* + z^*)\|^2$$

$$- \frac{\nu^{-1}}{\sigma_2} \mathbb{E}\left[ \|y_f^{k+1} + z_f^{k+1} - (y^* + z^*)\|^2 \right]$$

$$+ \left( 2\gamma\delta^2 - \delta \right) \|z_g^k - z^k\|^2 - 2\mathbb{E}\left[ \langle x^{k+1} - x^*, y^{k+1} - y^* \rangle \right] + \mathrm{G}_F(x_g^k, x^*)$$

$$+ \frac{\overline{L}}{Lb} \left( \mathrm{G}_F(\omega^k, x^*) - \mathrm{G}_F(x_g^k, x^*) - \langle \nabla F(x_g^k) - \nabla F(x^*) - \nu x_g^k + \nu x^*, \omega^k - x_g^k \rangle \right).$$

Using $\gamma$ definition (28) and $\delta$ definition (27) we get

$$\left( \frac{1}{\theta} + \frac{\beta}{2} \right) \mathbb{E}\left[ \|y^{k+1} - y^*\|^2 \right] + \frac{\beta}{2\sigma_2} \mathbb{E}\left[ \|y_f^{k+1} - y^*\|^2 \right] + \frac{1}{\gamma} \|\hat{z}^{k+1} - z^*\|^2 + \frac{4}{3\gamma} \|m^{k+1}\|_{\mathbf{P}}^2$$

$$\leq \left( \frac{1}{\gamma} - \delta \right) \|\hat{z}^k - z^*\|^2 + \left( 1 - (8\chi)^{-1} \right) \frac{4}{3\gamma} \|m^k\|_{\mathbf{P}}^2 + \frac{1}{\theta} \|y^k - y^*\|^2$$

$$+ \frac{\beta(1 - \sigma_2/2)}{2\sigma_2} \|y_f^k - y^*\|^2 + \frac{\nu^{-1}(1 - \sigma_2/2)}{\sigma_2} \|y_f^k + z_f^k - (y^* + z^*)\|^2$$

$$- \frac{\nu^{-1}}{\sigma_2} \mathbb{E}\left[ \|y_f^{k+1} + z_f^{k+1} - (y^* + z^*)\|^2 \right]$$

$$- 2\mathbb{E}\left[ \langle x^{k+1} - x^*, y^{k+1} - y^* \rangle \right] + \mathrm{G}_F(x_g^k, x^*)$$

$$+ \frac{\overline{L}}{Lb} \left( \mathrm{G}_F(\omega^k, x^*) - \mathrm{G}_F(x_g^k, x^*) - \langle \nabla F(x_g^k) - \nabla F(x^*) - \nu x_g^k + \nu x^*, \omega^k - x_g^k \rangle \right).$$

After rearranging and using $\Psi_{yz}^k$ definition (31) we get

$$\mathbb{E}\left[ \Psi_{yz}^{k+1} \right] \leq \max \left\{ (1 + \theta\beta/2)^{-1}, (1 - \gamma\delta), (1 - \sigma_2/2), (1 - (8\chi)^{-1}) \right\} \Psi_{yz}^k$$

$$- 2\mathbb{E}\left[ \langle x^{k+1} - x^*, y^{k+1} - y^* \rangle \right] + \mathrm{G}_F(x_g^k, x^*)$$

$$+ \frac{\overline{L}}{Lb} \left( \mathrm{G}_F(\omega^k, x^*) - \mathrm{G}_F(x_g^k, x^*) - \langle \nabla F(x_g^k) - \nabla F(x^*) - \nu x_g^k + \nu x^*, \omega^k - x_g^k \rangle \right)$$

$$\leq \left( 1 - \frac{\sqrt{\mu}}{32\chi\sqrt{L}} \right) \Psi_{yz}^k$$

$$- 2\mathbb{E}\left[ \langle x^{k+1} - x^*, y^{k+1} - y^* \rangle \right] + \mathrm{G}_F(x_g^k, x^*)$$

$$+ \frac{\overline{L}}{Lb} \left( \mathrm{G}_F(\omega^k, x^*) - \mathrm{G}_F(x_g^k, x^*) - \langle \nabla F(x_g^k) - \nabla F(x^*) - \nu x_g^k + \nu x^*, \omega^k - x_g^k \rangle \right).$$

$$\square$$

**Lemma B.10.** *Let $\lambda$ be defined as follows:*

$$\lambda = \frac{n}{b} \left( \frac{1}{2} + \frac{\overline{L}}{Lb\tau_1} \right). \tag{33}$$

Let $p_1$ be defined as follows:

$$p_1 = \frac{1}{2\lambda}. \tag{34}$$

Let $p_2$ be defined as follows:

$$p_2 = \frac{\overline{L}}{\lambda L b \tau_1}. \tag{35}$$

Then the following inequality holds:

$$\mathbb{E}\left[\Psi_x^k + \Psi_{yz}^k + \lambda \mathrm{G}_F(\omega^{k+1}, x^*)\right]$$
$$\leq \left(1 - \frac{1}{32} \min\left\{\frac{b}{n}, b\sqrt{\frac{\mu}{nL}}, \frac{b^2 L}{n\overline{L}}\sqrt{\frac{\mu}{L}}, \frac{\sqrt{\mu}}{\chi\sqrt{L}}\right\}\right)(\Psi_x^0 + \Psi_{yz}^0 + \lambda \mathrm{G}_F(\omega^k, x^*)). \tag{36}$$

*Proof.* Combining (16) and (32) gives

$$\mathbb{E}\left[\Psi_x^{k+1} + \Psi_{yz}^{k+1}\right] \leq \left(1 - \frac{1}{20}\min\left\{\sqrt{\frac{\mu}{L}}, b\sqrt{\frac{\mu}{nL}}\right\}\right)\Psi_x^k + \left(1 - \frac{\sqrt{\mu}}{32\chi\sqrt{L}}\right)\Psi_{yz}^k$$
$$- \frac{\overline{L}}{Lb\tau_1}\mathrm{G}_F(x_g^k, x^*) + \frac{\overline{L}}{Lb\tau_1}\mathrm{G}_F(\omega^k, x^*) - \frac{1}{2}\mathrm{G}_F(x_f^k, x^*)$$
$$\leq \left(1 - \frac{1}{32}\min\left\{b\sqrt{\frac{\mu}{nL}}, \frac{\sqrt{\mu}}{\chi\sqrt{L}}\right\}\right)(\Psi_x^k + \Psi_{yz}^k)$$
$$- \frac{\overline{L}}{Lb\tau_1}\mathrm{G}_F(x_g^k, x^*) + \frac{\overline{L}}{Lb\tau_1}\mathrm{G}_F(\omega^k, x^*) - \frac{1}{2}\mathrm{G}_F(x_f^k, x^*). \tag{37}$$

Using (9) we get the following inequality:

$$\mathbb{E}\left[\mathrm{G}_F(\omega^{k+1}, x^*)\right] \leq p_1 \mathrm{G}_F(x_f^k, x^*) + p_2 \mathrm{G}_F(x_g^k, x^*) + (1 - p_1 - p_2)\mathrm{G}_F(\omega^k, x^*). \tag{38}$$

Multiplying (38) on $\lambda$ and combining with (37) we get

$$\mathbb{E}\left[\Psi_x^{k+1} + \Psi_{yz}^{k+1} + \lambda \mathrm{G}_F(\omega^{k+1}, x^*)\right]$$
$$\leq \left(1 - \frac{1}{32}\min\left\{b\sqrt{\frac{\mu}{nL}}, \frac{\sqrt{\mu}}{\chi\sqrt{L}}\right\}\right)(\Psi_x^k + \Psi_{yz}^k) + \lambda(1 - p_1)\mathrm{G}_F(\omega^k, x^*).$$

Estimating $p_1$, using $\tau_1$ and $\tau_0$ definitions (10), (11)

$$p_1 = \frac{b}{n}\left(2\left(\frac{1}{2} + \frac{\overline{L}}{Lb\tau_1}\right)\right)^{-1} = \frac{b}{n}\left(1 + \frac{2\overline{L}}{Lb\tau_1}\right)^{-1}$$

$$\geq \frac{b}{2n}\min\left\{1, \left(\frac{2\overline{L}}{Lb\tau_1}\right)^{-1}\right\} = \min\left\{\frac{b}{2n}, \frac{b^2 L \tau_1}{4n\overline{L}}\right\}$$

$$\geq \min\left\{\frac{b}{2n}, \frac{b^2 L \tau_2}{10n\overline{L}}\right\} = \min\left\{\frac{b}{2n}, \frac{b^2 L}{10n\overline{L}}\min\left\{\frac{1}{2}, \max\left\{1, \frac{\sqrt{n}}{b}\right\}\sqrt{\frac{\mu}{L}}\right\}\right\}$$

$$\geq \min\left\{\frac{b}{2n}, \frac{b^2 L}{20n\overline{L}}, \frac{b^2 L}{10n\overline{L}}\max\left\{1, \frac{\sqrt{n}}{b}\right\}\sqrt{\frac{\mu}{L}}\right\} \geq \min\left\{\frac{b}{20n}, \frac{b^2 L}{10n\overline{L}}\sqrt{\frac{\mu}{L}}\right\}.$$

Therefore we conclude

$$\mathbb{E}\left[\Psi_x^{k+1} + \Psi_{yz}^{k+1} + \lambda \mathrm{G}_F(\omega^{k+1}, x^*)\right]$$
$$\leq \left(1 - \frac{1}{32}\min\left\{\frac{b}{n}, b\sqrt{\frac{\mu}{nL}}, \frac{b^2 L}{n\overline{L}}\sqrt{\frac{\mu}{L}}, \frac{\sqrt{\mu}}{\chi\sqrt{L}}\right\}\right)(\Psi_x^k + \Psi_{yz}^k + \lambda \mathrm{G}_F(\omega^k, x^*)).$$

This implies

$$\mathbb{E}\left[\Psi_x^k + \Psi_{yz}^k + \lambda \mathrm{G}_F(\omega^k, x^*)\right]$$
$$\leq \left(1 - \frac{1}{32}\min\left\{\frac{b}{n}, b\sqrt{\frac{\mu}{nL}}, \frac{b^2 L}{n\overline{L}}\sqrt{\frac{\mu}{L}}, \frac{\sqrt{\mu}}{\chi\sqrt{L}}\right\}\right)^k (\Psi_x^0 + \Psi_{yz}^0 + \lambda \mathrm{G}_F(x^0, x^*)).$$

Using $\Psi_x^k$ definition (15) we get

$$\mathbb{E}\left[\|x^k - x^*\|^2\right] \le \eta \mathbb{E}\left[\Psi_x^k\right] \le \eta \mathbb{E}\left[\Psi_x^k + \Psi_{yz}^k + \lambda \mathrm{G}_F(\omega^k, x^*)\right]$$

$$\le \left(1 - \frac{1}{32}\min\left\{\frac{b}{n}, b\sqrt{\frac{\mu}{nL}}, \frac{b^2 L}{n\overline{L}}\sqrt{\frac{\mu}{L}}, \frac{\sqrt{\mu}}{\chi\sqrt{L}}\right\}\right)^k \eta(\Psi_x^0 + \Psi_{yz}^0 + \lambda \mathrm{G}_F(\omega^0, x^*)).$$

Choosing $C = \eta(\Psi_x^0 + \Psi_{yz}^0 + \lambda \mathrm{G}_F(\omega^k, x^*))$ and using the number of iterations

$$k = 32\max\left\{\frac{n}{b}, \frac{\sqrt{n}}{b}\sqrt{\frac{L}{\mu}}, \frac{n\overline{L}}{b^2 L}\sqrt{\frac{L}{\mu}}, \chi\sqrt{\frac{L}{\mu}}\right\}\log\frac{C}{\varepsilon}$$

$$= \mathcal{O}\left(\max\left\{\frac{n}{b}, \frac{\sqrt{n}}{b}\sqrt{\frac{L}{\mu}}, \frac{n\overline{L}}{b^2 L}\sqrt{\frac{L}{\mu}}, \chi\sqrt{\frac{L}{\mu}}\right\}\log\frac{1}{\varepsilon}\right)$$

we get

$$\|x^k - x^*\|^2 \le \epsilon.$$

Therefore the number of iterations of Algorithm (1) is bounded by

$$k = \mathcal{O}\left(\left(\frac{n}{b} + \frac{\sqrt{n}}{b}\sqrt{\frac{L}{\mu}} + \frac{n\overline{L}}{b^2 L}\sqrt{\frac{L}{\mu}} + \chi\sqrt{\frac{L}{\mu}}\right)\log\frac{1}{\epsilon}\right),$$

which concludes the proof. $\square$

Let's prove the Corollary 3.3.

*Proof.* The choice of the number of communication iterations $\sim \chi$ per algorithm iteration and a specific choice of $b = \max\{\sqrt{n\overline{L}/L}, n\sqrt{\mu/L}\}$ provides the following upper bound on the number of algorithm iterations:

$$N = \mathcal{O}\left(\sqrt{\frac{L}{\mu}}\log\frac{1}{\epsilon}\right).$$

From this, it immediately follows that the upper bound on the number of communications is as follows:

$$\mathcal{O}\left(\chi\sqrt{\frac{L}{\mu}}\log\frac{1}{\epsilon}\right).$$

Now, let's estimate the number of oracle calls at each node. It is not difficult to show the following upper bound:

$$Nb = \mathcal{O}\left(\left(n + \sqrt{n}\sqrt{\frac{L}{\mu}} + b\sqrt{\frac{L}{\mu}} + \frac{n\overline{L}}{bL}\sqrt{\frac{L}{\mu}}\right)\log\frac{1}{\epsilon}\right) = \mathcal{O}\left(\left(n + \sqrt{n}\sqrt{\frac{\overline{L}}{\mu}}\right)\log\frac{1}{\epsilon}\right),$$

which completes the proof.

$\square$

## C  PROOF OF THEOREM 4.3

The high-level concept underlying lower bounds in decentralized optimization involves creating a decentralized counterexample problem, where information exchange between two vertex clusters is slow. More specifically, the vertices in the counterexample are divided into three types: the first type can potentially "transfer" the gradient from even positions to the next, introducing a new dimension, the second type can do so from odd positions, the third type does nothing. We take a "bad" function for the corresponding optimization problem and divide it by the corresponding node types in such a way that different clusters contain

components of the "bad" function that can approach the solution only after "communicating" with nodes from another cluster.

As our graph counterexample, we will use the graph from Metelev et al. (2024) because it allows us to obtain a lower bound not only in the setting of "changing graphs" but also in the setting of "slowly changing graphs", which will be a good addition.

Let's define $T_{a,b}$ as a graph consisting of two "stars" with sizes $a+1$ and $b+1$, whose centers are connected to an isolated vertex. In total, the graph will have $a+b+3$ vertices.

Let's say the left part of the graph $\mathcal{P}_1$ is the set of $a+1$ vertices of the first star, and the right part $\mathcal{P}_2$ is correspondingly the set of $b+1$ vertices of the second star. The middle vertex $v_m$ is the vertex connected to the centers $v_l$ and $v_r$ of the left and right stars, respectively.

If $v \in \mathcal{P}_1$, we define the "hop to the right" operation as follows: remove the edge $(v_l, v_m)$ and add the edge $(v, v_r)$. As a result, $v_m$ ceases to be the middle vertex, being replaced by the vertex $v$. The operation "hop to the left" is defined in the same way.

Now, let's describe the sequence of graphs that will make up the changing network. The first graph will be of the form $T_{0,m-3}$, followed by a series of "hops to the left", which increase the left part $\mathcal{P}_1$ of the graph and decrease the right. This will continue until the graph $T_{m-3,0}$ appears. After this, a series of "hops to the right" occur until the network returns to its original form. Then, the cycle repeats.

**Lemma C.1.** *For this sequence of graphs, there exists a corresponding sequence of positive weights $(A_k)_{k=0}^{\infty}$ and a sequence of Laplacian matrices $(W(k))_{k=0}^{\infty}$ for these weighted graphs, such that it satisfies 2.5 with*

$$\chi \leq 8m. \tag{39}$$

*Proof.* This is a direct consequence of Lemma 8 from Metelev et al. (2024). □

Note that vertices $v_l$ and $v_r$ in the process of changing the network are always on the left and right parts, respectively. Denote by $\{g_i\}_{i=1}^m : y \in \ell_2 \to \mathbb{R}$ the set of auxiliary functions corresponding to the vertices:

$$g_i(y) = \begin{cases} \frac{\mu}{2}\|y\|^2 + \frac{(L-\mu)}{4}\left[(y_1 - 1)^2 + \sum_{k=1}^{\infty}(y_{2k} - y_{2k+1})^2\right], & i = v_l, \\ \frac{\mu}{2}\|y\|^2 + \frac{(L-\mu)}{4}\sum_{k=1}^{\infty}(y_{2k-1} - y_{2k})^2, & i = v_r, \\ \frac{\mu}{2(m-2)}\|y\|^2, & i \notin \{v_l, v_r\}. \end{cases} \tag{40}$$

Let's describe the local functions on the nodes: let $x \in \ell_2^n$, then define $f_{ij}(x) = g_i(x_j)$, where $x_j \in \ell_2$. Accordingly, it turns out that $f_{ij} : x \in \ell_2^n \to \mathbb{R}$, but its gradient affects only the $j$th subspace of $\ell_2^n$, in which $x_k = 0$ for $k \neq j$. Hence, $F_i(x) = \frac{1}{n}\sum_{j=1}^n g_i(x_j)$.

Such a structure allows achieving that the "transfer" of the gradient to the next dimension in each subspace occurs once every $\Omega(m) = \Omega(\chi)$ communication iterations.

The solution to this optimization problem will be the vector $(x^*, \ldots, x^*) \in \ell_2^n$, $x^* = (1, q, q^2, \ldots) \in \ell_2$, $q = \frac{\sqrt{\frac{2}{3}L/\mu + \frac{1}{3}} - 1}{\sqrt{\frac{2}{3}L/\mu + 1} + \frac{1}{3}}$ .

Let $(e_1, e_2, \ldots, e_n)$ be sets of vectors that form a basis in the space $\ell_2^n$. Let $x_{ij}$ denote the coordinates along a set of vectors $e_j$ on the variable on the $i$th node.

Following the ideas of Hendrikx et al. (2021), consider the expression

$$A \triangleq \sum_{i=1}^m \sum_{j=1}^n \|x_{ij} - x^*\|^2.$$

Let's define the quantities $k_j = \min\{k \in \mathbb{N}_0 | \forall l \geq k, \forall i \in \{1, \ldots, m\} \to x_{ijl} = 0\}$. Using this definition and the convexity of $q^{2x}$ we get

$$A \geq \frac{m}{1-q^2}\sum_{i=1}^n q^{2k_j} \geq \frac{nm}{1-q^2}q^{\frac{2}{n}\sum_{j=1}^n k_j}. \tag{41}$$

Let $T_c$ and $T_s$ be the number of communication rounds and the number of oracle calls at node $v_l$, respectively. Between the network state $T_{0,m-3}$ and the next such state there are $2m - 6$ communication iterations, during which two "transfers" of the gradient from an odd position to an even one cannot occur. Therefore we get

$$k_j \leq 1 + \frac{T_c}{m-3}. \tag{42}$$

Note that each $j$ corresponds to at least $\lceil k_j/2 \rceil$ oracle calls to the function $f_{ij}$ for $i = v_l$, hence we get

$$\sum_{j=1}^n k_j \leq 2T_s. \tag{43}$$

Using (41), (42) and (43) we get

$$A \geq \frac{nm}{1-q^2} \max\left\{\left(1 - \frac{2}{\sqrt{\frac{2}{3}L/\mu + \frac{1}{3}} + 1}\right)^{2+2t_c/(m-3)}, \left(1 - \frac{2}{\sqrt{\frac{2}{3}L/\mu + \frac{1}{3}} + 1}\right)^{4t_s/n}\right\}. \tag{44}$$

Based on the form of the function we can conclude that $\kappa_s = \frac{nL}{\mu} = n\kappa_b$, then using $x^0 = 0$, $\|x_{ij}^0 - x_{ij}^*\|^2 = (1-q^2)^{-1}$ and (39) we get

$$\frac{1}{nm} \sum_{i=1}^m \sum_{j=1}^n \frac{\|x_{ij} - x^*\|^2}{\|x_{ij}^0 - x^*\|^2}$$

$$\geq \max\left\{\left(1 - \frac{2}{\sqrt{\frac{2}{3}\kappa_b + \frac{1}{3}} + 1}\right)^{2+16t_c/(\chi-24)}, \left(1 - \frac{2n}{\sqrt{n}\sqrt{\frac{2}{3}\kappa_s + n/3} + n}\right)^{4t_s/n}\right\},$$

which concludes the proof.

## D    PROOFS FOR ALGORITHM 2

Before we start, let us denote

$$\mathbf{M}(k) = (\mathbf{I}_m - \mathbf{W}(k)) \otimes \mathbf{I}_d \tag{45}$$

and

$$\rho = \frac{1}{\chi} \tag{46}$$

for the convenient analysis. Moreover, we need to introduce some definitions as

$$\bar{x}^k = \frac{1}{m}(\mathbf{1}_m^\top \otimes \mathbf{I}_d)x^k,$$

$$\bar{v}^k = \frac{1}{m}(\mathbf{1}_m^\top \otimes \mathbf{I}_d)v^k,$$

$$S^k = (S_1^k, \ldots, S_m^k),$$

$$\nabla_{S^k} F(x^k) = \left(\nabla_{S_1^k} F_1(x_1^k), \ldots, \nabla_{S_m^k} F_m(x_m^k)\right) \in \mathbb{R}^{md},$$

$$\nabla_{S_i^k} F_i(x_i^k) = \frac{1}{b} \sum_{j \in S_i^k} \nabla f_{ij}(x_i^k)$$

Also we need to formulate some useful propositions:

**Proposition D.1.** *If $\bar{v}^0 = \frac{1}{m}(\mathbf{1}_m^\top \otimes \mathbf{I}_d)y^0$, then for any $k \geq 1$, according to Algorithm 2, we get*

$$\bar{v}^k = \frac{1}{m}(\mathbf{1}_m^\top \otimes \mathbf{I}_d)y^k, \tag{47}$$

*and*

$$\bar{x}^{k+1} = \bar{x}^k - \frac{\eta}{m}(\mathbf{1}_m^\top \otimes \mathbf{I}_d)y^k. \tag{48}$$

*Proof.* We prove it using the induction. For $k = 0$ it is trivial because of start point. Now suppose that at the $k$-th iteration, the relation (47) is true:

$$\bar{v}^k = \frac{1}{m}(\mathbf{1}_m^\top \otimes \mathbf{I}_d)y^k.$$

Hence, at the $(k+1)$-th iteration, we have

$$\begin{aligned}
\bar{v}^{k+1} &= \frac{1}{m}(\mathbf{1}_m^\top \otimes \mathbf{I}_d)v^{k+1} \\
&= \frac{1}{m}(\mathbf{1}_m^\top \otimes \mathbf{I}_d)v^k + \frac{1}{m}(\mathbf{1}_m^\top \otimes \mathbf{I}_d)(\mathbf{M}(k) - \mathbf{I}_{md})v^k + \frac{1}{m}(\mathbf{1}_m^\top \otimes \mathbf{I}_d)\left(y^{k+1} - y^k\right) \\
&= \frac{1}{m}(\mathbf{1}_m^\top \otimes \mathbf{I}_d)v^k + \frac{1}{m}(\mathbf{1}_m^\top \otimes \mathbf{I}_d)\left(y^{k+1} - y^k\right) \\
&= \frac{1}{m}(\mathbf{1}_m^\top \otimes \mathbf{I}_d)y^{k+1},
\end{aligned}$$

where the third line follows from Assumption 2.5:

$$(\mathbf{1}_m^\top \otimes \mathbf{I}_d)(\mathbf{M}(k) - \mathbf{I}_{md}) = -(\mathbf{1}_m^\top \otimes \mathbf{I}_d)(\mathbf{W}(k) \otimes \mathbf{I}_d) = -(\mathbf{1}_m^\top \mathbf{W}(k) \otimes \mathbf{I}_d) = 0.$$

Thus, we complete the proof of (47). For (48),

$$\begin{aligned}
\bar{x}^{k+1} &= \bar{x}^k + \frac{1}{m}(\mathbf{1}_m^\top \otimes \mathbf{I}_d)(\mathbf{M}(k) - \mathbf{I}_{md})x^k - \frac{\eta}{m}(\mathbf{1}_m^\top \otimes \mathbf{I}_d)v^k \\
&= \bar{x}^k - \eta\bar{v}^k = \bar{x}^k - \frac{\eta}{m}(\mathbf{1}_m^\top \otimes \mathbf{I}_d)y^k.
\end{aligned}$$

$\square$

**Proposition D.2.** *If $\mathbf{W}(k)$ satisfy Assumption 2.5 and $\mathbf{M}(k)$ is taken from (45), then $\forall x \in \mathbb{R}^{md}$, we have*

$$\|\mathbf{M}(k)x - \frac{1}{m}(\mathbf{1}_m \otimes \mathbf{I}_d)(\mathbf{1}_m^\top \otimes \mathbf{I}_d)x\|^2 \leq (1-\rho)\|x - \frac{1}{m}(\mathbf{1}_m \otimes \mathbf{I}_d)(\mathbf{1}_m^\top \otimes \mathbf{I}_d)x\|^2, \qquad (49)$$

*Proof.* Note that

$$\mathbf{M}(k)(\mathbf{1}_m \otimes \mathbf{I}_d) = ((\mathbf{I}_m - \mathbf{W}(k)) \otimes \mathbf{I}_d)(\mathbf{1}_m \otimes \mathbf{I}_d) = ((\mathbf{I}_m - \mathbf{W}(k))\mathbf{1}_m \otimes \mathbf{I}_d) = \mathbf{1}_m \otimes \mathbf{I}_d.$$

Therefore,

$$\|\mathbf{M}(k)x - \frac{1}{m}(\mathbf{1}_m \otimes \mathbf{I}_d)(\mathbf{1}_m^\top \otimes \mathbf{I}_d)x\|^2 = \|\mathbf{M}(k)\left(x - \frac{1}{m}(\mathbf{1}_m \otimes \mathbf{I}_d)(\mathbf{1}_m^\top \otimes \mathbf{I}_d)x\right)\|^2.$$

Decomposing $x - \frac{1}{m}(\mathbf{1}_m \otimes \mathbf{I}_d)(\mathbf{1}_m^\top \otimes \mathbf{I}_d)x$ by eigenvectors of $\mathbf{M}(k)$ and using that

$$\mathbf{1}_{md}^\top \left(\mathbf{I}_{md} - \frac{1}{m}(\mathbf{1}_m\mathbf{1}_m^\top \otimes \mathbf{I}_d)\right) = 0,$$

we claim the final result.

*Remark* D.3. The proposition above is equivalent to

$$\|\mathbf{M}(k)x^k - (\mathbf{1}_m \otimes \mathbf{I}_d)\bar{x}^k\|^2 \leq (1-\rho)\|x^k - (\mathbf{1}_m \otimes \mathbf{I}_d)\bar{x}^k\|^2.$$

$\square$

### D.1 DESCENT LEMMA

**Lemma D.4. *(Descent lemma)*** *Let Assumption 2.2 and Assumption 2.5 hold. Then, after $k$ iterations of Algorithm 2, we get*

$$\mathbb{E}F(\bar{x}^{k+1}) \leq \mathbb{E}F(\bar{x}^k) - \frac{\eta}{2}\mathbb{E}\|\nabla F(\bar{x}^k)\|^2 + \frac{\eta}{m}\mathbb{E}\|\nabla F(x^k) - y^k\|^2 + \frac{\eta L^2}{m}\mathbb{E}\|x^k - (\mathbf{1}_m \otimes \mathbf{I}_d)\bar{x}^k\|^2$$

$$- \left(\frac{\eta}{2} - \frac{\eta^2 L}{2}\right)\mathbb{E}\|\bar{v}^k\|^2. \qquad (50)$$

*Proof.* Starting with $L$-smoothness:

$$F(\bar{x}^{k+1}) \leq F(\bar{x}^k) - \eta \left\langle \bar{v}^k, \nabla F(\bar{x}^k) \right\rangle + \frac{\eta^2 L}{2} \|\bar{v}^k\|^2$$

$$= F(\bar{x}^k) - \frac{\eta}{2}\|\nabla F(\bar{x}^k)\|^2 - \frac{\eta}{2}\|\bar{v}^k\|^2 + \frac{\eta}{2}\|\nabla F(\bar{x}^k) - \bar{v}^k\|^2 + \frac{\eta^2 L}{2}\|\bar{v}^k\|^2$$

$$= F(\bar{x}^k) - \frac{\eta}{2}\|\nabla F(\bar{x}^k)\|^2 + \frac{\eta}{2}\|\nabla F(\bar{x}^k) - \bar{v}^k\|^2 - \left(\frac{\eta}{2} - \frac{\eta^2 L}{2}\right)\|\bar{v}^k\|^2$$

$$\leq F(\bar{x}^k) - \frac{\eta}{2}\|\nabla F(\bar{x}^k)\|^2 + \frac{\eta}{2}\|\nabla F(\bar{x}^k) - \frac{1}{m}(\mathbf{1}_m^\top \otimes \mathbf{I}_d)y^k\|^2 - \left(\frac{\eta}{2} - \frac{\eta^2 L}{2}\right)\|\bar{v}^k\|^2$$

$$\leq F(\bar{x}^k) - \frac{\eta}{2}\|\nabla F(\bar{x}^k)\|^2 + \frac{\eta}{2m}\|(\mathbf{1}_m \otimes \mathbf{I}_d)\nabla F(\bar{x}^k) - \nabla F(x^k) + \nabla F(x^k) - y^k\|^2$$

$$- \left(\frac{\eta}{2} - \frac{\eta^2 L}{2}\right)\|\bar{v}^k\|^2$$

$$\leq F(\bar{x}^k) - \frac{\eta}{2}\|\nabla F(\bar{x}^k)\|^2 + \frac{\eta}{m}\|\nabla F(x^k) - y^k\|^2 + \frac{\eta L^2}{m}\|x^k - (\mathbf{1}_m \otimes \mathbf{I}_d)\bar{x}^k\|^2$$

$$- \left(\frac{\eta}{2} - \frac{\eta^2 L}{2}\right)\|\bar{v}^k\|^2, \tag{51}$$

where in the last inequality we use $(a+b)^2 \leq 2a^2 + 2b^2$. Taking the expectation, we claim the final result. $\qquad\square$

## D.2 Auxiliary lemmas

**Lemma D.5.** *Let Assumption 2.3 holds. Hence, after $k$ iterations the following is fulfilled:*

$$\mathbb{E}\|\nabla F(x^{k+1}) - y^{k+1}\|^2 \leq (1-p)\mathbb{E}\|\nabla F(x^k) - y^k\|^2 + \frac{(1-p)\hat{L}^2}{b}\mathbb{E}\|x^{k+1} - x^k\|^2.$$

*Proof.*

$$\mathbb{E}\|\nabla F(x^{k+1}) - y^{k+1}\|^2 = p\mathbb{E}\|\nabla F(x^{k+1}) - \nabla F(x^{k+1})\|^2$$

$$+ (1-p)\mathbb{E}\|\nabla F(x^{k+1}) - y^k - \nabla_{S^k} F(x^{k+1}) + \nabla_{S^k} F(x^k)\|^2$$

$$= (1-p)\mathbb{E}\|\nabla F(x^{k+1}) - \nabla F(x^k) + \nabla F(x^k) - y^k - \nabla_{S^k} F(x^{k+1}) + \nabla_{S^k} F(x^k)\|^2$$

$$= (1-p)\mathbb{E}\|\nabla F(x^{k+1}) - \nabla F(x^k) - \nabla_{S^k} F(x^{k+1}) + \nabla_{S^k} F(x^k)\|^2$$

$$+ (1-p)\mathbb{E}\|\nabla F(x^k) - y^k\|^2, \tag{52}$$

Rewriting $\nabla_{S^k} F(x)$ as claimed before, using that $\mathbb{E}\|X - \mathbb{E}X\|^2 \leq \mathbb{E}\|X\|^2$, clarifying that indices in one batch are chosen independently and using the $\hat{L}$-average smoothness, one can obtain

$$\mathbb{E}\|\nabla F(x^{k+1}) - y^{k+1}\|^2 \leq (1-p)\mathbb{E}\|\nabla F(x^k) - y^k\|^2 + \frac{(1-p)\hat{L}^2}{b}\mathbb{E}\|x^{k+1} - x^k\|^2, \tag{53}$$

what ends the proof. $\qquad\square$

*Remark* D.6. The proof is similar to the proof of Lemma 3 in Li et al. (2021), but we write it for each node in the same time.

Now we need to bound some extra terms for our Lyapunov's function. We use the next notation

$$\Omega_1^k = \mathbb{E}\|x^k - (\mathbf{1}_m \otimes \mathbf{I}_d)\bar{x}^k\|^2,$$

$$\Omega_2^k = \mathbb{E}\|v^k - (\mathbf{1}_m \otimes \mathbf{I}_d)\bar{v}^k\|^2.$$

**Lemma D.7.** *Let Assumption 2.5 holds. Therefore, for the Algorithm 2, we have*

$$\Omega_1^{k+1} \leq \left(1 - \frac{\rho}{2}\right)\Omega_1^k + \frac{3\eta^2}{\rho}\Omega_2^k,$$

$$\Omega_2^{k+1} \leq \left(1 - \frac{\rho}{2}\right)\Omega_2^k + \frac{3}{\rho}\mathbb{E}\|y^{k+1} - y^k\|^2.$$

*Proof.* Substituting the iteration of Algorithm 2 into $\Omega_1^{k+1}$, we get

$$
\begin{aligned}
&\|x^{k+1} - (\mathbf{1}_m \otimes \mathbf{I}_d)\bar{x}^{k+1}\|^2 \\
&= \|\mathbf{M}(k)x^k - \eta v^k - (\mathbf{1}_m \otimes \mathbf{I}_d)\bar{x}^k + (\mathbf{1}_m \otimes \mathbf{I}_d)\eta\bar{v}^k\|^2 \\
&\le (1+\beta)(1-\rho)\|x^k - (\mathbf{1}_m \otimes \mathbf{I}_d)\bar{x}^k\|^2 + \left(1 + \frac{1}{\beta}\right)\eta^2\|v^k - (\mathbf{1}_m \otimes \mathbf{I}_d)\bar{v}^k\|^2 \\
&\le \left(1 - \frac{\rho}{2}\right)\|x^k - (\mathbf{1}_m \otimes \mathbf{I}_d)\bar{x}^k\|^2 + \left(1 + \frac{2}{\rho}\right)\eta^2\|v^k - (\mathbf{1}_m \otimes \mathbf{I}_d)\bar{v}^k\|^2 \\
&\le \left(1 - \frac{\rho}{2}\right)\|x^k - (\mathbf{1}_m \otimes \mathbf{I}_d)\bar{x}^k\|^2 + \frac{3\eta^2}{\rho}\|v^k - (\mathbf{1}_m \otimes \mathbf{I}_d)\bar{v}^k\|^2,
\end{aligned}
\tag{54}
$$

where we choose $\beta = \frac{\rho}{2}$. For $\Omega_2^{k+1}$ respectively

$$
\begin{aligned}
\|v^{k+1} - (\mathbf{1}_m \otimes \mathbf{I}_d)\bar{v}^{k+1}\|^2 &= \|v^{k+1} - (\mathbf{1}_m \otimes \mathbf{I}_d)\bar{v}^k + (\mathbf{1}_m \otimes \mathbf{I}_d)\bar{v}^k - (\mathbf{1}_m \otimes \mathbf{I}_d)\bar{v}^{k+1}\|^2 \\
&= \|v^{k+1} - (\mathbf{1}_m \otimes \mathbf{I}_d)\bar{v}^k\|^2 - m\|\bar{v}^{k+1} - \bar{v}^k\|^2 \\
&\le \|v^{k+1} - (\mathbf{1}_m \otimes \mathbf{I}_d)\bar{v}^k\|^2.
\end{aligned}
$$

Thus by the update rule of Algorithm 2, one can obtain

$$
\begin{aligned}
\|v^{k+1} - (\mathbf{1}_m \otimes \mathbf{I}_d)\bar{v}^{k+1}\|^2 &\le \|v^{k+1} - (\mathbf{1}_m \otimes \mathbf{I}_d)\bar{v}^k\|^2 \\
&= \|\mathbf{M}(k)v^k + y^{k+1} - y^k - (\mathbf{1}_m \otimes \mathbf{I}_d)\bar{v}^k\|^2 \\
&\le \left(1 - \frac{\rho}{2}\right)\|v^k - (\mathbf{1}_m \otimes \mathbf{I}_d)\bar{v}^k\|^2 + \left(1 + \frac{2}{\rho}\right)\|y^{k+1} - y^k\|^2 \\
&\le \left(1 - \frac{\rho}{2}\right)\|v^k - (\mathbf{1}_m \otimes \mathbf{I}_d)\bar{v}^k\|^2 + \frac{3}{\rho}\|y^{k+1} - y^k\|^2.
\end{aligned}
\tag{55}
$$

Taking the expectation in both bounds, we claim the final result. □

As a consequence of Lemma D.5 and Lemma D.7, we need to bound some redundant expressions.

**Lemma D.8.** *Let Assumptions 2.2, Assumption 2.3 and 2.5 hold. Then, after $k$ iterations of Algorithm 2, we get*

$$
\mathbb{E}\|y^{k+1} - y^k\|^2 \le (1+p)\hat{L}^2\mathbb{E}\|x^{k+1} - x^k\|^2 + 2p\mathbb{E}\|\nabla F(x^k) - y^k\|^2,
$$
$$
\mathbb{E}\|x^{k+1} - x^k\|^2 \le 2\widetilde{C}\mathbb{E}\|x^k - (\mathbf{1}_m \otimes \mathbf{I}_d)\bar{x}^k\|^2 + 2\eta^2\mathbb{E}\|v^k - (\mathbf{1}_m \otimes \mathbf{I}_d)\bar{v}^k\|^2 + 2\eta^2 m\mathbb{E}\|\bar{v}^k\|^2,
$$

*where $\widetilde{C} = \max_k \|\mathbf{M}(k) - \mathbf{I}_{md}\|^2 = \max_k \sigma_{\max}(\mathbf{M}(k) - \mathbf{I}_{md})^2 \le 4$.*

*Proof.* Start with substituting $y^{k+1}$:

$$
\begin{aligned}
\mathbb{E}\|y^{k+1} - y^k\|^2 &= p\mathbb{E}\|\nabla F(x^{k+1}) - y^k\|^2 + (1-p)\mathbb{E}\|\nabla_{S^k} F(x^{k+1}) - \nabla_{S^k} F(x^k)\|^2 \\
&= p\mathbb{E}\|\nabla F(x^{k+1}) - \nabla F(x^k) + \nabla F(x^k) - y^k\|^2 \\
&\quad + (1-p)\mathbb{E}\|\nabla_{S^k} F(x^{k+1}) - \nabla_{S^k} F(x^k)\|^2 \\
&\le p(1+\beta)L^2\mathbb{E}\|x^{k+1} - x^k\|^2 + p\left(1 + \frac{1}{\beta}\right)\mathbb{E}\|\nabla F(x^k) - y^k\|^2 \\
&\quad + (1-p)\mathbb{E}\|\nabla_{S^k} F(x^{k+1}) - \nabla_{S^k} F(x^k)\|^2.
\end{aligned}
\tag{56}
$$

Let us bound the last term in (56). We have

$$
\begin{aligned}
\mathbb{E}\|\nabla_{S^k} F(x^{k+1}) - \nabla_{S^k} F(x^k)\|^2 &= \mathbb{E}\sum_{i=1}^m \|\nabla_{S_i^k} F_i(x_i^{k+1}) - \nabla_{S_i^k} F_i(x_i^k)\|^2 \\
&= \mathbb{E}\sum_{i=1}^m \|\frac{1}{b}\sum_{\ell \in \{S_i^k\}} \nabla f_{i\ell}(x_i^{k+1}) - \nabla f_{i\ell}(x_i^k)\|^2
\end{aligned}
$$

$$= \mathbb{E} \sum_{i=1}^{m} \frac{1}{b^2} \| \sum_{\ell \in \{S_i^k\}} \nabla f_{i\ell}(x_i^{k+1}) - \nabla f_{i\ell}(x_i^k) \|^2$$

$$\leq \mathbb{E} \sum_{i=1}^{m} \frac{1}{b} \sum_{\ell \in \{S_i^k\}} \| \nabla f_{i\ell}(x_i^{k+1}) - \nabla f_{i\ell}(x_i^k) \|^2$$

$$\leq \mathbb{E} \sum_{i=1}^{m} \frac{\hat{L}^2}{b} \sum_{\ell \in \{S_i^k\}} \| x_i^{k+1} - x_i^k \|^2$$

$$= \mathbb{E} \sum_{i=1}^{m} \hat{L}^2 \| x_i^{k+1} - x_i^k \|^2$$

$$= \hat{L}^2 \mathbb{E} \| x^{k+1} - x^k \|^2. \tag{57}$$

Hence, substituting (57) into (56), choosing $\beta$ as 1 and using $L \leq \hat{L}$ (because of Jensen's inequality), one can obtain

$$\mathbb{E} \| y^{k+1} - y^k \|^2 \leq (1+p) \hat{L}^2 \mathbb{E} \| x^{k+1} - x^k \|^2 + 2p \mathbb{E} \| \nabla F(x^k) - y^k \|^2. \tag{58}$$

The second expression can be bounded in the following way:

$$\| x^{k+1} - x^k \|^2 = \| (\mathbf{M}(k) - \mathbf{I}_{md}) x^k - \eta v^k \|^2$$

$$= \| (\mathbf{M}(k) - \mathbf{I}_{md})(x^k - (\mathbf{1}_m \otimes \mathbf{I}_d) \bar{x}^k) - \eta v^k \|^2$$

$$\leq 2\widetilde{C} \| x^k - (\mathbf{1}_m \otimes \mathbf{I}_d) \bar{x}^k \|^2 + 2\eta^2 \| v^k \|^2$$

$$= 2\widetilde{C} \| x^k - (\mathbf{1}_m \otimes \mathbf{I}_d) \bar{x}^k \|^2 + 2\eta^2 \| v^k - (\mathbf{1}_m \otimes \mathbf{I}_d) \bar{v}^k \|^2 + 2\eta^2 m \| \bar{v}^k \|^2. \tag{59}$$

Taking the expectation, we claim the final result. $\qquad \square$

Now we denote some expressions from Lemma D.5 and Lemma D.8 as follows

$$\Delta^k = \mathbb{E} \| \nabla F(x^k) - y^k \|^2,$$

$$\Delta_x^k = \mathbb{E} \| x^{k+1} - x^k \|^2.$$

Consequently, substituting the bound of a first expression from Lemma D.8 in Lemma D.7, we get

$$\Omega_1^{k+1} \leq \left(1 - \frac{\rho}{2}\right) \Omega_1^k + \frac{3\eta^2}{\rho} \Omega_2^k,$$

$$\Omega_2^{k+1} \leq \left(1 - \frac{\rho}{2}\right) \Omega_2^k + \frac{3}{\rho}(2p\Delta^k + (1+p)\hat{L}^2 \Delta_x^k). \tag{60}$$

Moreover, we can write

$$\Delta^{k+1} \leq (1-p)\Delta^k + \frac{(1-p)\hat{L}^2}{b} \Delta_x^k,$$

$$\Delta_x^k \leq 2\widetilde{C}\Omega_1^k + 2\eta^2 \Omega_2^k + 2\eta^2 m \mathbb{E} \| \bar{v}^k \|^2.$$

### D.3    PROOF OF THEOREM 3.4

*Proof.* Rewriting the descent lemma in new notation, we have

$$\mathbb{E} F(\bar{x}^{k+1}) \leq \mathbb{E} F(\bar{x}^k) - \frac{\eta}{2} \mathbb{E} \| \nabla F(\bar{x}^k) \|^2 + \frac{\eta}{m} \Delta^k + \frac{\eta L^2}{m} \Omega_1^k - \left(\frac{\eta}{2} - \frac{\eta^2 L}{2}\right) \mathbb{E} \| \bar{v}^k \|^2.$$

Also we can construct a Lyapunov's function in the following way:

$$\Phi_k = \mathbb{E} F(\bar{x}^k) - F^* + C_0 \Delta^k + s_1 \Omega_1^k + s_2 \Omega_2^k. \tag{61}$$

Then, adding some terms to the left-hand side of descent lemma mentioned above, one can obtain

$$\Phi_{k+1} = \mathbb{E} F(\bar{x}^{k+1}) - F^* + C_0 \Delta^{k+1} + s_1 \Omega_1^{k+1} + s_2 \Omega_2^{k+1}$$

$$\leq \mathbb{E}F(\bar{x}^k) - F^* - \frac{\eta}{2}\mathbb{E}\|\nabla F(\bar{x}^k)\|^2 + \frac{\eta}{m}\Delta^k + \frac{\eta L^2}{m}\Omega_1^k - \left(\frac{\eta}{2} - \frac{\eta^2 L}{2}\right)\mathbb{E}\|\bar{v}^k\|^2$$

$$+ C_0\left((1-p)\Delta^k + \frac{(1-p)\hat{L}^2}{b}\Delta_x^k\right) + s_1\left(\left(1 - \frac{\rho}{2}\right)\Omega_1^k + \frac{3\eta^2}{\rho}\Omega_2^k\right)$$

$$+ s_2\left(\left(1 - \frac{\rho}{2}\right)\Omega_2^k + \frac{3}{\rho}(2p\Delta^k + (1+p)\hat{L}^2\Delta_x^k)\right).$$

Grouping the terms, we get

$$\Phi_{k+1} \leq \mathbb{E}F(\bar{x}^k) - F^* - \frac{\eta}{2}\mathbb{E}\|\nabla F(\bar{x}^k)\|^2 + \Delta^k\left((1-p)C_0 + \frac{\eta}{m} + \frac{6ps_2}{\rho}\right)$$

$$+ \Omega_1^k\left(\frac{\eta L^2}{m} + \left(1 - \frac{\rho}{2}\right)s_1\right) + \Omega_2^k\left(\frac{3\eta^2 s_1}{\rho} + \left(1 - \frac{\rho}{2}\right)s_2\right)$$

$$+ \Delta_x^k\left(\frac{(1-p)\hat{L}^2 C_0}{b} + \frac{3(1+p)\hat{L}^2 s_2}{\rho}\right) - \left(\frac{\eta}{2} - \frac{\eta^2 L}{2}\right)\mathbb{E}\|\bar{v}^k\|^2. \tag{62}$$

Hence, denoting

$$A = (1-p)C_0 + \frac{\eta}{m} + \frac{6ps_2}{\rho},$$

$$B = \frac{(1-p)\hat{L}^2 C_0}{b} + \frac{3(1+p)\hat{L}^2 s_2}{\rho},$$

$$C = \frac{\eta L^2}{m} + \left(1 - \frac{\rho}{2}\right)s_1,$$

$$D = \frac{3\eta^2 s_1}{\rho} + \left(1 - \frac{\rho}{2}\right)s_2,$$

and substituting these constants into (62), we get

$$\Phi_{k+1} \leq \mathbb{E}F(\bar{x}^k) - F^* - \frac{\eta}{2}\mathbb{E}\|\nabla F(\bar{x}^k)\|^2 + A\Delta^k + C\Omega_1^k + D\Omega_2^k + B\Delta_x^k$$

$$- \left(\frac{\eta}{2} - \frac{\eta^2 L}{2}\right)\mathbb{E}\|\bar{v}^k\|^2. \tag{63}$$

Using the definition of $\Delta_x^k$ and Lemma D.8 in (63), we finally have

$$\Phi_{k+1} \leq \mathbb{E}F(\bar{x}^k) - F^* - \frac{\eta}{2}\mathbb{E}\|\nabla F(\bar{x}^k)\|^2 + A\Delta^k + (C + 2\widetilde{C}B)\Omega_1^k + (D + 2\eta^2 B)\Omega_2^k$$

$$- \left(\frac{\eta}{2} - \frac{\eta^2 L}{2} - 2\eta^2 nB\right)\mathbb{E}\|\bar{v}^k\|^2$$

$$= \mathbb{E}F(\bar{x}^k) - F^* + s_1\Omega_1^k + s_2\Omega_2^k + A\Delta^k - \frac{\eta}{2}\mathbb{E}\|\nabla F(\bar{x}^k)\|^2$$

$$+ (C + 2\widetilde{C}B - s_1)\Omega_1^k + (D + 2\eta^2 B - s_2)\Omega_2^k - \left(\frac{\eta}{2} - \frac{\eta^2 L}{2} - 2\eta^2 mB\right)\mathbb{E}\|\bar{v}^k\|^2. \tag{64}$$

Looking at the form of the descent lemma, we want to require the following:

1. $C_0 = A$.

2. $\frac{\eta}{2} - \frac{\eta^2 L}{2} - 2\eta^2 mB \geq 0$.

3. $C + 2\widetilde{C}B - s_1 \leq 0$.

4. $D + 2\eta^2 B - s_2 \leq 0$.

Before we start to solve this system relative to $\eta$, we assume the following form of constants $s_1$ and $s_2$:

$$s_1 = \frac{c_1(\rho, p, b)\hat{L}^2}{mL},$$

$$s_2 = \frac{c_2(\rho, p, b)L}{m\hat{L}^2}. \tag{65}$$

**First part**

From the first requirement we get

$$C_0 = \frac{\eta}{mp} + \frac{6s_2}{\rho}. \tag{66}$$

**Second part**

From the second requirement:

$$\frac{\eta}{2} - \frac{\eta^2 L}{2} - 2\eta^2 m \left( \frac{(1-p)\hat{L}^2 C_0}{b} + \frac{3(1+p)\hat{L}^2 s_2}{\rho} \right) \geq 0. \tag{67}$$

After substituting $C_0$ into (67), we have

$$\frac{\eta}{2} - \frac{\eta^2 L}{2} - \frac{2\eta^3(1-p)\hat{L}^2}{bp} - \frac{12\eta^2 m(1-p)\hat{L}^2 s_2}{b\rho} - \frac{6\eta^2 m(1+p)\hat{L}^2 s_2}{\rho} \geq 0.$$

Using (65), one can obtain

$$\frac{\eta}{2} - \frac{\eta^2 L}{2} - \frac{2\eta^3(1-p)\hat{L}^2}{bp} - \frac{12\eta^2(1-p)Lc_2(\rho, p, b)}{b\rho} - \frac{6\eta^2(1+p)Lc_2(\rho, p, b)}{\rho} \geq 0.$$

Dividing both sides by $\eta$:

$$\frac{1}{2} - \frac{\eta L}{2} - \frac{2\eta^2(1-p)\hat{L}^2}{bp} - \frac{12\eta(1-p)Lc_2(\rho, p, b)}{b\rho} - \frac{6\eta(1+p)Lc_2(\rho, p, b)}{\rho} \geq 0.$$

Multiplying the left side by 2 and entering a variable $r = \eta L$,

$$1 - r - \frac{4(1-p)r^2\hat{L}^2}{bpL^2} - \frac{24(1-p)c_2(\rho, p, b)r}{b\rho} - \frac{12(1+p)c_2(\rho, p, b)r}{\rho} \geq 0. \tag{68}$$

Consequently, we could consider the next inequality

$$1 - r - \frac{4(1-p)r^2\hat{L}^2}{bpL^2} - \frac{36c_2(\rho, p, b)r}{\rho} \geq 0. \tag{69}$$

Since $\frac{24(1-p)}{b} + 12(1+p) \leq 36$, if $r_0 = \eta_0 L$ satisfies (69), then $r_0$ satisfies (68) too. Hence, we could solve (69) to find a bound on $r$. Therefore,

$$r \leq \frac{-\left(1 + \frac{36c_2(\rho, p, b)}{\rho}\right) + \sqrt{\left(1 + \frac{36c_2(\rho, p, b)}{\rho}\right)^2 + \frac{16(1-p)\hat{L}^2}{bpL^2}}}{\frac{8(1-p)\hat{L}^2}{bpL^2}}$$

$$= \frac{2}{\left(1 + \frac{36c_2(\rho, p, b)}{\rho}\right) + \sqrt{\left(1 + \frac{36c_2(\rho, p, b)}{\rho}\right)^2 + \frac{16(1-p)\hat{L}^2}{bpL^2}}}.$$

Then,

$$\eta \leq \frac{2}{L\left(\left(1 + \frac{36c_2(\rho, p, b)}{\rho}\right) + \sqrt{\left(1 + \frac{36c_2(\rho, p, b)}{\rho}\right)^2 + \frac{16(1-p)\hat{L}^2}{bpL^2}}\right)}.$$

Using $(a + b)^2 \leq 2a^2 + 2b^2$, we claim that

$$\eta \leq \frac{2}{L\left(\left(1 + \frac{36c_2(\rho, p, b)}{\rho}\right) + \sqrt{2 + \frac{2592c_2^2(\rho, p, b)}{\rho^2} + \frac{16(1-p)\hat{L}^2}{bpL^2}}\right)}. \tag{70}$$

**Third part**

From the third requirement one can obtain

$$\frac{\eta L^2}{m} + \left(1 - \frac{\rho}{2}\right)s_1 + 2\widetilde{C}\left(\frac{(1-p)\hat{L}^2 C_0}{b} + \frac{3(1+p)\hat{L}^2 s_2}{\rho}\right) - s_1 \leq 0. \tag{71}$$

Substituting $C_0$ in (71):

$$\frac{\eta L^2}{m} - \frac{\rho}{2}s_1 + \frac{2\widetilde{C}(1-p)\hat{L}^2}{b}\left(\frac{\eta}{mp} + \frac{6s_2}{\rho}\right) + \frac{6\widetilde{C}(1+p)\hat{L}^2 s_2}{\rho} \leq 0.$$

Hence, we get

$$\frac{\eta L^2}{m} - \frac{\rho}{2}s_1 + \frac{2\widetilde{C}(1-p)\hat{L}^2\eta}{bmp} + \frac{12s_2\widetilde{C}(1-p)\hat{L}^2}{b\rho} + \frac{6\widetilde{C}(1+p)\hat{L}^2 s_2}{\rho} \leq 0.$$

Combining two last terms:

$$\frac{\eta L^2}{m} - \frac{\rho}{2}s_1 + \frac{2\widetilde{C}(1-p)\hat{L}^2\eta}{bmp} + \frac{\widetilde{C}\hat{L}^2 s_2}{\rho}\left(\frac{12(1-p)}{b} + 6(1+p)\right) \leq 0.$$

Grouping terms with $\eta$:

$$\eta\left(\frac{L^2}{m} + \frac{2\widetilde{C}(1-p)\hat{L}^2}{bmp}\right) - \frac{\rho}{2}s_1 + \frac{\widetilde{C}\hat{L}^2 s_2}{\rho}\left(\frac{12(1-p)}{b} + 6(1+p)\right) \leq 0.$$

Using the (65), one can obtain

$$\eta\left(\frac{L^2}{m} + \frac{2\widetilde{C}(1-p)\hat{L}^2}{bmp}\right) + \frac{\widetilde{C}Lc_2(\rho,p,b)}{\rho m}\left(\frac{12(1-p)}{b} + 6(1+p)\right) \leq \frac{c_1(\rho,p,b)\hat{L}^2\rho}{2mL}.$$

Consequently,

$$\frac{2\eta L^2}{\rho m}\left(1 + \frac{2\widetilde{C}(1-p)\hat{L}^2}{bpL^2}\right) + \frac{2\widetilde{C}Lc_2(\rho,p,b)}{\rho^2 m}\left(\frac{12(1-p)}{b} + 6(1+p)\right) \leq \frac{c_1(\rho,p,b)\hat{L}^2}{mL}.$$

Multiplying both sides by $\frac{m}{L}$:

$$\frac{2\eta L}{\rho}\left(1 + \frac{2\widetilde{C}(1-p)\hat{L}^2}{bpL^2}\right) + \frac{2\widetilde{C}c_2(\rho,p,b)}{\rho^2}\left(\frac{12(1-p)}{b} + 6(1+p)\right) \leq \frac{c_1(\rho,p,b)\hat{L}^2}{L^2}. \tag{72}$$

Then, we can consider next inequality

$$\frac{2\eta L}{\rho}\left(1 + \frac{2\widetilde{C}(1-p)\hat{L}^2}{bpL^2}\right) + \frac{36\widetilde{C}c_2(\rho,p,b)}{\rho^2} \leq c_1(\rho,p,b), \tag{73}$$

where we use $\frac{12(1-p)}{b} + 6(1+p) \leq 18 - 6p \leq 18$. Hence, if we choose $\eta$ equal to some $\eta_0$ at which (73) holds, then (72) holds too. Therefore, we can bound $\eta$:

$$\eta \leq \frac{\frac{\rho c_1(\rho,p,b)\hat{L}^2}{L^2} - \frac{36\widetilde{C}c_2(\rho,p,b)}{\rho}}{2L\left(1 + \frac{2\widetilde{C}(1-p)\hat{L}^2}{bpL^2}\right)}.$$

Using $\hat{L} \geq L$:

$$\eta \leq \frac{\rho c_1(\rho,p,b) - \frac{36\widetilde{C}c_2(\rho,p,b)}{\rho}}{2L\left(1 + \frac{2\widetilde{C}(1-p)\hat{L}^2}{bpL^2}\right)}. \tag{74}$$

**Fourth part**

From the fourth requirement, we get:

$$\frac{3\eta^2 s_1}{\rho} + \left(1 - \frac{\rho}{2}\right)s_2 + 2\eta^2\left(\frac{(1-p)\hat{L}^2 C_0}{b} + \frac{3(1+p)\hat{L}^2 s_2}{\rho}\right) - s_2 \leq 0. \tag{75}$$

Substituting the (66), we have

$$\frac{3\eta^2 s_1}{\rho} - \frac{\rho}{2}s_2 + \frac{2(1-p)\eta^3 \hat{L}^2}{bmp} + \frac{12(1-p)\eta^2 \hat{L}^2 s_2}{b\rho} + \frac{6\eta^2(1+p)\hat{L}^2 s_2}{\rho} \le 0.$$

Then, after combining last two terms, we get

$$\frac{3\eta^2 s_1}{\rho} - \frac{\rho}{2}s_2 + \frac{2(1-p)\eta^3 \hat{L}^2}{bmp} + \frac{\eta^2 \hat{L}^2 s_2}{\rho}\left(\frac{12(1-p)}{b} + 6(1+p)\right) \le 0.$$

Using the (65), one can obtain

$$\frac{3\eta^2 c_1(\rho,p,b)\hat{L}^2}{m\rho L} - \frac{\rho c_2(\rho,p,b)L}{2m\hat{L}^2} + \frac{2(1-p)\eta^3 \hat{L}^2}{bmp}$$
$$+ \frac{\eta^2 L c_2(\rho,p,b)}{\rho m}\left(\frac{12(1-p)}{b} + 6(1+p)\right) \le 0.$$

Consequently,

$$\frac{\eta L}{\rho}\left(\frac{3\eta c_1(\rho,p,b)\hat{L}^2}{mL^2} + \frac{2\rho(1-p)\eta^2 \hat{L}^2}{bmpL} + \frac{\eta c_2(\rho,p,b)}{m}\left(\frac{12(1-p)}{b} + 6(1+p)\right)\right)$$
$$- \frac{\rho c_2(\rho,p,b)L}{2m\hat{L}^2} \le 0. \tag{76}$$

If we choose $\eta \le \frac{\rho}{L}$, then we could consider next inequality

$$\frac{3\eta c_1(\rho,p,b)\hat{L}^2}{mL^2} + \frac{2\rho(1-p)\eta^2 \hat{L}^2}{bmpL} + \frac{\eta c_2(\rho,p,b)}{m}\left(\frac{12(1-p)}{b} + 6(1+p)\right)$$
$$- \frac{\rho c_2(\rho,p,b)L}{2m\hat{L}^2} \le 0. \tag{77}$$

If (77) holds for some $\eta_0$, where $\eta_0 L \le \rho$, then (76) holds respectively. Hence, we could solve (77) relative to $\eta$. For convenience, multiply both sides of the equation by $m$:

$$\frac{3\eta c_1(\rho,p,b)\hat{L}^2}{L^2} + \frac{2\rho(1-p)\eta^2 \hat{L}^2}{bpL} + \eta c_2(\rho,p,b)\left(\frac{12(1-p)}{b} + 6(1+p)\right) - \frac{\rho c_2(\rho,p,b)L}{2\hat{L}^2} \le 0.$$

Moreover, we could use $\frac{12(1-p)}{b} + 6(1+p) \le 18$ and $\rho \le 1$. Therefore, using $L \le \hat{L}$, we can consider

$$\frac{\hat{L}^2}{L^2}(3\eta c_1(\rho,p,b) + 18\eta c_2(\rho,p,b)) + \frac{2(1-p)\eta^2 \hat{L}^2}{bpL} - \frac{\rho c_2(\rho,p,b)L}{2\hat{L}^2} \le 0. \tag{78}$$

Then, if $\eta_0$ satisfies (78), consequently it satisfies (77) and (76). So, we could solve (78):

$$\frac{2(1-p)\eta^2 \hat{L}^2}{bpL} + \frac{\eta \hat{L}^2}{L^2}(3c_1(\rho,p,b) + 18c_2(\rho,p,b)) - \frac{\rho c_2(\rho,p,b)L}{2\hat{L}^2} \le 0.$$

Solving the inequality, we get

$$\eta \le \frac{-(3c_1(\rho,p,b) + 18c_2(\rho,p,b)) + \sqrt{(3c_1(\rho,p,b) + 18c_2(\rho,p,b))^2 + \frac{8(1-p)}{bp}\frac{\rho c_2(\rho,p,b)}{2}}}{\frac{4(1-p)\hat{L}^2}{bpL}}$$

$$= \frac{4\rho c_2(\rho,p,b)}{4L\left((3c_1(\rho,p,b) + 18c_2(\rho,p,b)) + \sqrt{(3c_1(\rho,p,b) + 18c_2(\rho,p,b))^2 + \frac{8(1-p)}{bp}\frac{\rho c_2(\rho,p,b)}{2}\frac{L^4}{\hat{L}^4}}\right)}$$

$$= \frac{\rho c_2(\rho,p,b)}{L\left((3c_1(\rho,p,b) + 18c_2(\rho,p,b)) + \sqrt{(3c_1(\rho,p,b) + 18c_2(\rho,p,b))^2 + \frac{8(1-p)}{bp}\frac{\rho c_2(\rho,p,b)}{2}\frac{L^4}{\hat{L}^4}}\right)}.$$

Using that $(a+b)^2 \leq 2a^2 + 2b^2$ and $\frac{L^4}{\widetilde{L}^4} \leq \frac{\hat{L}^2}{L^2}$, we can give a bit rough estimate of $\eta$:

$$\eta \leq \frac{\rho c_2(\rho, p, b)}{L \left( 3c_1(\rho, p, b) + 18c_2(\rho, p, b) + \sqrt{18c_1^2(\rho, p, b) + 648c_2^2(\rho, p, b) + \frac{4(1-p)\rho c_2(\rho, p, b)}{bp} \frac{\hat{L}^2}{L^2}} \right)}. \tag{79}$$

**Selection of $c_1(\rho, p, b)$ and $c_2(\rho, p, b)$**

Let us take these parameters in the following way

$$c_1(\rho, p, b) = 2\widetilde{C}(1+\rho)\left( \sqrt{\frac{(1-p)\hat{L}^2}{bpL^2}} + \frac{1}{\widetilde{C}} \right),$$

$$c_2(\rho, p, b) = \frac{\rho^2}{18\widetilde{C}}.$$

From (70), we get

$$\eta \leq \frac{2}{L \left( \left(1 + \frac{2\rho}{\widetilde{C}}\right) + \sqrt{2 + \frac{8\rho^2}{\widetilde{C}^2} + \frac{16(1-p)\hat{L}^2}{bpL^2}} \right)}.$$

Consequently, we could roughen the estimate by $\rho \leq 1$:

$$\eta \leq \frac{2}{L \left( \left(1 + \frac{2}{\widetilde{C}}\right) + \sqrt{2 + \frac{8}{\widetilde{C}^2} + \frac{16(1-p)\hat{L}^2}{bpL^2}} \right)}. \tag{80}$$

From (74), one can obtain

$$\eta \leq \frac{2\widetilde{C}(\rho + \rho^2)\left( \sqrt{\frac{(1-p)\hat{L}^2}{bpL^2}} + \frac{1}{\widetilde{C}} \right) - 2\rho}{2L \left(1 + \frac{2\widetilde{C}(1-p)\hat{L}^2}{bpL^2}\right)}.$$

Hence, final bound is

$$\eta \leq \frac{2\rho^2 + 2\widetilde{C}(\rho^2 + \rho)\sqrt{\frac{(1-p)\hat{L}^2}{bpL^2}}}{L \left(1 + \frac{2\widetilde{C}(1-p)\hat{L}^2}{bpL^2}\right)}. \tag{81}$$

From (79), we have

$$\eta \leq \frac{\rho^3}{18\widetilde{C}L \left( 6\widetilde{C}(1+\rho)\left( \sqrt{\frac{(1-p)\hat{L}^2}{bpL^2}} + \frac{1}{\widetilde{C}} \right) + \frac{\rho^2}{\widetilde{C}} + \sqrt{72\widetilde{C}^2(1+\rho)^2 \left( \sqrt{\frac{(1-p)\hat{L}^2}{bpL^2}} + \frac{1}{\widetilde{C}} \right)^2 + \frac{2\rho^4}{\widetilde{C}^2} + \frac{2(1-p)\rho^3 \hat{L}^2}{9\widetilde{C}bp}L^2} \right)}.$$

Using that $(a+b)^2 \leq 2a^2 + 2b^2$ and $\rho \leq 1$, we claim

$$\eta \leq \frac{\rho^3}{18\widetilde{C}L \left( 12 + \frac{1}{\widetilde{C}} + 12\widetilde{C}\sqrt{\frac{(1-p)\hat{L}^2}{bpL^2}} + \sqrt{288 + \frac{2}{\widetilde{C}^2} + \frac{288\widetilde{C}^2(1-p)\hat{L}^2}{bpL^2} + \frac{2(1-p)\hat{L}^2}{9\widetilde{C}bpL^2}} \right)}. \tag{82}$$

From $\eta \leq \frac{\rho}{L}$ and bounds (80), (81) and (82) the next result follows:

$$\Phi_{k+1} \leq \Phi_k - \frac{\eta}{2}\mathbb{E}\|\nabla F(\bar{x}^k)\|^2.$$

Summarizing over $t$, we claim

$$\frac{1}{N}\sum_{k=0}^{N-1}\mathbb{E}\|\nabla F(\bar{x}^k)\|^2 \leq \frac{2(\Phi_0 - \Phi_k)}{\eta N},$$

where $\Phi_0 = F(x^0) - F^* = \Delta$ because of initialization. Hence, for reaching $\frac{1}{N} \sum_{k=0}^{N-1} \mathbb{E}\|\nabla F(\bar{x}^k)\|^2 \le \epsilon^2$, we need

$$N = \mathcal{O}\left(\frac{L\Delta\left(1 + \sqrt{\frac{(1-p)\hat{L}^2}{bpL^2}}\right)}{\rho^3\epsilon^2}\right)$$

iterations. Choosing $\hat{x}^N$ uniformly from $\{\bar{x}^k\}_{k=0}^{N-1}$, we claim the final result. $\qquad\square$

### D.4 PROOF OF COROLLARY 3.5

*Proof.* First, we need to clarify that multi-stage consensus technique allows to avoid $\chi^3$ factor in Theorem 3.4, but apply $\chi$ to a number of communications. Hence, choosing $b = \frac{\sqrt{n}\hat{L}}{L}, p = \frac{b}{n+b}$, we get

$$N_{comm} = \mathcal{O}\left(\frac{\chi L\Delta\left(1 + \sqrt{\frac{n\hat{L}^2}{b^2L^2}}\right)}{\epsilon^2}\right) = \mathcal{O}\left(\frac{\chi L\Delta}{\epsilon^2}\right).$$

Moreover, number of local computations (in average) is equal to

$$n + N_{comm}(pn + (1-p)b) = n + C\frac{\chi L\Delta}{\epsilon^2}\left(\frac{2n\sqrt{n}\frac{\hat{L}}{L}}{n + \sqrt{n}\frac{\hat{L}}{L}}\right) \le n + C\chi\frac{\sqrt{n}\hat{L}\Delta}{\epsilon^2}$$

$$= \mathcal{O}\left(n + \frac{\sqrt{n}\hat{L}\Delta}{\epsilon^2}\right),$$

where $C$ is a constant from $\mathcal{O}(\cdot)$. This finishes the proof. $\qquad\square$

### D.5 LOWER BOUNDS FOR NONCONVEX SETTING

The main idea of lower bound construction is to provide an example of a bad function for which we can estimate the minimum required number of iterations or oracle calls to solve the problem. Hence, we need to consider some class of problems, oracles, and algorithms among which we shall dwell.

Before we start, let us propose some additional facts for a clear proof.

Consider the next function:

$$l(x) = -\Psi(1)\Phi([x]_1) + \sum_{j=2}^{d}\left(\Psi(-[x]_{j-1})\Phi(-[x]_j) - \Psi([x]_{j-1})\Phi([x]_j)\right), \qquad (83)$$

where

$$\Psi(z) = \begin{cases} 0 & z \le \frac{1}{2}; \\ \exp\left(1 - \frac{1}{(2z-1)^2}\right) & z > \frac{1}{2}, \end{cases}$$

$$\Phi(z) = \sqrt{e}\int_{-\infty}^{z} e^{-\frac{t^2}{2}}\, dt. \qquad (84)$$

It has already been shown in Arjevani et al. (2023) (see Lemma 2) that $l(x)$ satisfies the following properties:

1. $\forall x \in \mathbb{R}^d \ l(x) - \inf_x l(x) \le \Delta_0 d$ with $\Delta_0 = 12$.

2. $l(x)$ is $L_0$-smooth with $L_0 = 152$.

3. $\forall x \in \mathbb{R}^d \ \|\nabla l(x)\|_\infty \leq G_0$ with $G_0 = 23$.

4. $\forall x \in \mathbb{R}^d : [x]_d = 0 \ \|\nabla l(x)\|_\infty \geq 1$.

Moreover, let us introduce the next definition

$$\text{prog}(x) = \begin{cases} 0 & x = 0; \\ \max_{1 \leq j \leq d}\{j : [x]_j \neq 0\} & \text{otherwise.} \end{cases} \tag{85}$$

Hence, the function $f$ is called zero-chain, if

$$\text{prog}(\nabla f(x)) \leq \text{prog}(x) + 1.$$

This means that if we start at point $x = 0$, after a gradient estimation we earn at most one non-zero coordinate of $x$. What is more, $l(x)$ is zero-chain function.

Let us formulate an auxiliary lemma which helps to estimate the lower bound.

**Lemma D.9.** *Consider the function $l(x)$ which is defined above. Suppose that*

$$\hat{l}_1(x) = -\Psi(1)\Phi([x]_1) + \sum_{j \ odd; \ j \geq 2} \left( \Psi(-[x]_{j-1})\Phi(-[x]_j) - \Psi([x]_{j-1})\Phi([x]_j) \right),$$

$$\hat{l}_2(x) = \sum_{j \ even} \left( \Psi(-[x]_{j-1})\Phi(-[x]_j) - \Psi([x]_{j-1})\Phi([x]_j) \right).$$

*Hence, if we divide $\hat{l}_i(x)$ into $n$ parts in the following way:*

$$\hat{l}_i(x) = \frac{1}{n}\sum_{k=1}^n \hat{l}_{ik}(x),$$

*where*

$$\hat{l}_{1k}(x) = \begin{cases} -n\Psi(1)\Phi([x]_1) + \sum_{j \geq 2, \ j \equiv 1 \ mod \ 2n} n\left(\Psi(-[x]_{j-1})\Phi(-[x]_j) - \Psi([x]_{j-1})\Phi([x]_j)\right), & k = 1; \\ \sum_{j \equiv 2k-1 \ mod \ 2n} n\left(\Psi(-[x]_{j-1})\Phi(-[x]_j) - \Psi([x]_{j-1})\Phi([x]_j)\right), & k > 1; \end{cases}$$

$$\hat{l}_{2k}(x) = \sum_{j \equiv 2k \ mod \ 2n} n\left(\Psi(-[x]_{j-1})\Phi(-[x]_j) - \Psi([x]_{j-1})\Phi([x]_j)\right),$$

*then*

$$\frac{1}{n}\sum_{k=1}^n \|\nabla\hat{l}_{ik}(y) - \nabla\hat{l}_{ik}(x)\|^2 \leq nL_0^2\|y - x\|^2$$

*for $i = 1, 2$ and for all $x, y \in \mathbb{R}^d$.*

*Proof.* Let us consider the structure of $\nabla\hat{l}_{ik}(x)$. This part of $\hat{l}_i(x)$ depends only on some coordinates of $x$. Hence, given the definition of each slice, we can identify which coordinates of $\hat{l}_{ik}(x)$ can be non-zero. For example, $\nabla\hat{l}_{11}(x)$ can be non-zero only in components $1, 2n, 2n + 1, 4n, 4n + 1, \ldots$ because this function depends only on these coordinates.

Moreover, since $n \geq 2$ (when $n = 1$, the fact above is obvious), if we consider $\hat{l}_{ik}(x)$ and $\hat{l}_{ij}(x)$, then there is no intersection of sets of potentially non-zero coordinates of gradients of these functions due to the construction. Using that full gradient is

$$\nabla\hat{l}_i(x) = \frac{1}{n}\sum_{k=1}^n \nabla\hat{l}_{ik}(x),$$

one can obtain

$$\frac{1}{n}\sum_{k=1}^n \|\nabla\hat{l}_{ik}(y) - \nabla\hat{l}_{ik}(x)\|^2 = n\|\nabla\hat{l}_i(y) - \nabla\hat{l}_i(x)\|^2 \leq nL_0^2\|y - x\|^2.$$

$\square$

*Remark* D.10. Lemma D.9 asserts that in essence the function under consideration and its pieces satisfy the assumptions from Theorem 4.5. The main effect consists of the scaling factor $\frac{1}{\sqrt{n}}$.

**Proof of Theorem 4.5**

*Proof.* We need to introduce functions $F_i$, structure of a time-varying graphs and mixing matrices respectively to construct the lower bound. Then, we can consider next functions

$$l_1(x) = \frac{m}{\left\lceil \frac{m}{3} \right\rceil} \left( -\Psi(1)\Phi([x]_1) + \sum_{j \text{ odd}} \left( \Psi(-[x]_{j-1})\Phi(-[x]_j) - \Psi([x]_{j-1})\Phi([x]_j) \right) \right),$$

$$l_2(x) = \frac{m}{\left\lceil \frac{m}{3} \right\rceil} \left( \sum_{j \text{ even}} \left( \Psi(-[x]_{j-1})\Phi(-[x]_j) - \Psi([x]_{j-1})\Phi([x]_j) \right) \right).$$

As a sequence of graphs, we take star graphs, for each of which the center changes with time according some rules, which we explain later. We derive the mixing matrix from the Laplacian matrix of the graph at the moment $t$ in the next way:

$$\mathbf{W}(t) = \mathbf{I} - \frac{1}{\lambda_{max}(L(t))} L(t).$$

This matrix is obviously a mixing matrix by reason of symmetry and doubly stochasticity. Moreover, $\rho(t) = 1 - \mu_2(\mathbf{W}(t))$, where $\mu_2(\mathbf{W}(t))$ is the second largest eigenvalue of $\mathbf{W}(t)$. Consequently, using the spectrum of $L(t)$, one can obtain that $\rho(t) = \rho = \frac{1}{m}$.
Let us specify the functions $F_i$ at each node:

$$F_i(x) = \begin{cases} \frac{LC^2}{3L_0} l_1\left(\frac{x}{C}\right) & 1 \le i \le \left\lceil \frac{m}{3} \right\rceil \Leftrightarrow i \in S_1, \\ \frac{LC^2}{3L_0} l_2\left(\frac{x}{C}\right) & \left\lceil \frac{m}{3} \right\rceil + 1 \le i \le 2\left\lceil \frac{m}{3} \right\rceil \Leftrightarrow i \in S_2, \\ 0 & \text{otherwise} \Leftrightarrow i \in S_3, \end{cases}$$

where we clarify $C$ later.
Also we need to separate each function into $n$ blocks. It is enough to divide $F_i(x)$ according to Lemma D.9 with corresponding multiplicative constants. Therefore, since $l_1(x)$ and $l_2(x)$ are $3L_0$-smooth, $F_i(x)$ is $L$-smooth for every $C > 0$.
We also can bound $F(0) - \inf_x F(x)$ using

$$F(0) - \inf_x F(x) \le \frac{1}{m} \sum_{i=1}^m (F_i(x) - \inf_x F_i(x)) \le \frac{LC^2 \Delta_0 d}{3L_0}.$$

Hence, we need

$$\frac{LC^2 \Delta_0 d}{3L_0} \le \Delta.$$

Now we are ready to divide our proof into three parts.
**Number of communications**
We want the transfer of information between sets $S_1$ and $S_2$ to not occur for as long as possible. This requires that the center of the star graph is not a vertex from $S_1$ or $S_2$, or it is not a vertex of $S_3$ that already has information from other sets of vertices. Therefore, let us specify the changes of the graphs with time according to the following principle: first we go through all the vertices of the set $S_3$, and after that we choose the vertex that allows the exchange of information between $S_1$ and $S_2$. Then, mentioning that $\frac{1}{m} \sum_{i=1}^m F_i(x) = \frac{LC^2}{3L_0} l\left(\frac{x}{C}\right)$ and

$$\text{prog}(\nabla F_i(x)) \begin{cases} = \text{prog}(x) + 1 & (\text{prog}(x) \text{ is even and } i \in S_1) \text{ or } (\text{prog}(x) \text{ is odd and } i \in S_2); \\ \le \text{prog}(x) & \text{otherwise,} \end{cases}$$

we claim that for increasing the $\text{prog}(x)$ at 1 we need at least $m - 2\lceil\frac{m}{3}\rceil + 1$ iterations (without considering local computations). Therefore, after $N$ iterations

$$\text{prog}(N) = \max_{1 \leq i \leq m,\ 0 \leq t \leq N} \text{prog}(x_i^t) \leq \left\lfloor \frac{N}{m - 2\lceil\frac{m}{3}\rceil + 1} \right\rfloor + 1.$$

Also it is easy to make sure that if $m \geq 3$, then $m - 2\lceil\frac{m}{3}\rceil + 1 \geq \frac{m}{4}$. Then

$$\text{prog}(N) \leq \left\lfloor \frac{4N}{m} \right\rfloor + 1.$$

**Number of local computations**
Here we use the same idea as in first part. Let us consider the next oracle computation: we take one of pieces on each node uniformly, i.e. $\mathbb{P}\{\text{block with index } k \text{ is chosen}\} = \frac{1}{n}$ for every $k = 1, \ldots, n$. Hence, at the current moment, we need a **specific** piece of function, because according to structure of $l(x)$, each gradient estimation can "defreeze" at most one component and only a computation on a certain block makes it possible. Let us define the number of required gradient calculations as $n_{avg}$. Therefore,

$$\mathbb{E}\{n_{avg}\} = \sum_{i=1}^{\infty} \frac{i}{n} \left( \frac{n-1}{n} \right)^{i-1} = n,$$

where $\frac{1}{n}\left(\frac{n-1}{n}\right)^{i-1}$ is a is the probability that at $i$-th moment we take the correct piece. Thus, after $K$ local computations on each node we can change at most $\lfloor\frac{K}{n}\rfloor + 1$ coordinates.
**Final result**
Hence, if considered algorithm makes $N$ communications and $K$ local computations on each node, then

$$\text{prog}(N, K) = \max_{1 \leq i \leq m,\ 0 \leq t \leq N} \text{prog}(x_i^t) \leq \min\left( \left\lfloor \frac{4N}{m} \right\rfloor + 1, \left\lfloor \frac{K}{n} \right\rfloor + 1 \right)$$

Consequently, for every $N \geq \frac{m}{4}$ and $K \geq n$ consider

$$d = 2 + \min\left( \left\lfloor \frac{4N}{m} \right\rfloor, \left\lfloor \frac{K}{n} \right\rfloor \right).$$

It is easy to verify thar

$$d < \min\left( \frac{16N}{m}, \frac{4K}{n} \right).$$

Moreover, we choose $C$ as

$$C = \left( \frac{3L_0\Delta}{L\Delta_0 \min\left(\frac{16N}{m}, \frac{4K}{n}\right)} \right)^{\frac{1}{2}}.$$

Hence, clarifying that $\text{prog}(N, K) < d$, we have

$$\mathbb{E}\|\nabla F(\hat{x}_N)\|^2 \geq \min_{[x]_d=0}\|\nabla F(\hat{x}_N)\|^2 = \frac{L^2 C^2}{9L_0^2} \min_{[x]_d=0}\|\nabla l(\hat{x}_N)\|^2 \geq \frac{L^2 C^2}{9L_0^2}$$

$$= \max\left( \frac{L\Delta m}{48NL_0\Delta_0}, \frac{L\Delta n}{12KL_0\Delta_0} \right) \geq \frac{L\Delta m}{96NL_0\Delta_0} + \frac{L\Delta n}{24KL_0\Delta_0}$$

$$= \Omega\left( \frac{L\Delta m}{N} + \frac{L\Delta n}{K} \right),$$

where the second inequality holds from fourth property of $l(x)$.
Consequently, applying Lemma D.9 to $\{F_i\}_{i=1}^m$ and noting that $\chi = \Theta(m)$, we finish the proof. $\qquad\square$

