# OpenReview forum: "Decentralized Finite-Sum Optimization over Time-Varying Networks"
_ICLR.cc/2025/Conference — Submitted to ICLR 2025_

### Official Review · Reviewer_SSdw · 2024-10-23

**Soundness:** 3
**Presentation:** 3
**Contribution:** 2
**Rating:** 5
**Confidence:** 4

**Summary:**

This paper studies decentralized finite-sum optimization over time-varying networks for smooth objectives.

In the strongly-convex setting, the ADOM+VR algorithm consists in mixing the ADOM+ (accelerated decentralized optimization for time-varying graphs) and Katyusha (Accelerated variance-reduced single-machine) algorithms. Similarly, the lower bound mixes the ones from ADOM+ and ADFS.

In the non-convex setting, gradient tracking (a standard way to obtain "exact" decentralized algorithms) is combined with the PAGE VR algorithm. Similarly, the lower bound mixes that of Yuan et al (2022) and ADOM+.

In both cases, combining existing optimal approaches yields (almost) matching upper and lower complexity bounds. Toy experiments are given to illustrate the practical performances.

**Strengths:**

- the overall approach is clear and natural
- (almost) matching upper and lower bounds
- results for both strongly convex and non-convex settings

**Weaknesses:**

- Contributions are incremental: mostly combining accelerated variance-reduced estimators with accelerated decentralized algorithms over time-varying graphs (or gradient tracking for the non-convex setting)
- The setting (finite-sum decentralized optimization over time-varying graphs) is rather niche
- The algorithms are quite rigid, with long communication stages (through $W(k)$) and long computation stages (through mini-batching), which is effective but not very elegant, and unlikely to extend well to, e.g., asynchronous settings. Approaches like DADAO (Nabli and Oyallon, 2023) are likely to lead to better solutions, as well as maybe close the gap between lower and upper bounds in the strongly-convex setting.
- The non-convex algorithm has a pretty bad dependence on the graph constants ($\chi^3$), and is only saved by multi-consensus steps (which again, would likely break in asynchronous settings for instance).

**Questions:**

- It is said in Corollary 3.3 that the number of communications per iteration is of order $\chi$, but in Algorithm 1 there is only one communication per iteration (matrix $W(k)$ is used). I understand that we should use Algorithm 1 with the multi-consensus matrix $W(k, \kappa)$ instead, is that true? This should be clarified.

---

> ### Author Response · Authors · 2024-12-03
> **Answer to Reviewer SSdw**
>
> Dear Reviewer,
>
> Thank you for your time and effort while reviewing our paper. Below we answer the questions you have raised.
>
> **Incremental contribution and niche setting.**
>
> Please see the common answer to Reviewers.
>
> **Algorithms use multi-consensus and may perform badly in asynchronous setting.**
>
> Our algorithms are not designed for the asynchronous setting, but still GT-PAGE is optimal in its setup. Multi-step communication is a typical trick in decentralized optimization [1, 2]. In our understanding, this trick is mainly needed to eliminate the additional factor $\chi$ in the number of gradient calls, i.e. the trick is mostly theoretical. Anyway, using this technique leads to optimal complexities, which is effective, as you mentioned. Moreover, Acc-GT [5], which is an optimal method for time-varying networks, has a network dependence $\chi^{3/2}$ without multi-consensus. Finally, you mentioned that our methods are *unlikely to extend well* to asynchronous setup, but this fact cannot be quickly checked in our discussion.
>
> **DADAO**
>
> Thank you for pointing out the DADAO paper. We understand your comment that in the asynchronous setup DADAO is *likely to lead* to better solutions and *may* close the complexity gap. But please see that our paper already provides methods and closes the gap in the nonconvex setting. Therefore, we acknowledge that other approaches are possible but do not see it as a weakness of our work.
>
>
> **Questions**
>
> *  Multi-consensus matrix.
>
> Thank you for raising a question on multi-consensus. It was meant that each $W(k)$ is replaced by a product of $T$ consequent matrices $\tilde W(kT, T) = W(kT + (T - 1)) W(kT + (T - 2))\ldots W(kT)$ that corresponds to $T$ consequent communication rounds.
>
>
>
> **References**
>
> [1] Scaman, Kevin, et al. "Optimal algorithms for smooth and strongly convex distributed optimization in networks." international conference on machine learning. PMLR, 2017.
>
> [2] Kovalev, Dmitry, et al. "Lower bounds and optimal algorithms for smooth and strongly convex decentralized optimization over time-varying networks." Advances in Neural Information Processing Systems 34 (2021): 22325-22335.
>
> [3] URL: https://scikit-learn.org/stable/modules/generated/sklearn.linear_model.LogisticRegression.html
>
> [4] URL: https://github.com/scikit-learn/scikit-learn/blob/main/sklearn/linear_model/_sag.py#L87
>
> [5] Li, Huan, and Zhouchen Lin. "Accelerated gradient tracking over time-varying graphs for decentralized optimization." Journal of Machine Learning Research 25.274 (2024): 1-52.

---

### Official Review · Reviewer_CnvV · 2024-11-02

**Soundness:** 3
**Presentation:** 2
**Contribution:** 2
**Rating:** 5
**Confidence:** 4

**Summary:**

This paper studies stochastic decentralized optimization problems of strongly-convex and non-convex settings over static and time-varying networks. For the strongly-convex setting, the ADOM+VR algorithm is proposed and the convergence property is established; For the non-convex setting, the GT-PAGE algorithm is proposed and the convergence property is established. The authors also study the lower bounds of both strongly-convex and non-convex decentralized optimization problems theoretically. The efficacy of the proposed algorithms is validated in simulations.

**Strengths:**

The topic of the work is well aligned with ICLR community and it is significant to extend the lower bound analysis for decentralized optimization problems. The work also contains great amount of theoretical work.

**Weaknesses:**

1. The organization of the manuscript is not clear and different pieces (section 3.2, 3.3, 4.1) are isolated from each other. Since they are presented in this manuscript, it is assumed they have some internal relationship. For example, is the optimal non-convex algorithm, GT-PAGE, also optimal in the strongly-convex setting? Why is the strongly-convex algorithm based on ADOM+ algorithm while the non-convex algorithm is based on PAGE algorithm?

2. It is concerned that the Assumption 2.5 is too strong, i.e., assuming the existence of $\chi \geq 1$ (also why $\geq 1$ rather than $ > 1$?).  This work studies time-varying networks and the matrices $W(k)$ change at every time step $k$. How is it guaranteed that such $\chi$ exists? Are there any other requirements for the network connectivity or matrix construction to guarantee the existence of $\chi$?

**Questions:**

1. The organization clarity (please see weakness-1). In order to better elaborate the contributions in this work, it is suggested that the authors re-write all bullet points in the contribution paragraph. For the ADOM+VR algorithm, please state the type of optimization problem/network setting and why it is optimal (is it applied to a different optimization problem or the convergence speed is faster than others); For the GT-PAGE algorithm, please also state the type of optimization problem and why it is optimal. It is not strong enough to only mention the elements of different algorithms in the contribution.

2. The achievability of Assumption 2.5 (please see weakness-2).

3. In experiments, are those plots presenting training set performance or testing set performance? It is expected both training and testing performance are shown. Moreover, it would be good to show classification accuracy on test set.

---

> ### Author Response · Authors · 2024-12-03
> **Answer to Reviewer CnvV**
>
> Dear Reviewer, thank you for your thorough review.
>
> **Weaknesses**.
>
> **Paper organization**.
>
> The aim of the paper is to propose optimal algorithms for decentralized finite-sum optimization. In modern optimization theory [3], a problem class and an algorithm class are determined by introducing corresponding assumptions. In our paper, the problem class is decentralized finite-sum problems as defined in Section 2 and the method class is determined as first-order decentralized algorithms in Section 4.1. After that, the iteration complexity of the algorithm is compared to lower complexity bounds for the given problem class. If these two complexities coincide up to a constant term, the corresponding method is called optimal. Optimal methods for strongly convex and nonconvex scenarios use different techniques and have different theoretical analysis. In particular, Nesterov momentum allows to reach optimality in the strongly convex case but is useless in the nonconvex case. Therefore, a method optimal for one scenario cannot be used for the different scenario out-of-the-box even for usual (i.e. not decentralized) optimization.
>
> In our work we used previously existing techniques to build new methods for each of the cases (strongly convex and nonconvex). That is why in the strongly convex case we used ADOM+ and in the nonconvex case we applied PAGE. As mentioned above, a method designed for the strongly convex scenario cannot be directly applied to the nonconvex one and vice versa. We managed to obtain an optimal algorithm for the nonconvex case, closing the previously existing gap in the literature. We also obtained an algorithm approaching optimality for the strongly convex case.
>
> **Assumption 2.5 on $\chi$**.
>
> It is possible to construct a matrix sequence $W(k)$ satisfying Assumption 2.5 under realistic assumptions on the time-varying network. The only restriction on the network is that the graph $\mathcal{G}^k$ is connected at each iteration. After that, we choose $W(k) = L(\mathcal{G}^k) / \lambda_{\max}(L(\mathcal{G}^k)$, where $L(\mathcal{G}^k) = D(\mathcal{G}^k) - A(\mathcal{G}^k)$ denotes the graph Laplacian matrix (see lines 196-198 of our paper; here D(\mathcal{G}^k) denoted the diagonal matrix containing the node degrees and A(\mathcal{G}^k) is the adjacency matrix). For each of the graphs $\mathcal{G}^k$, denote its Laplacian condition number $\chi_k = \frac{\lambda_{\max}(L(\mathcal{G}^k))}{\lambda_{\min}^+(L(\mathcal{G}^k))}$. Since the graph is connected, $\chi < +\infty$, and since $\lambda_{\max}(\mathcal{G}^k)\geq \lambda_{\min}^+(\mathcal{G}^k)$ we have $\chi_k\geq 1$. Moreover, since the set of vertices is fixed, there is a finite number of different graphs on these vertices. It is straightforward to check that $\chi = \max_k \chi_k$ satisfies Assumption 2.5 and $1\leq\chi < +\infty$.
>
> Also note that for a fully-connected graph we have $\chi = 1$. One gossip iteration on a fully connected network corresponds to full averaging, which corresponds to Assumption 2.5. This explains why we do not impose that $\chi > 1$.
>
> Our work does not stick to taking graph Laplacians: Assumption 2.5 allows to use any other matrices that satisfy the gossip requirements.

---

> > ### Author Response · Authors · 2024-12-03
> > **Answer to Reviewer CnVV continues**
> >
> > **Questions**
> >
> > * The organization clarity (please see weakness-1). In order to better elaborate the contributions in this work, it is suggested that the authors re-write all bullet points in the contribution paragraph. For the ADOM+VR algorithm, please state the type of optimization problem/network setting and why it is optimal (is it applied to a different optimization problem or the convergence speed is faster than others); For the GT-PAGE algorithm, please also state the type of optimization problem and why it is optimal. It is not strong enough to only mention the elements of different algorithms in the contribution.
> >
> > See line 053 describing GT-PAGE: “*For nonconvex decentralized optimization over time-varying graphs, we propose an optimal algorithm GT-PAGE (Algorithm 2)*”. This line contains the type of optimization problem: *nonconvex decentralized optimization* and the network setting: *time-varying graphs*. The meaning of optimality is the similarity of complexity bounds of the method and lower complexity bounds for the problem class (see our answer to Weakness 1). We will add the discussion on the definition of optimality in the revised version of the paper. Please also see the common answer to Reviewers.
> >
> > For ADOM+VR, we will update the text as follows: “*We propose a method for **strongly convex** decentralized finite-sum optimization over time-varying graphs ADOM+VR (Algorithm 1)*”.
> >
> > * The achievability of Assumption 2.5 (please see weakness-2).
> >
> > See answer to weakness 2.
> >
> >
> > * In experiments, are those plots presenting training set performance or testing set performance? It is expected both training and testing performance are shown. Moreover, it would be good to show classification accuracy on test set.
> >
> > The plots show only the training set error. Since our theory only covers the convergence speed of the algorithms and does not touch generalization properties, we decided to illustrate our findings by running the methods on the train set and plotting the optimality measure (distance to optimum or gradient norm) and not cover quality metrics such as accuracy.
> >
> >
> > **References**
> >
> > [1] Nedic, Angelia, Alex Olshevsky, and Wei Shi. "Achieving geometric convergence for distributed optimization over time-varying graphs." SIAM Journal on Optimization 27.4 (2017): 2597-2633.
> >
> > [2] Kovalev, Dmitry, et al. "Lower bounds and optimal algorithms for smooth and strongly convex decentralized optimization over time-varying networks." Advances in Neural Information Processing Systems 34 (2021): 22325-22335.
> >
> > [3] Nesterov, Yurii. Lectures on convex optimization. Vol. 137. Berlin: Springer, 2018.

---

### Official Review · Reviewer_2Swy · 2024-11-05

**Soundness:** 3
**Presentation:** 2
**Contribution:** 2
**Rating:** 3
**Confidence:** 4

**Summary:**

This manuscript addresses a decentralized stochastic optimization problem with a finite sample set at each node over time-varying networks. The authors propose two algorithms tailored for strongly convex and non-convex objective functions, respectively, and provide a lower bound analysis to discuss the optimality of the proposed algorithms.

**Strengths:**

- This work provides rich theoretical analytical results for different objective function assumptions. By comparing with the proposed lower bound, it is shown that the proposed algorithm is optimal in their scenario;
- The lower bound in this paper takes into account the smoothness coefficient of each node, which is more refined.

**Weaknesses:**

While this work presents a set of results for distributed stochastic optimization problems, I found the paper difficult to follow, with unclear motivation and insufficient discussion on the necessity of this study. My specific concerns are as follows:

- On the significance of this work: The paper addresses time-varying topologies in distributed networks, but it is unclear why these topologies pose unique challenges for distributed algorithms. In my view, as long as Assumption 2.5 holds (i.e., the topology is connected at each iteration) and multi-round communication acceleration is employed, a time-varying topology does not seem substantially more challenging than that of fixed topology. Thus, this work appears to be tackling a corner case, especially considering that similar problems have been widely studied, e.g., Kovalev et al., 2021a, Luo and Ye, 2022, Huang and Yuan, 2022, Li and Lin, 2021. The authors need to further clarify the novelty and contribution of this work against these exisitng works.
- On the Assumptions on Smoothness: The assumptions regarding smoothness are somewhat confusing. The authors consider three types of smoothness but do not compare their differences or clarify which assumptions are strongest. Additionally, Assumption 2.1 requires that each sample’s objective function is smooth. It would be helpful to discuss whether this holds in typical machine learning tasks and if it is verifiable in practice.
- Regrading the proof of Theorem 3.2, compared to (Kovalev et al., 2021a), it seems to differ in only one constrained VR gradient estimation error (c.f. Lemma B.1), while the other proofs are almost identical to Kovalev et al., 2021a, differing only in some parameter choices. This makes the technical contribution of this paper vague.
- On Readability and Clarity: The paper’s readability could be significantly improved. In the algorithm design section, the authors rely heavily on prior literature to explain their algorithmic approach, offering limited unique insights. This reliance diminishes the perceived novelty of the work. Additionally, the paper contains many symbols that are either used before being defined or left undefined altogether (e.g., $\lambda^{+}_{\text{min}}$), making it challenging to follow.

**Questions:**

- Why the two proposed algorithms require different function smoothness assumptions (c.f. Assumption 2.1, 2.2 and 2.3)?
- The LibSVM dataset used for the experiments in this work seems not adequate; the reviewer would like to know if the algorithm could be applied to more complex datasets such as cifar-10/100 to further validate the effect of the algorithm.
- How is the time-varying topology being implemented in the experiments?
- Why the reference Metelev et al. (2024) is not discussed in the main text? In fact, the time-varying topological sequences used in this work for obtaining lower bound adopt their strategy.

**Details Of Ethics Concerns:**

N.A.

---

> ### Author Response · Authors · 2024-12-03
> **Answer to Reviewer 2Swy**
>
> Dear Reviewer,
>
> Thank you for your thorough review and comments. We are grateful that you acknowledged the strengths of our work.
>
> **Contribution of the work**.
>
> The contributions of our paper are algorithms for decentralized finite-sum optimization over time-varying networks, optimal in the nonconvex case, as well as lower bounds. Previously methods were known for:
>
> * decentralized non-stochastic optimization over time-varying networks (Kovalev et al., 2021a, Li and Lin, 2021);
>
> * decentralized stochastic optimization without finite-sum structure over time-varying networks (Huang and Yuan, 2022);
>
> * finite-sum optimization over static networks (Luo and Ye, 2022).
>
> As you can see, the combination of time-varying graphs and finite-sum structure has not been previously studied. Our work closes the gap in this direction (in the nonconvex case).
>
>
> **Proofs identical to previous works**.
> The main technical challenge lies in the proof of GT-PAGE. Please see the common answer to Reviewers for details.
> **Time-varying topology is not challenging**.
>
> We respectfully disagree that *time-varying topology does not seem substantially more challenging than that of fixed topology*. Static and time-varying networks are two different classes of problems. A typical approach to decentralized problems is describing the constraint set $x_1 = \ldots = x_m$ by an affine constraint involving gossip matrix $W$, i.e. $Wx = 0$. The main obstacle of time-varying networks, even if they stay connected at each iteration, is that matrix $W_k$ changes between iterations, thus changing the description of the constraint set. This obstacle requires new approaches for decentralized problems. Working with time-varying graphs requires additional techniques such as gradient tracking [3,4] and error feedback [2,5]. In fact, after optimal methods for static graphs were obtained in [1] it took four years for the community to obtain optimal methods for time-varying graphs [2].
>
> **Assumptions on Smoothness**.
>
> Assumption 2.1 holds, for example, for logistic regression. Consider one summand of form $f_{ij}(x) = \log(1 + \exp(b_{ij}\langle a_{ij}, x\rangle))$, where $x$ is the problem weight vector, $a_{ij}$ denotes the feature vector and $b_{ij}\in\{-1, 1\}$ is the feature label. It can be checked (i.e. by computing the hessian) that $L_{ij} = \|\|a_{ij}\|\|^2 / 4$.
>
> For the relation of smoothness constants, see line 177: $L\leq \overline{L}\leq nL$, where $n$ is the dimension. This can be seen by triangle inequality and by the fact that for convex smooth functions $g$ and $h$ we have $L(g)\leq L(g + h)$. Analogically it can be shown that $L\leq \hat L\leq \sqrt{n} L$. We will add the corresponding discussion to the revised version of the work.
>
>
> **Paper readability**.
>
> In the results section, we do describe previously known results. However, we believe that our explanations show the way of obtaining new results and think that this increases the readability of the paper. If we do not provide clarifications, our results might seem more challenging and unique, but this will be done at the cost of keeping the reader uninformed of the basics underlying the proposed algorithms. Summing up, we do not see a problem in explaining the existing results.
>
> Thank you for pointing out the undefined symbols. We will correct this issue in the revised version of the paper.
>
>
> **Questions**
>
> * Why the two proposed algorithms require different function smoothness assumptions (c.f. Assumption 2.1, 2.2 and 2.3)?**
>
> The methods are based on their corresponding counterparts in decentralized optimization. In the nonconvex case, we use algorithm PAGE [6] that requires the average smoothness assumption similar to Assumption 2.3. In the strongly convex case, we adopt Katyusha [7] that uses worst-case constants as in Assumptions 2.1 and 2.2. In other words, such choice of assumptions is caused by backward compatibility.
>
> * The LibSVM dataset used for the experiments in this work seems not adequate; the reviewer would like to know if the algorithm could be applied to more complex datasets such as cifar-10/100 to further validate the effect of the algorithm.
>
> Thank you for your suggestion. The choice of simple datasets and models enables to tune the step-sizes and other algorithm parameters according to theory.
>
> * How is the time-varying topology being implemented in the experiments?
>
> It is implemented using random geometric graphs [7].
>
> * Why the reference Metelev et al. (2024) is not discussed in the main text? In fact, the time-varying topological sequences used in this work for obtaining lower bound adopt their strategy.
>
> Thank you for noticing it. Indeed, this reference is used for the lower bounds and we mention it in line 1677 in the appendix. We will mention this paper in the main part of our work.

---

> > ### Author Response · Authors · 2024-12-03
> > **Answer to Reviewer 2Swy continues**
> >
> > **References**
> >
> > [1] Scaman, Kevin, et al. "Optimal algorithms for smooth and strongly convex distributed optimization in networks." international conference on machine learning. PMLR, 2017.
> >
> > [2] Kovalev, Dmitry, et al. "ADOM: accelerated decentralized optimization method for time-varying networks." International Conference on Machine Learning. PMLR, 2021.
> >
> > [3] Li, Huan, and Zhouchen Lin. "Accelerated gradient tracking over time-varying graphs for decentralized optimization." Journal of Machine Learning Research 25.274 (2024): 1-52.
> >
> > [4] Nedic, Angelia, Alex Olshevsky, and Wei Shi. "Achieving geometric convergence for distributed optimization over time-varying graphs." SIAM Journal on Optimization 27.4 (2017): 2597-2633.
> >
> > [5] Kovalev, Dmitry, et al. "Lower bounds and optimal algorithms for smooth and strongly convex decentralized optimization over time-varying networks." Advances in Neural Information Processing Systems 34 (2021): 22325-22335.
> >
> > [6] Li, Zhize, et al. "PAGE: A simple and optimal probabilistic gradient estimator for nonconvex optimization." International conference on machine learning. PMLR, 2021.
> >
> > [7] URL: https://networkx.org/documentation/stable/auto_examples/drawing/plot_random_geometric_graph.html

---

### Official Review · Reviewer_JkLV · 2024-11-05

**Soundness:** 3
**Presentation:** 3
**Contribution:** 3
**Rating:** 8
**Confidence:** 2

**Summary:**

This paper studies decentralized finite-sum optimization problem over time-varying graphs. Theorem 4.3 and Theorem 4.5 establish lower bounds on computational and communication complexities, resp., for strongly convex and non-convex optimization. Furthermore, the paper presented two algorithms ADOM+VR and GT-PAGE, resp., for strongly convex and non-convex optimization while comparing its performance with the state-of-the-art methods both analytically (see Tables 1 & 2) and numerically. Notably, GT-PAGE is optimal by achieving the lower complexity bounds while ADOM+VR is optimal in terms of communication iterations. Some open problems have also been highlighted. The numerical examples for the LibSVM dataset show the superior performance of the algorithm in terms of communication and computational complexities. Interestingly, the optimal algorithm GT-PAGE achieve presents better yet not strongly superior performance in comparison to the state-of-the-art methods.

**Strengths:**

- The paper studies an important problem.
- The paper is well-written.
- Notation, assumptions, and results are presented clearly.
- The established lower bounds on the complexities and presenting optimal algorithms achieving these bounds are solid contributions.
- Numerical examples further strengthen the paper's claims.

**Weaknesses:**

- The paper is fairly technical. To smoothen the paper's technical content and to improve the paper's accessibility, more qualitative discussions can be included.
- The introduction directly jumps to the optimization problem formulation. More motivating examples for the problem formulation would improve the paper.
- In the introduction, Tables 1 and 2 present the complexities in terms of parameters n, L, \mu, and some others. However, it is not clear what these parameters, e.g., L and \mu, refer to.
- Numerical examples do not provide error bars.

**Questions:**

- Can the authors clarify whether there is no need to represent error bars across independent experiments to mitigate the impact of stochasticity in the numerical examples?

---

> ### Author Response · Authors · 2024-12-03
> **Answer to Reviewer JkLV**
>
> Dear Reviewer,
>
> Thank you for your thorough review and comments. We are grateful that you pointed out the strengths of our paper.
>
> **Qualitative discussions**.
>
> Thank you for your suggestion. In the revised version of the paper, we will add a discussion to stress the paper contributions.
>
> **Smoothness parameters in Tables 1 and 2**.
>
> We will add clarifications and links to Assumptions in Section 2 in the annotations to Tables 1 and 2.
>
> **Motivating examples**.
>
> Typically in optimization literature one writes out the problem formulation right away. The examples are usually given in the numerical experiments section, like in our paper.
>
> **Error bars in numerical experiments**.
>
> Thank you for pointing this out. The error bars could be added to the plots, but this is not very typical in optimization papers.

---

### Meta-Review · Area_Chair_SeZo · 2024-12-14

**Metareview:**

While the authors see some possible merit in this work, after the rebuttal and discussion most of the reviewers still view the paper as being below the acceptance threshold (two weakly and one strongly).  There most common recurring concern was the limited technical novelty, and there were also some suggestions on other issues such as paper organization and providing more metrics in experiments.  There could be some routes to strengthening the paper, e.g., handling other settings such as relaxing synchronicity, but the consensus is not to accept in the current form.

**Additional Comments On Reviewer Discussion:**

The discussion was on the low side, but the reviewers confirmed that they read the rebuttal and still maintain their recommendation.

---

### Decision · Program_Chairs · 2025-01-22

Reject